# Perceptual Similarity for Measuring Decision-Making Style and Policy Diversity in Games

**Chiu-Chou Lin**                                                    *dsobscure@outlook.com*
*Department of Computer Science*
*National Yang Ming Chiao Tung University, Hsinchu 30010, Taiwan*

**Wei-Chen Chiu**                                                    *walon@cs.nctu.edu.tw*
*Department of Computer Science*
*National Yang Ming Chiao Tung University, Hsinchu 30010, Taiwan*

**I-Chen Wu**                                                        *icwu@cs.nycu.edu.tw*
*Department of Computer Science*
*National Yang Ming Chiao Tung University, Hsinchu 30010, Taiwan*
*Research Center for Information Technology Innovation*
*Academia Sinica, Taipei 11529, Taiwan*

**Reviewed on OpenReview:** *https://openreview.net/forum?id=30C9AWBW49*

## Abstract

Defining and measuring decision-making styles, also known as playstyles, is crucial in gaming, where these styles reflect a broad spectrum of individuality and diversity. However, finding a universally applicable measure for these styles poses a challenge. Building on *Playstyle Distance*, the first unsupervised metric to measure playstyle similarity based on game screens and raw actions by identifying comparable states with discrete representations for computing policy distance, we introduce three enhancements to increase accuracy: multiscale analysis with varied state granularity, a perceptual kernel rooted in psychology, and the utilization of the intersection-over-union method for efficient evaluation. These innovations not only advance measurement precision but also offer insights into human cognition of similarity. Across two racing games and seven Atari games, our techniques significantly improve the precision of zero-shot playstyle classification, achieving an accuracy exceeding 90% with fewer than 512 observation-action pairs—less than half an episode of these games. Furthermore, our experiments with *2048* and *Go* demonstrate the potential of discrete playstyle measures in puzzle and board games. We also develop an algorithm for assessing decision-making diversity using these measures. Our findings improve the measurement of end-to-end game analysis and the evolution of artificial intelligence for diverse playstyles.

## 1 Introduction

The pursuit of diversity in decision-making is one of the intrinsic motivations that drive human behavior, resulting in individuality and creativity (Rheinberg, 2020). This is evident in the context of games, where decision-making manifests as various playstyles, each reflecting unique strategies and characteristics (Bean & Groth-Marnat, 2016; Yannakakis et al., 2013). Alongside the pursuit of diversity, another central aspect of decision-making focuses on achieving optimal performance. Significant advances in decision-making performance have been witnessed, especially with the development of Deep Reinforcement Learning (DRL) (Russell & Norvig, 2020). The effectiveness of DRL was first showcased in arcade video games (Mnih et al., 2015). Subsequent applications to board games emphasized its potential, achieving superhuman skills (Silver et al., 2018). This success expanded into various types of games, from Agent57's superhuman performance in Atari games to groundbreaking feats in Dota 2 and StarCraft II (Badia et al., 2020; Berner et al., 2019;

Vinyals et al., 2019). Beyond gaming, DRL applications extend to robot control (Andrychowicz et al., 2020) and natural language processing (Ouyang et al., 2022), among others.

Yet, while DRL continues to show promise in diverse applications, understanding and analyzing playstyles with minimal domain-specific knowledge remains a complex endeavor. Data from diverse sources are essential to improve agent strength and efficiency (Fan & Xiao, 2022; Fan et al., 2023), just as acquiring skills from different styles is vital for agents to generalize across tasks (Eysenbach et al., 2019). Although a robust playstyle measure fosters a spectrum of playing strategies, it also reveals the inherent challenges in measuring these styles, particularly in environments without built-in features for style measurement. Consequently, achieving precise playstyle measurement remains a formidable task (Tychsen & Canossa, 2008).

There are several methods to evaluate playstyles or perform player modeling (Yannakakis et al., 2013), from heuristic rules design to in-game feature exploration (Tychsen & Canossa, 2008; Bontchev & Georgieva, 2018; Mader & Tassin, 2022). Supervised learning facilitates the discrimination of playstyles (Brombacher et al., 2017), while unsupervised clustering offers behavioral insights (Drachen et al., 2009; 2013; Ferguson et al., 2020). Another avenue involves contrastive learning to identify playstyles and behaviors among players (McIlroy-Young et al., 2021; Agarwal et al., 2021). Through these methods, the notion of playstyle can be gauged using distance or similarity measures across game datasets, addressing dynamic and evolving challenges in different scenarios. The concept is reflected in the work by Lupu et al. (2021), which specifies policy diversity using the divergence of action distributions but requires parametrized policies.

The recent innovation by Lin et al. (2021) introduces the *Playstyle Distance* measure, which stands out by directly measuring playstyle from game screens and raw action pairs. Unlike common methods that compare latent features or rely on parametrized policies, this approach measures action distribution distances directly from raw gameplay samples, reducing the reliance on game features, predefined style labels or even extensive training sets for learning latent features or policies. Its effectiveness hinges on the critical role of state discretization. By discretizing observations like game screens, we can identify similar and comparable states, allowing for a direct comparison of action distributions of each state, thereby defining the decision-making style based on the expected distance of policies.

While the *Playstyle Distance* offers an advance in end-to-end and unsupervised playstyle measurement, our research endeavors to elevate this foundation. We introduce techniques to improve playstyle measurement, harnessing the advantage of discrete states. Initially, we leverage multiscale analysis with varied state granularity, emulating human judgment of similarity from multiple attributes and viewpoints (Medin et al., 1993). We then derive a perceptual kernel from psychophysics (Fechner, 1966) in psychology to obtain a probabilistic similarity value, which is more in harmony with human comprehension than distance values. Moreover, incorporating the concept of the *Jaccard index* (Murphy, 1996), we broaden the focus of measurement beyond intersection samples, harnessing all observed game data to improve measurement accuracy. These techniques not only improve the precision of *Playstyle Distance* but also provide a new aspect to understand similarity through the lens of human cognition. From their fusion emerges the *Playstyle Similarity* measure.

To underscore our contributions, we propose three improvements for playstyle measurement. With these improvements, we alleviate the trade-off between using small or large discrete state spaces in discrete playstyle measures and provide a more explainable similarity with probability. Additionally, we achieve over 90% accuracy in zero-shot playstyle classification tasks with fewer than 512 observation-action pairs, which is less than half an episode in the examined games, including two racing games and seven Atari games. Furthermore, our experiments with *2048* and *Go* demonstrate the potential of discrete playstyle measures in puzzle and board games. Additionally, we introduce an algorithm to measure the diversity of decision-making, which showcases the applicability of our measures for tasks that are challenging to quantitatively evaluate without built-in features. This algorithm is based on the simple idea of comparing the similarity of a new trajectory to previous trajectories using a unified probabilistic similarity threshold. These explorations enhance our understanding of end-to-end game analysis and AI training, and also harmoniously merge playstyle measurement with human cognition.

## 2 Background and Related Works

In this section, we discuss playstyle in depth, provide a historical overview, and highlight the importance of discrete representation in the creation of general playstyle measures.

### 2.1 Playstyle and Measurement

Establishing a universally accepted playstyle measure is a formidable challenge, as perceptions of playstyle are influenced by myriad factors and often harbor subjective nuances. Consequently, any playstyle measure should specify its evaluative parameters transparently to ensure that its measurements are persuasive. Historically, tailored metrics, characterized by heuristic rules or specific in-game features, often presented the most precision for dedicated case studies. For example, the study by Lample & Chaplot (2017) used measures such as object counts, kills, and deaths in shooting games. However, due to their inherent manual nature, these measures are often domain-specific and limited to specific behaviors.

For video games, methods in player modeling can help us find interesting player behaviors and personality (Yannakakis et al., 2013; Costa Jr & McCrae, 1995). A key problem in these behavior analyses is how to define effective input features and the corresponding target outputs. Even if we can detect and taxonomize some behaviors, a real playstyle can be a complex combination of several behaviors. For example, the Bartle taxonomy in game character theory, where players can be separated into four types: achievers, explorers, socializers, and killers (Bartle, 1996), is a high-level concept of playstyles and the detailed behaviors of these taxonomies can be different in different genres of games. The true playstyle can be represented as the characteristic of playing behavior sets to fuse into a holistic intention of players (Lin et al., 2021); thus, processing all possible information from raw gameplay should be included in playstyle measurements.

To achieve such wider applicability, some researchers have resorted to supervised learning to identify styles (Brombacher et al., 2017). However, this method requires labeled training data and may encounter difficulties in detecting styles not present in the training set. Unsupervised clustering offers a different angle, emphasizing latent feature distances for classification (Drachen et al., 2009; 2013; Ferguson et al., 2020). But this approach may obscure the semantic meaning of the measures, particularly when image data is the primary source that is common in video games. A notable approach is the *Behavioral Stylometry* proposed by McIlroy-Young et al. (2021) using the idea of contrastive learning. This measure, designed for chess, encodes chess moves into a game vector, aggregates these vectors to represent a style, and then compares this representation against a reference set. Central to this method is the contrastive learning technique *Generalized End-to-End* (Wan et al., 2018) used to learn latent features to identify the most similar player in the given datasets.

For a more generalized measurement of playstyle, one could consider measuring the similarity of policies. Methods that extend to specify similarity or diversity by comparing the action distribution of two policies have also been explored (Agarwal et al., 2021; Lupu et al., 2021). Notably, these methods often require a parametrized policy for comparisons. This limitation is addressed by the *Playstyle Distance* measure (Lin et al., 2021). Instead of emphasizing latent features or parametrized policies, *Playstyle Distance* focuses on the action distributions of given samples. Raw observations, such as game screens, are discretized and then used for determining which action samples are comparable. Such a method resonates more with human instinct, echoing the case-by-case assessment we often deploy.

### 2.2 Framework of Playstyle Distance

To delve deeper into the generality and importance of the *Playstyle Distance* in playstyle measurement, we examine its foundation as follows. A pivotal component of its methodology is the use of the Vector Quantized-Variational AutoEncoder (VQ-VAE), which specializes in discrete representations by mapping continuous encodings to the nearest vectors in a predefined codebook (van den Oord et al., 2017). Building upon VQ-VAE, the Hierarchical State Discretization (HSD) in *Playstyle Distance* ensures a concise and hierarchical state space. This is essential for identifying overlapping discrete states while preserving the feature integrity of observation reconstruction and gameplay details.

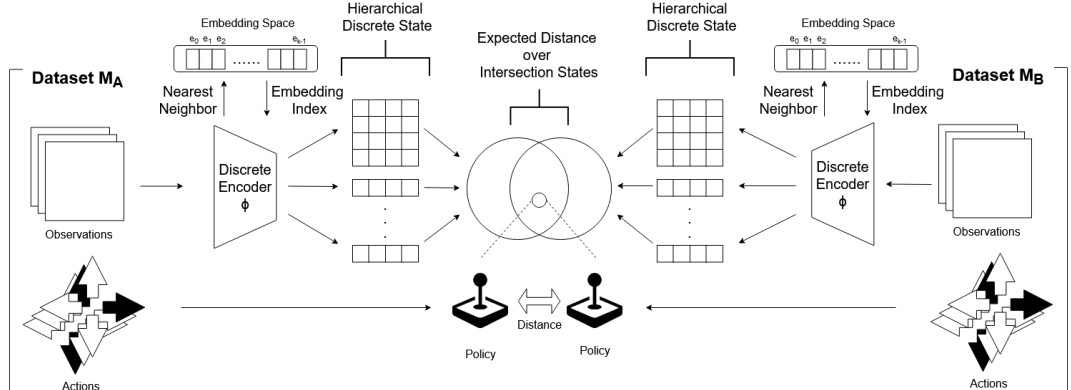

Figure 1: Illustration of the *Playstyle Distance* computation using a hierarchical discrete state encoder $\phi$. The Venn diagram highlights the intersection of discrete states for distance calculation.

Central to this framework is the discrete state encoder, denoted as $\phi$. Observations $o$ and their associated actions $a$ are mapped to datasets $M_i \sim Style_i$. The encoder $\phi$ translates these observations into a compact state representation $s$, formulated as:

$$S_\phi : \phi(o) \to s$$

In the initial *Playstyle Distance* approach, the hierarchical encoder $\phi$ has the capability to generate multiscale discrete states. However, the foundational literature employs only a singular state space for computations; the hierarchical structure is used to control the state space and maintain the quality of discrete representation learning. Using a state space that is too large or too small a state space may lead to unstable measurements.

From state $s$, action distributions are deduced using a sampling distribution:

$$\{a|(o,a) \in M, \phi(o) \to s\} \Rightarrow \pi_M(s)$$

Here, $\pi$ represents the policy, depicting action distributions for a given state. Subsequently, the distances between these distributions are determined using the metric $D(\pi_X, \pi_Y)$, where the 2-Wasserstein distance ($W_2$) serves as the standard (Vaserstein, 1969). Recognized for measuring the 'effort' to transform one distribution into another, the Wasserstein distance is apt for policy comparisons, analogous to quantifying the 'effort' to transition between playstyles.

The essence of this measure can be succinctly captured as:

$$d_\phi(M_A, M_B) = \frac{1}{2}d_\phi(M_A|M_B) + \frac{1}{2}d_\phi(M_B|M_A),$$
$$\text{where } d_\phi(M_X|M_Y) = \mathbb{E}_{o \sim M_Y, \phi(o) \in \phi(M_X) \cap \phi(M_Y)}[D(\pi_X(\phi(o)), \pi_Y(\phi(o)))] \tag{1}$$

To summarize, the *Playstyle Distance* framework presents a method to contrast playstyles, reducing the need for predefined heuristics and datasets, thereby increasing the applicability in various games. For a graphical representation of the framework, see Figure 1.

## 3 Discrete Playstyle Measures

In this section, we delve into a series of discrete playstyle measures derived from *Playstyle Distance*. We first discuss the limitations of *Playstyle Distance*. We then expand it into a multiscale approach by leveraging the hierarchical structure of states. Subsequently, we explore converting the action distribution distance into a perceptual similarity rooted in cognitive psychology, utilizing a perceptual kernel function. Thirdly, we broaden our scope from merely intersection states to the union of states, aiming for a more efficient estimation of all observed data. Concluding the section, we integrate these improvements into a comprehensive measure we term as *Playstyle Similarity*.

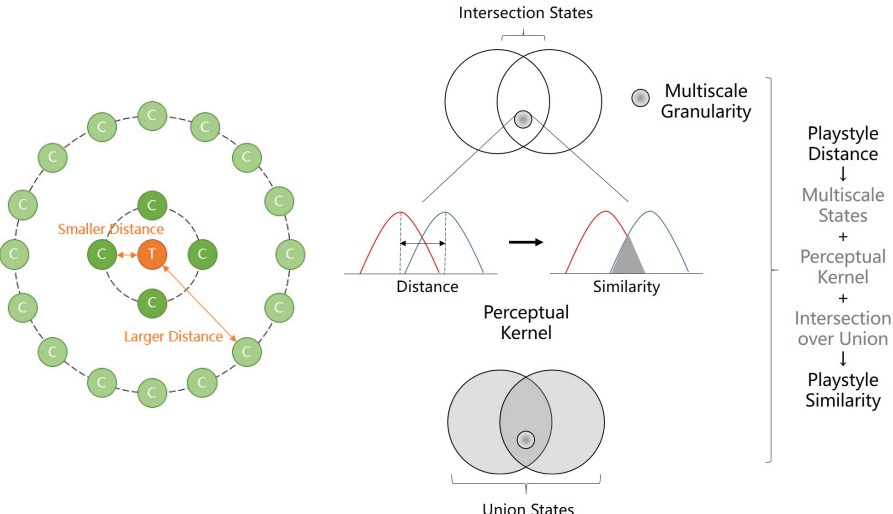

(a) Degree of Similarity      (b) From Playstyle Distance to Playstyle Similarity

Figure 2: (a) Degree of Similarity: This demonstrates how multiple candidate points C can share identical distance values from a target point T, emphasizing that as distance increases, the degree of similarity information diminishes. (b) From Playstyle Distance to Playstyle Similarity: This transformation begins by processing an observation sample into multiple discrete states of varying granularity. A perceptual kernel then transforms these distance values into probabilistic similarities, using the concept of overlapping regions for intuitive understanding. Lastly, the application of the intersection-over-union method refines the similarity based on the Jaccard index, enhancing measurement comprehensiveness across all observed data.

## 3.1 Effect of Multiscale States

*Playstyle Distance* hinges on discrete states for performing style measurements. Consequently, it resorts to a constrained state space sourced from the HSD model. A sample count threshold is applied to the intersection state to ensure the quality of action distribution, necessitating at least $t$ samples in both datasets under comparison; failing which, the state is excluded from the intersection. This filtering sometimes discards important information and the trade-off between using small or large state spaces also poses a dilemma for singular state space; thus, we propose using multiscale analysis to alleviate these problems.

Human cognition perceives similarity as a convergence of multiple attributes leading to a holistic understanding (Goldstone & Barsalou, 1998). Hence, we advocate for employing varied granularity of discrete states to augment measurement capabilities, analogous to human judgment that varies from a broad view to intricate details. The HSD model's design inherently possesses a large state space for observational reconstruction and the discernment of gameplay nuances (Lin et al., 2021). Though the state space may be large, intersections are not void if observations are sufficiently similar or come from identical gameplay. It is worth noting that Lin et al. (2021) were able to identify intersection states even with unprocessed screen pixels in Atari games. In a different scenario, when treating each state as the same, we can invariably find the single intersection state. In this context, distance simply gauges the action distribution over the entirety of the game, akin to traditional methods deploying post-game action statistics.

Broadly speaking, we can enhance the original state encoder function, $\phi$, evolving it into a state encoder mapping, $\Phi$, wherein $\Phi$ is an assemblage of mapping functions, $\phi \in \Phi$, $S_\Phi : \Phi(o) \rightarrow \{\phi(o) \in S_\phi | \phi \in \Phi\}$. Consequently, the projected state of dataset $M$ is defined as: $\Phi(M) \rightarrow \{\phi(o) \in S_\phi | o \in M, \phi \in \Phi\}$. We can then reinterpret Equation 1 as:

$$d_\Phi(M_A, M_B) = \frac{d_\Phi(M_A|M_B)}{2} + \frac{d_\Phi(M_B|M_A)}{2}, \text{where } d_\Phi(M_X|M_Y) = \frac{1}{|\Phi|} \sum_{\phi \in \Phi} d_\phi(M_X|M_Y) \qquad (2)$$

This reformulated measure demonstrates superior accuracy in playstyle classification tasks in our experiments, even negating the need for a sample threshold count $t$. This improvement is likely due to the integration of hierarchical discrete states of varying granularity, which dilutes the impact of outliers during distance computation and leverages more useful details. Furthermore, the adoption of a multiscale state space effectively mitigates the trade-off between a compact intersection space and the preservation of intricate information details. This balance becomes crucial in complex games or those that require a vast state space to encode trajectory data.

## 3.2  Perception of Similarity

One potential shortcoming of the *Playstyle Distance* stems from the nature of distance itself. While distance is a common measure for determining similarity, a larger distance value conveys primarily that two entities are different, without giving much insight into the degree of their similarity. For example, given a point in 2D space, the candidate points with the same distance to the given point form a circle. As the distance increases, the size of this candidate circle also increases, and the similarity information is diluted as illustrated in Figure 2a. This phenomenon has been observed in human decision-making as the *Magnitude Effect*, suggesting diminished sensitivity to larger numbers (Kahneman & Tversky, 1979). This aligns with the *Weber–Fechner Law* in psychophysics, where the relationship between stimulus and perception is logarithmic; as the magnitude of stimuli increases, sensitivity diminishes (Fechner, 1966). Drawing from the concept of similarity, we can infer that a smaller distance provides more definitive information about similarity. As distance grows, the distinction becomes vaguer. Therefore, we argue for a measure that reflects higher sensitivity to smaller distances, emulating human perceptual behaviors.

We propose a probability-based model for similarity. In this model, greater similarity (i.e., smaller distance) corresponds to a probability closer to 100%, while lesser similarity (larger distance) approaches 0%. This proposed probability function aligns with the logarithmic human perceptual sensitivity to differences. Specifically, we use the exponential kernel to describe the probability of similarity, with the mapping function given by $P(d) = \frac{1}{e^d}$, where $d$ is the distance value from the policy distance function $D(\pi_X, \pi_Y)$ with 2-Wasserstein distance. This perceptual relation is the only relation under our assumptions from human cognition and probability. We provide a proof using differential equations in Appendix A.1.

This exponential transformation can also be found in the radial basis function (Vert et al., 2004) and Bhattacharyya coefficient (Bhattacharyya, 1946). The Bhattacharyya coefficient $BC(P, Q)$ measures the similarity between two probability distributions $P$ and $Q$, and it is related to the overlapping region between these two distributions. It is defined as $BC(P, Q) = \int_{\mathcal{X}} \sqrt{P(x)Q(x)}dx$. The Bhattacharyya distance, derived from the coefficient, is $D_B(P, Q) = -ln(BC(P, Q))$, and the inversion is $BC(P, Q) = exp(-D_B(P, Q))$.

Thus, we define a new playstyle measure $PS_\Phi^\cap(M_A, M_B)$ with probability of similarity in Equation 3:

$$PS_\Phi^\cap(M_A, M_B) = \frac{\sum_{s \in \bigcup_{\phi \in \Phi} \phi(M_A) \cap \phi(M_B)} P(D_\Phi^M(\pi_{M_A}(s), \pi_{M_B}(s)))}{|\bigcup_{\phi \in \Phi} \phi(M_A) \cap \phi(M_B)|}, \text{where } D_\Phi^M(\pi_X, \pi_Y) = \frac{D(\pi_X, \pi_Y)}{\overline{D}_\Phi^M} \tag{3}$$

The measure has been simplified by adopting a uniform average distance instead of an expected value. This not only streamlines calculations but also underscores the significance of encoder granularity. In particular, an intricate encoder with a vast state space may be accorded greater weight, especially if the intricate encoder reveals more intersection states. To match our probabilistic framework (Appendix A.1) we rescale the distances with a constant, $\overline{D}_\Phi^M$, ensuring the expected distance converges to 1. The constant $\overline{D}_\Phi^M$ can be calculated by averaging all observed distance on each discrete state in comparisons. Collectively, our revamped measure provides a probabilistic lens to interpret similarity, firmly rooted in cognitive theory and tailored for human comprehension. There is more discussion about the role of the distance metric in Appendix A.2.1, including the implications of adopting the Bhattacharyya distance metric.

### 3.3 Beyond Intersection

Before presenting our final measure, it is pertinent to revisit the foundational concept of the *Playstyle Distance*: the intersection of states. Identifying comparable states before measuring policy similarities encounters challenges when the intersecting samples are limited. A smaller intersection proportion can result in unstable or insufficient samples for measuring playstyles. Such a small intersection could indicate two scenarios. First, distinct state-visiting distributions might signify different playstyles. In contrast, uncontrollable factors external to playstyles, such as environmental randomness or decisions from other players, may also play a role, indicating the necessity for more extensive sampling.

A prudent approach would assess the proportion of intersecting samples relative to the total observed samples. In the realm of collection comparison, the *Jaccard index* (Murphy, 1996), also known as *Intersection over Union*, emerges as a prevalent similarity measure. The *Jaccard index* can serve as an effective playstyle measure under specific conditions. It is particularly apt when game observations clearly delineate playstyle distinctions. For instance, in deterministic environments where states can be distinctly segmented by different actions, the *Jaccard index* appears to be a fitting measure. However, complications arise when certain states recur due to game rules or every state is visited. The task of distinguishing different playstyles based solely on observations becomes considerably challenging. This is evident in single-state games, such as K-arm bandits (Sutton & Barto, 2018), where measuring playstyles only from states becomes an impractical endeavor.

Despite potential challenges, our empirical findings suggest that the *Jaccard index* serves as a handy measure, when the state space is large and the randomness in the game is low. The incorporation of the *Jaccard index* into a playstyle measure with a multiscale state space is expressed in Equation 4:

$$J_\Phi(M_A, M_B) = \frac{|\bigcup_{\phi \in \Phi} \phi(M_A) \cap \phi(M_B)|}{|\bigcup_{\phi \in \Phi} \phi(M_A) \cup \phi(M_B)|} \tag{4}$$

### 3.4 Playstyle Similarity

Throughout our exploration, we have derived and discussed various discrete playstyle measures. Collating these insights, we introduce a comprehensive measure termed as the *Playstyle Similarity*. Defined as $PS_\Phi^\cup(M_A, M_B)$, it synthesizes our earlier discussions into a singular measure as illustrated below:

$$\begin{aligned} PS_\Phi^\cup(M_A, M_B) &= J_\Phi(M_A, M_B) \times PS_\Phi^\cap(M_A, M_B) \\ &= \frac{\sum_{s \in \bigcup_{\phi \in \Phi} \phi(M_A) \cap \phi(M_B)} P(D_\Phi^M(\pi_{M_A}(s), \pi_{M_B}(s)))}{|\bigcup_{\phi \in \Phi} \phi(M_A) \cup \phi(M_B)|} \end{aligned} \tag{5}$$

What makes this measure novel is its unique treatment of intersection states. While the *Jaccard index* assigns a uniform weight (of 1) to each intersecting state regardless of the similarity between the action distributions, our approach infuses a more nuanced probability-based weighting. The values range between 0 and 1, increasing proportionally with similarity. This modification alleviates the potential limitation of using the *Jaccard index* for playstyle measurements.

Furthermore, our approach ensures a consistent interpretation of zero values. For states not part of the intersection, where the distance between action distributions is maximal (approaching infinity), they can be understood as totally dissimilar. Playstyle Distance cannot directly incorporate with the Jaccard index due to its nature as a negative similarity measure. The overview of the transformation from Playstyle Distance to Playstyle Similarity is illustrated in Figure 2b.

## 4 Experiment Settings

In this section, we explain the specifics of our experiment setup, focusing on the datasets, sources of encoder models, and our playstyle classification methodology.

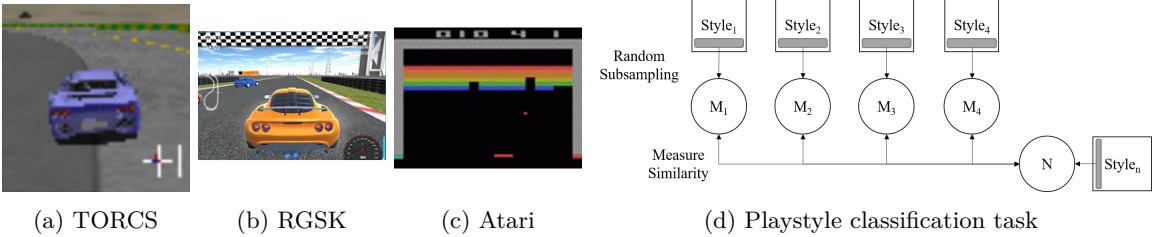

| (a) TORCS | (b) RGSK | (c) Atari | (d) Playstyle classification task |

Figure 3: Three game platforms and the illustration of zero-shot playstyle classification tasks.

### 4.1 Game Platforms, Datasets, and Model Source

Our study encompasses three distinct game platforms, as depicted in Figures 3a, 3b, and 3c:

1. **TORCS**: This racing game features stable, controlled rule-based AI players (Yoshida et al., 2017). The datasets derived from TORCS include a total of 25 playstyles based on 5 different target driving speeds and 5 different action noise levels. Each observation consists of a sequence of 4 consecutive RGB images with a size of $64 \times 64$. The action space is 2-dimensional and continuous.

2. **RGSK - Racing Game Starter Kit**: This racing game, available on the Unity Asset Store (Juliani et al., 2020), showcases human players. From RGSK, we have data from a total of 24 players, exhibiting individual playstyles. Human players are told to follow one specific style factor in 4 different style dimensions, including using nitro acceleration or not, driving on road surface or the grass surface, keeping the car in the inner or outer of the track, and passing a corner via drifting or slowing down with a brake. Each style dimension includes 6 players. Each observation from this game comprises 4 consecutive RGB images of size $72 \times 128$, with 27 discrete actions.

3. **Atari games with DRL agents**: The dataset spans 7 different Atari games (Bellemare et al., 2013) from this platform. Each game includes 20 AI models, all of which demonstrate varied playstyles. These AI models originate from the DRL framework, *Dopamine* (Castro et al., 2018). Each observation involves 4 consecutive grayscale images of resolution $84 \times 84$. The action space is discrete and varies depending on the game.

It is crucial to clarify that our research did not involve the training of new encoder models. Instead, we leveraged three pretrained encoder models and corresponding datasets for each game, provided by Lin et al. (2021). The associated resources are available in their official release.[1] The game details are listed in Table 1.

Table 1: Game details. The game 2048 and Go are used in Section 6 for extra evaluation.

| Game Platform | Agent Type | Style Count | Observation Size | Action Space |
|---|---|---|---|---|
| TORCS | Rule-based Agent | 25 | [4, 3, 64, 64] | Continuous 2-dimension |
| RGSK | Human Player | 24 | [4, 3, 72, 128] | Discrete 27 |
| Atari | DRL Agent | 20 | [4, 1, 84, 84] | Discrete 4 to 18 |
| 2048* | RL Agent | 10 | [4, 4] | Discrete 4 |
| Go* | Human Player | 200 | [18, 19, 19] | Discrete 362 |

### 4.2 Playstyle Classification and State Space Levels

Our playstyle classification adheres to the zero-shot methodology. As depicted in Figure 3d, we start with a query dataset $N$, sampled from a playstyle $Style_n$. We then compare this to multiple reference datasets $M$, each sampled from different playstyles $Style$. We perform 100 rounds of random subsampling for each playstyle; our primary performance metric for this task is the accuracy of playstyle classification. If dataset $N$ exhibits the highest similarity to a reference dataset $M_i$, it suggests that $Style_n = Style_i$. The reported accuracy represents an average, derived from results obtained using the three discrete encoder models.

---

[1] https://paperswithcode.com/paper/an-unsupervised-video-game-playstyle-metric

Regarding the discrete state space levels, three tiers have been considered:

1. Space size 1, a basic mapping with state space 1, which maps all observations identically.

2. Space size $2^{20}$, as suggested by *Playstyle Distance*.

3. Space size $256^{64\sim144}$ or $256^{\mathrm{res}}$, a level trained by HSD for the base hierarchy, depending on the resolution **res** of convolution features from the game screens.

## 5 Results in Video Games

In this section, we assess the efficacy of our methods on two racing games and seven Atari games. Initially, we demonstrate how a multiscale state space can aid in the selection of proper state spaces and potentially enhance the accuracy of *Playstyle Distance*. Subsequently, we compare several baselines, illustrating that probabilistic values for measuring similarity offer a viable alternative to distance values. Next, we incorporate all observed data to evaluate measures across all platforms. Furthermore, we discuss the continuous playstyle spectrum, demonstrating the consistency of measure values under slightly different behaviors. Lastly, we compare potential unsupervised measures beyond discrete playstyle measures, using observation latent features common in generative styles. Overall, our intention in this section is to determine whether new measures can enhance playstyle measurement in video games.

### 5.1 Multiscale State Space Efficacy

We evaluate the proposed multiscale state space and compare it with the singular state space used by *Playstyle Distance* in Table 2. We record the mean accuracy and corresponding standard deviation over 100 rounds of random subsampling for each discrete encoder. Each dataset sampled from the given playstyles comprises 1024 observation-action pairs. For example, in TORCS, there are 25 playstyles; we poll on 25 query sets, and each dataset has 1024 observation-action pairs sampled from its playstyle. We compute this query set with another 25 candidate sets, each having 1024 pairs sampled from their playstyles. When this round of polling is finished, we count whether the most similar candidate set has the same playstyle as the query set. This polling process will run over 100 times for sampling different subsets of actual playstyle datasets. Additionally, we examine the intersecting states' sample threshold count $t$, where an intersecting state requires at least $t$ samples in both compared datasets for a stable action distribution estimation.

In our comparisons, conventional methods, such as supervised learning and contrastive learning, fall short for this classification due to the lack of playstyle labels or groups in the training datasets, resulting in a random model. Thus, we focus our comparisons on *Playstyle Distance*, as detailed in Equation 2. For the multiscale version, which incorporates three discrete state spaces—$\{1, 2^{20}, 256^{\mathrm{res}}\}$—we simplify our terminology by using the label "mix." The results, as shown in Table 2, indicate that using a multiscale discrete state space not only simplifies the selection of a proper state space by using all spaces and potentially yields superior results but also obviates the need for a sample threshold count $t$ for intersecting states in some games requiring a stable action distribution, such as TORCS, Asterix, and Breakout.

### 5.2 Probability vs. Distance

In addition to introducing the multiscale discrete state space, another key contribution of our work is the proposal to use probabilistic similarity from a perceptual perspective rather than employing negative distance as a measure of similarity. To elucidate the benefits of this modification, we examine the relationship between accuracy and dataset size of the sampled observation-action pairs. These pairs are evaluated under a single discrete state space $\{2^{20}\}$, without employing a sample count threshold, to provide a clear assessment of the transformation from distance to similarity. Further comparisons with different discrete state spaces can be found in Appendix A.2.

We evaluate several measures in this comparison:

- *Playstyle Distance*: $-d_\Phi$

Table 2: Playstyle accuracy (%) $\pm$ standard deviation (%) when employing various discrete state spaces in the multiscale version of *Playstyle Distance*, with a sample threshold count $t$ for intersecting states. Conventional methods, such as supervised learning and contrastive learning, are not suited for cases lacking playstyle labels in the training datasets, resulting in a random model with $\frac{1}{\text{Style Count}}$ accuracy. In contrast, discrete playstyle measures do not suffer from this limitation, as they do not require playstyle labels for training the discrete encoders. The accuracy and standard deviation values are averaged from 3 different encoder models. The results show that using a multiscale state space (mix) offers more convenient state space selection and potentially improves the accuracy. Additionally, the sample threshold count $t$ suggested by Lin et al. (2021) can be ignored, which was required in TORCS with a $2^{20}$ space and in Atari games like Asterix and Breakout with a $256^{\text{res}}$ space for stable measurement. The detailed statistics for each discrete encoder are listed in Section A.3.1.

| | 1 | $2^{20}$ $t=2$ | $2^{20}$ $t=1$ | $256^{\text{res}}$ $t=2$ | $256^{\text{res}}$ $t=1$ | **mix** $t=2$ | **mix** $t=1$ |
|---|---|---|---|---|---|---|---|
| TORCS | $35.1 \pm 9.1$ | $73.3 \pm 8.2$ | $66.5 \pm 7.9$ | $4.3 \pm 3.1$ | $60.9 \pm 9.4$ | $\mathbf{77.3} \pm 7.4$ | $\mathbf{77.5} \pm 7.9$ |
| RGSK | $81.0 \pm 7.2$ | $79.2 \pm 7.9$ | $\mathbf{93.7} \pm 4.7$ | $5.7 \pm 2.5$ | $25.6 \pm 7.2$ | $78.8 \pm 7.5$ | $\mathbf{93.5} \pm 4.3$ |
| Asterix | $25.2 \pm 9.0$ | $\mathbf{99.9} \pm 0.5$ | $\mathbf{100} \pm 0$ | $49.6 \pm 7.7$ | $32.7 \pm 8.0$ | $\mathbf{100} \pm 0$ | $\mathbf{100} \pm 0$ |
| Breakout | $32.7 \pm 9.2$ | $\mathbf{99.4} \pm 1.6$ | $\mathbf{99.9} \pm 0.6$ | $65.9 \pm 8.5$ | $29.9 \pm 9.4$ | $\mathbf{99.8} \pm 1.1$ | $\mathbf{99.9} \pm 0.2$ |
| MsPac. | $\mathbf{100} \pm 0$ | $99.9 \pm 0.5$ | $\mathbf{100} \pm 0$ | $92.8 \pm 4.0$ | $\mathbf{100} \pm 0$ | $\mathbf{100} \pm 0$ | $\mathbf{100} \pm 0$ |
| Pong | $49.9 \pm 9.7$ | $92.1 \pm 2.7$ | $92.3 \pm 2.6$ | $50.7 \pm 9.5$ | $52.2 \pm 9.9$ | $\mathbf{93.1} \pm 3.2$ | $92.4 \pm 2.6$ |
| Qbert | $\mathbf{99.9} \pm 0.5$ | $\mathbf{100} \pm 0$ | $\mathbf{100} \pm 0$ | $90.1 \pm 5.3$ | $91.6 \pm 4.6$ | $\mathbf{99.9} \pm 0.5$ | $\mathbf{100} \pm 0$ |
| Seaquest | $82.0 \pm 7.6$ | $\mathbf{99.7} \pm 1.2$ | $\mathbf{99.9} \pm 0.6$ | $17.1 \pm 5.2$ | $16.7 \pm 4.9$ | $\mathbf{99.9} \pm 0.3$ | $\mathbf{99.9} \pm 0.2$ |
| SpaceIn. | $73.1 \pm 8.5$ | $98.7 \pm 2.3$ | $\mathbf{99.7} \pm 1.2$ | $50.4 \pm 5.7$ | $49.6 \pm 8.4$ | $\mathbf{99.9} \pm 0.5$ | $\mathbf{99.9} \pm 0.6$ |

- *Playstyle Intersection Similarity*: $PS_\Phi^\cap$

- *Playstyle Inter BD Similarity*: $PS_\Phi^{\cap BD}$, a variant of $PS_\Phi^\cap$ that employs the Bhattacharyya distance in place of the 2-Wasserstein distance

- *Playstyle Inter BC Similarity*: $PS_\Phi^{\cap BC}$, the Bhattacharyya coefficient version, which omits the scaling coefficient before the perceptual kernel $\frac{1}{e^d}$

- *Random*: A uniform random baseline that is a common result from supervised learning or contrastive learning if there is no style label or group (like self and others) information in the training data.

Results presented in Figure 4 suggest that probabilistic similarity can be a good alternative to distance-based similarity, offering improved explainability in terms of measure values. Among the methods evaluated, the 2-Wasserstein distance with a perceptual kernel and the Bhattacharyya coefficient emerge as superior candidates. The intention behind using probabilistic similarity is that it provides a consistent measure of similarity across different games (via likelihood). For distance similarity, understanding the property's and distribution of distance is essential to interpret the measure value. Besides the explainability of similarity values, the transformed similarity value can be incorporated with the Jaccard index as described in Section 3.3. The evidence shows that results with probabilistic similarity are not worse than distance similarity and are slightly better on TORCS, which includes slightly different playstyles. The upcoming experiments in Sections 5.3 and 5.4 also support the idea of probabilistic similarity under slightly changed playstyles, such as rule-based TORCS agents and Atari game agents trained with the same algorithm.

## 5.3 Full Data Evaluation

Based on the previous evaluations, we further perform a comprehensive evaluation of various playstyle measures, including leveraging full data with union operations. The evaluation method mirrors the one presented in Section 5.2, but expands the scope beyond racing games and adopts a multiscale state space.

Detailed results for each Atari game have been moved to Appendix A.3.2. Instead, leveraging the consistent observation and action space shared across Atari games, we propose a unified Atari console evaluation. This

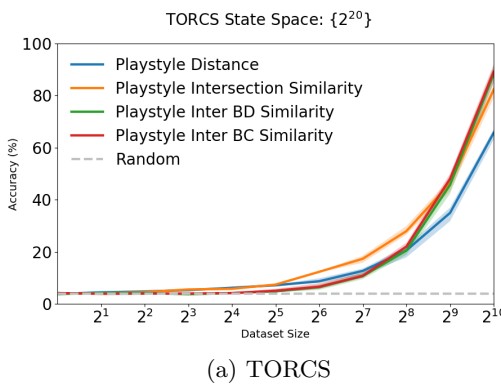
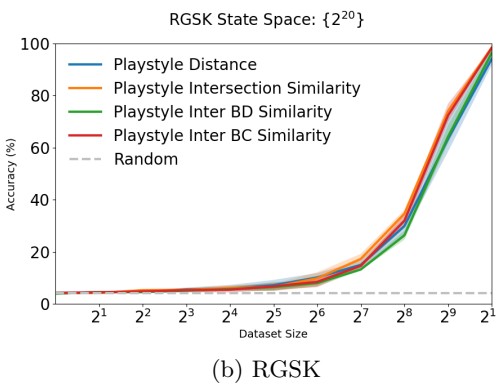

(a) TORCS

(b) RGSK

Figure 4: Comparison of Efficacy: Probabilistic vs. Distance Approaches. The plot illustrates the relationship between accuracy (Y-axis) and size of the sampled observation-action pairs (X-axis). The shaded area indicates the range between min and max accuracy among three encoder models.

evaluation views each DRL agent's gameplay on individual games as distinct playstyles, yielding $7 \times 20$ unique playstyles. As for the discrete state space, we include a single shared state mapping in addition to two hierarchical discrete encoders from the seven games; thus, there are totally $1 + 7 \times 2$ discrete state encoders. For actions, rather than aligning their semantics across games, we expand the action set to the largest count in Atari games, which is 18. This is based on the assumption that variations in game content can be interpreted as different states.

The platforms span TORCS, RGSK, and Atari games. We compare the following measures:

- *Playstyle Distance*: $-d_\Phi$

- *Playstyle Intersection Similarity*: $PS_\Phi^\cap$

- *Playstyle Inter BC Similarity*: $PS_\Phi^{\cap BC}$

- *Playstyle Jaccard Index*: $J_\Phi$

- *Playstyle Similarity*: $PS_\Phi^\cup$

- *Playstyle BC Similarity*: $PS_\Phi^{\cup BC}$, the union version of *Playstyle Inter BC Similarity*

- *Random*: A uniform random baseline.

Results displayed in Figure 5 show that the *Playstyle Similarity* outperforms its counterparts. Moreover, the *Jaccard index* has proven to be useful in practice. Our combined Atari console evaluation further underscores the robustness and adaptability of our measures.

Conclusively, our proposed *Playstyle Similarity* measure excels across these video game platforms. It is particularly impressive that it can identify playstyles with over 90% accuracy with just 512 observation-action pairs — less than half an episode across all tested games. This suggests the possibility of accurate playstyle prediction even before a game concludes, paving the way for real-time analysis.

## 5.4 Continuous Playstyle Spectrum in TORCS

This experiment investigates the response of similarity measure values to variations in a continuous playstyle spectrum within the TORCS environment. It particularly focuses on whether these measures can accurately rank playstyles, ensuring precise predictions for the closest playstyle (Top-1 similarity) and maintaining a correctly ordered sequence of playstyles based on similarity measures.

Utilizing the TORCS dataset, which includes five levels of target speeds (60, 65, 70, 75, 80) and five levels of action noise, provides a broad spectrum for examining continuous playstyle changes. To illustrate, consider

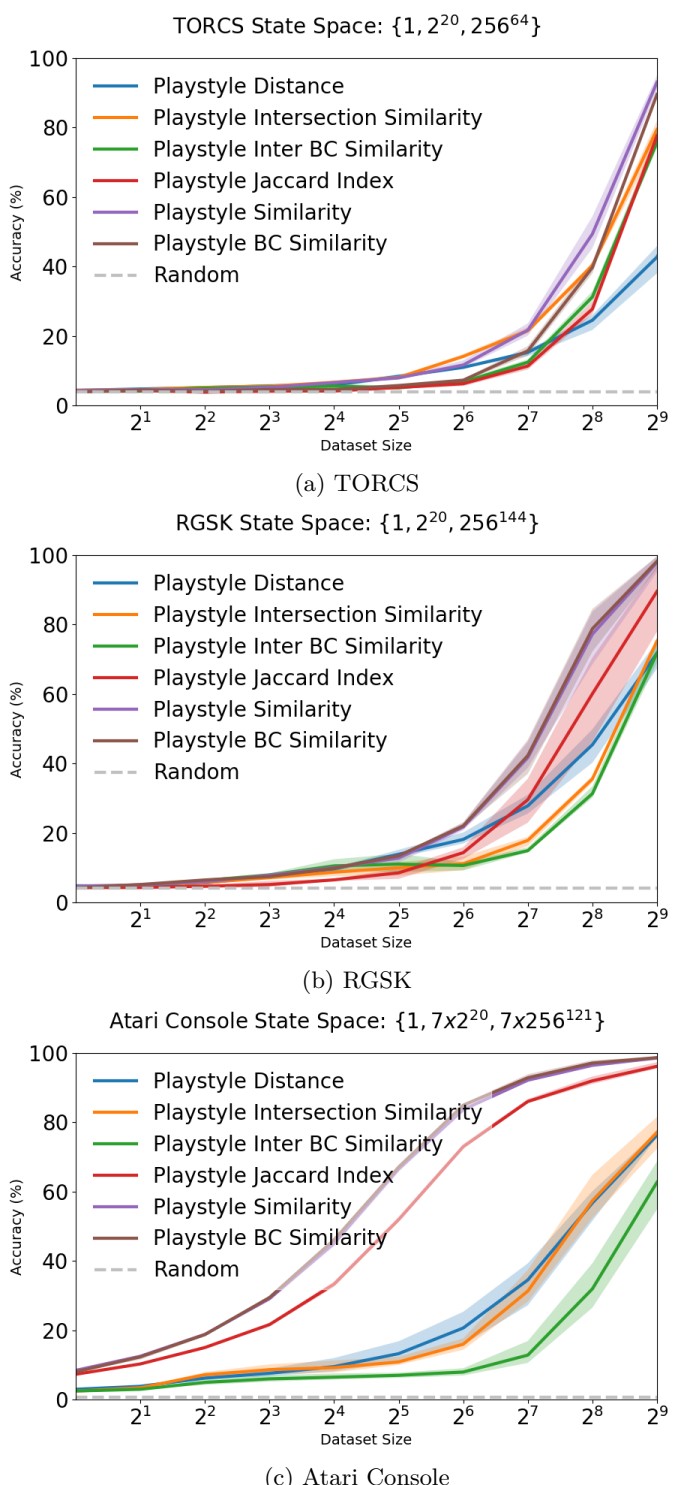

Figure 5: Playstyle Measure Evaluation in TORCS, RGSK, and Atari Console. The plots showcase the efficacy of different measures in the context of the "Full Data Evaluation" subsection. The shaded area indicates the range between min and max accuracy among three encoder models.

Table 3: Corner Case uses Playstyle Similarity (mix): consistent count = 9. The query playstyle is marked as orange, and those columns with consistency in measure values are marked in blue. For rows with consistency, marked as red, and for those values with consistency in both columns and rows, marked violet.

| Speed60N0 | Speed60 (C) | Speed65 (C) | Speed70 (C) | Speed75 (C) | Speed80 (C) |
|---|---|---|---|---|---|
| N0 (C) | 0.0753 | 0.0650 | 0.0608 | 0.0555 | 0.0472 |
| N1 (C) | 0.0590 | 0.0579 | 0.0549 | 0.0513 | 0.0446 |
| N2 | 0.0519 | 0.0533 | 0.0525 | 0.0458 | 0.0440 |
| N3 (C) | 0.0482 | 0.0473 | 0.0448 | 0.0435 | 0.0382 |
| N4 (C) | 0.0421 | 0.0413 | 0.0402 | 0.0349 | 0.0337 |

five playstyles labeled A through E, with A' being a variant closely aligned with A. We anticipate that the similarity between A' and A would be the highest, progressively decreasing towards E. This expectation sets the stage for our consistency test, wherein a similarity measure $M$ should validate the order $M(A', A) > M(A', B) > M(A', C) > M(A', D) > M(A', E)$ (Corner Case).

We assess the following measures:

1. Playstyle Distance with a $2^{20}$ state space (baseline)

2. Playstyle Distance using a mixed state space

3. Playstyle Intersection Similarity with a mixed state space

4. Playstyle Jaccard Index with a mixed state space

5. Playstyle Similarity with a mixed state space

With 100 rounds of random subsampling (each dataset consisting of 512 observation-action pairs) and using the first Hierarchical State Discretization (HSD) model, we examine the consistency of similarity values as playstyle shifts. A decrease in similarity consistent with playstyle changes is marked with a (C), indicating measure reliability.

For instance, choosing Speed60N0 as a target playstyle, we observe how similarity measures adjust across a row or column in response to increasing speed or action noise levels. A practical demonstration reveals how the measure values consistently decrease across increasing speed levels and noise intensities, illustrating the measure's ability to capture playstyle divergence accurately. Table 3 presents one such example with Playstyle Similarity (mix). For all measure values, please refer to our Appendix A.4.

Similar analyses extend to a Center Case scenario with Speed70N2 as the target playstyle, emphasizing the necessity for similarity measures to respect two key sequences for consistency:

1. $M(C', C) > M(C', B) > M(C', A)$

2. $M(C', C) > M(C', D) > M(C', E)$

These sequences confirm the measure's capacity to accurately reflect the gradual divergence of playstyles from a central reference point. The examples of Center Case can be found in Appendix A.4.

Table 4 summarizes the consistency count across all evaluated measures, showcasing the reliability and effectiveness of each similarity measure in maintaining a continuous playstyle spectrum.

Table 4: Consistency count of two continuous playstyle spectrum cases.

|  | Corner Case | Center Case |
|---|---|---|
| Playstyle Distance ($2^{20}$) | 3 | 2 |
| Playstyle Distance (mixed) | 4 | 2 |
| Playstyle Intersection Similarity (mixed) | 8 | 2 |
| Playstyle Jaccard Index (mixed) | 5 | 1 |
| Playstyle Similarity (mixed) | **9** | **3** |

## 5.5 Comparison of Potential Unsupervised Similarity Measures

With previous experiments, we have verified the effectiveness of these discrete playstyle measures. Although there are few methods for unsupervised playstyle measurement before Playstyle Distance, there are some measures popular in generative styles with latent feature similarity. For example, the two most popular measures for research in generative adversarial networks (GAN) with styles (Karras et al., 2019) are the Inception Score and Fréchet Inception Distance (FID) (Salimans et al., 2016; Heusel et al., 2017). These are based on using an image classification model, Inception (Szegedy et al., 2015), for scoring the generated images. The Inception Score is based on the prediction distribution of the classifier to detect real or generated images, which diverges from the target of playstyle measurement as all observations are real images from game environments. FID measures the 2-Wasserstein distance on the latent features of images and could be incorporated into playstyle measurement if we assume a playstyle features a unique observation distribution. However, calculating the W2 distance for high-dimensional continuous latent distributions is computationally intensive and does not yield good results in the game TORCS (Lin et al., 2021). The major complexity arises from the calculation related to the covariance matrix. A time-feasible alternative is using the similarity of the mean vector of these latent features. We can first average those latent features of observations into a mean vector to represent the playstyle and compare the similarity to candidate vectors obtained from the same process, which is analogous to the method in Behavior Stylometric (McIlroy-Young et al., 2021). These latent features used in the following experiments are the continuous latent features before vector quantization to the $2^{20}$ state space in the HSD models with 500 dimensions. The discrete version of these latents with information loss has already shown effectiveness in playstyle measurements, so we compare using continuous latents with two popular similarity measures for playstyle measurements: Euclidean Distance (L2 distance) and Cosine Similarity. Cosine Similarity is especially common in latent similarity applications (Chung et al., 2022; McIlroy-Young et al., 2021).

We assess the following measures:

1. Playstyle Distance with a $2^{20}$ state space

2. Playstyle Jaccard Index with a mixed state space

3. Playstyle Similarity with a mixed state space

4. Playstyle BC Similarity with a mixed state space

5. Random, the uniform random baseline

6. Euclidean Distance, using L2 distance as the similarity measure for observation latent features

7. Cosine Similarity, using Cosine Similarity as the similarity measure for observation latent features

We first check the results for TORCS and RGSK in Figure 6. It is clear that Euclidean Distance and Cosine Similarity do not provide good predictions, which is not surprising since the observations in the TORCS practice track are monotonous, and the playstyles are not directly correlated to observations. In contrast, playstyles in RGSK, such as nitro acceleration or preferred road surface, have a high correlation to visual features and even perform better than Playstyle Similarity under a few samples. When we further examine

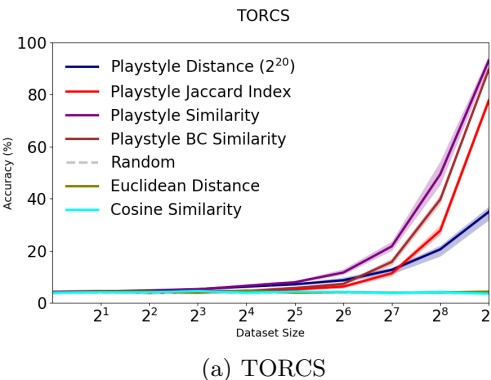
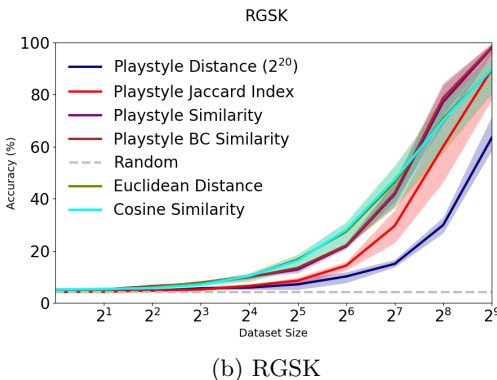

(a) TORCS          (b) RGSK

Figure 6: Accuracy comparison of potential unsupervised similarity measures on (a) TORCS and (b) RGSK. The shaded area indicates the range between the min and max accuracy among three encoder models. Observation latent-based methods like Euclidean Distance and Cosine Similarity perform nearly at random levels in TORCS and perform much better in RGSK due to different playstyles and game properties. The playstyles in RGSK have many visual features that reflect styles, such as the blue fire from the nitro acceleration system and the driving surface or position on the track.

these measures on Atari games in Figure 7, Euclidean Distance and Cosine Similarity usually share similar performance, and the results depend on the games. In all cases, Playstyle Similarity and its BC variant are superior to these potential candidates with datasets over 256 in size, regardless of the playstyle and games. The measure values based on action distributions are more explainable than observation latent features. Furthermore, observations are sometimes not controllable by the players performing playstyles but are controlled by game mechanisms, other players, or even sample bias. Discrete playstyle measures can defend against these influences since observations are used for comparison conditions rather than direct measurement computations. If these influencing factors make observations different, discrete playstyle measures act conservatively, requiring more samples or deeming them incomparable, while latent similarity approaches will directly take them into calculation without considering the influencing factors. Overall, these results show that our Playstyle Similarity and its BC variant are more general and effective in unsupervised playstyle measurement compared to other existing or potential measures.

# 6 Playstyle Measures Under High Uncertainty Games

To further assess the efficacy of our playstyle measures in games characterized by high uncertainty, we conducted experiments with the puzzle game 2048 and the board game Go (Figure 8).

## 6.1 2048: High Randomness Puzzle Game

Known for its single-player format and high degree of randomness, 2048 presents challenges in generating identical trajectories. We trained a reinforcement learning agent (Szubert & Jaskowski, 2014) over 10 million episodes, creating 10 distinct players by saving the model every 1 million episodes. For each player, we collected 1000 episodes, using the first 500 as the reference dataset and the remaining 500 as separate query datasets. This resulted in a total of 5000 query datasets for the experiment. The setup aimed to test the accuracy of playstyle measures in scenarios marked by high randomness and similar behaviors across small query datasets, simulating conditions that could challenge discrete playstyle measures. In this case, we directly utilized the $4 \times 4$ full board as the discrete states, which is equivalent to raw game screens in 2048 in terms of state information. The experimental settings used for this analysis are listed in Appendix A.6.

The results in Table 5 demonstrate that measures incorporating the Jaccard index negatively impact accuracy. In games with high randomness and large observation space, it is crucial to evaluate whether discrete states can identify key style factors while maintaining a manageable state space to find comparable samples.

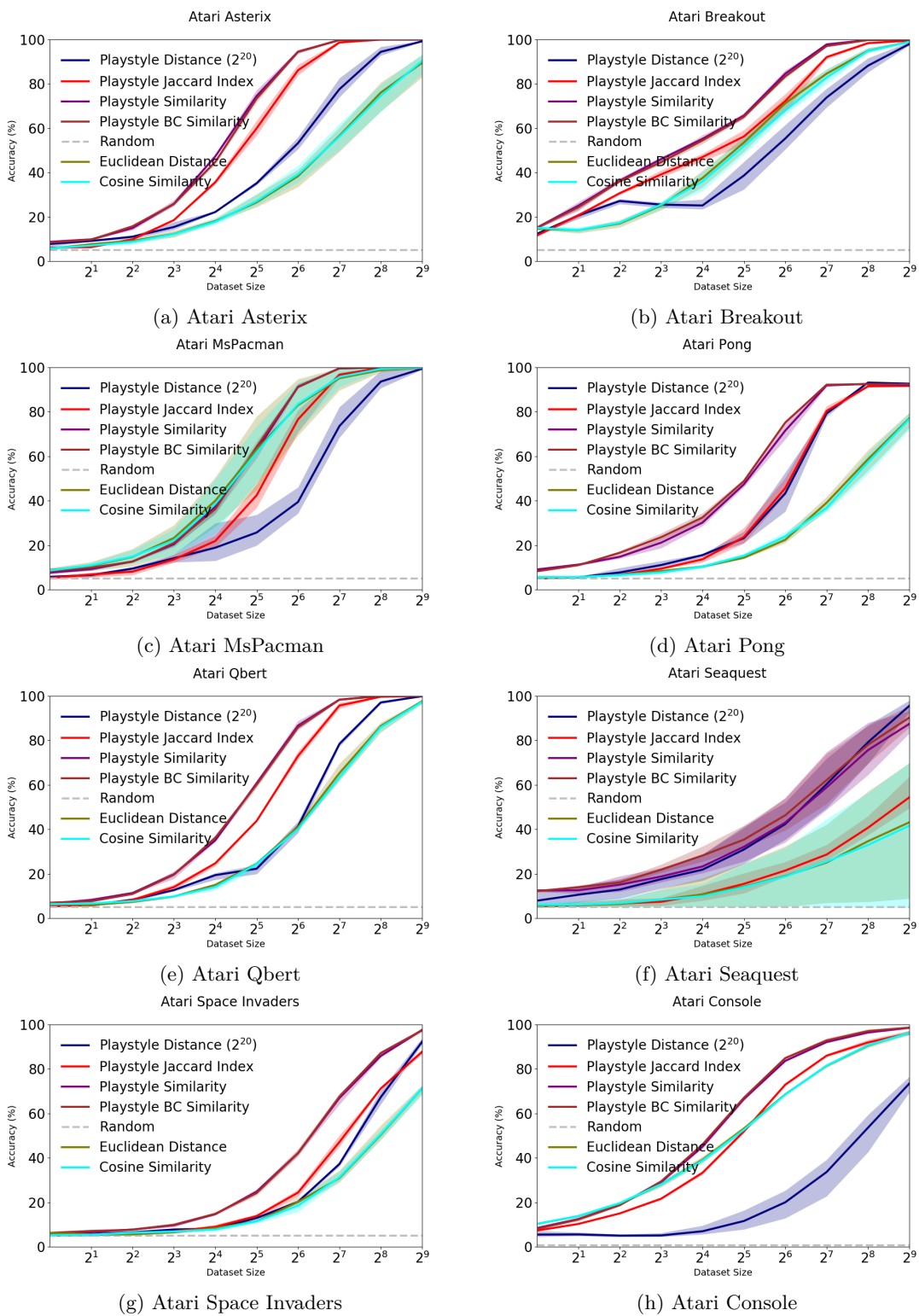

Figure 7: Accuracy comparison of potential unsupervised similarity measures on Atari games. The shaded area indicates the range between the min and max accuracy among three encoder models. Euclidean Distance and Cosine Similarity show nearly the same performance across different games, sometimes better and sometimes worse than Playstyle Distance.

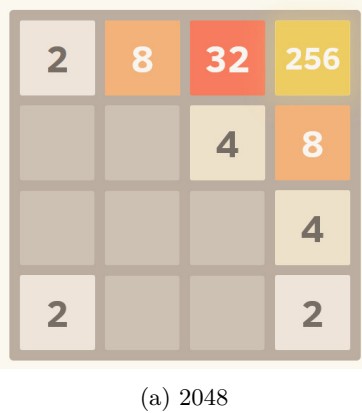

(a) 2048

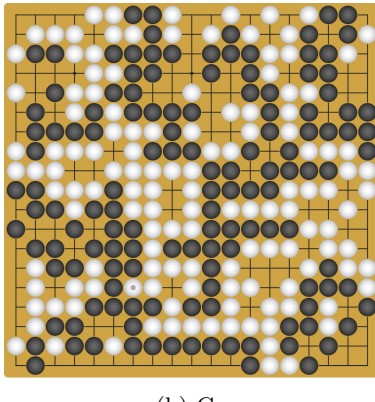

(b) Go

Figure 8: Game screens of 2048 and Go.

Table 5: Accuracy of game 2048 model identification.

|  | Accuracy |
| --- | --- |
| Playstyle Distance | **98.90%** |
| Playstyle Intersection Similarity | **98.52%** |
| Playstyle Inter BC Similarity | **98.90%** |
| Playstyle Jaccard Index | 49.22% |
| Playstyle Similarity | 71.26% |
| Playstyle BC Similarity | 71.26% |

## 6.2 Go: Two-Player Board Game

In addition to the inherent randomness of game environments, the inclusion of other players introduces further uncertainty in measuring playstyles. Consequently, we undertook a human playstyle identification task using the two-player board game, Go, which is known for its high game tree and state space complexity (Van Den Herik et al., 2002). This complexity can challenge discrete playstyle measures.

We implemented a variant of the HSD encoder (Lin et al., 2021) to obtain a discrete state encoder for this task. The major difference from the original HSD is that the reconstruction objective is replaced by predicting the win value, with prediction policy and value being the standard objectives in Alpha Zero series algorithms (Silver et al., 2018). The details about this discrete encoder are described in Section A.7. The Go dataset used in this study was sourced from Fox Go (Fox Go, 2024a;b) and provided by the team of the MiniZero framework (Wu et al., 2024). It includes 45,000 games from players with 9 Dan Go skill for training the encoder, with corresponding actions but without player information or style labels. Another dataset includes 200 human players with Go skill ranging from 1 Dan to 9 Dan, each contributing 100 games to the query datasets and 100 games to the candidate datasets. The discrete state space used for training the encoder includes $\{4^8, 16^8, 256^{361}\}$.

Results in Table 6 demonstrate the accuracy comparisons of the player identification task with different discrete playstyle measures. There are two playstyle scenarios in our evaluation. For the first 10 moves case, it is common and straightforward in board games that a preferred opening is a kind of playstyle and using a small state space can achieve 97.0% accuracy ($\{4^8\}$). However, when we do not specifically focus on the playstyles in the opening, we may want to use all game moves to capture any possible playstyles, and such a small state space negatively impacts the accuracy. Some boards cannot be compared in this kind of playstyle, but the small state space cannot provide information to separate these cases. Instead, large state spaces like $\{16^8\}$ or $\{256^{361}\}$ can handle this kind of problem. If we do not know which state space can give the best result, our multiscale state space is a good choice that leverages all state spaces. Even if it may be influenced by a very bad state space like $\{4^8\}$ in the full games case, the damage is limited. Also,

Table 6: Accuracy of Go 200 player identification with M games as the query set and also M games as the candidate set. The full comparison with different numbers of query and candidate sets is listed in Section A.7. The discrete encoder is trained from a variant of HSD (Lin et al., 2021), and the available state spaces in this encoder are $\{4^8, 16^8, 256^{361}\}$. The measures with "mix" notation imply using all three available state spaces simultaneously with our multiscale modification.

| Only First 10 Moves | M=1 | M=5 | M=10 | M=25 | M=50 | M=75 | M=100 |
|---|---|---|---|---|---|---|---|
| Playstyle Distance ($4^8$) | 1.5% | 11.5% | 33.5% | 71.0% | 87.0% | 93.0% | **97.0%** |
| Playstyle Distance ($16^8$) | 2.0% | 10.0% | 41.5% | 74.0% | 85.0% | 92.5% | 95.5% |
| Playstyle Distance ($256^{361}$) | 2.0% | 8.0% | 38.0% | 75.0% | 87.0% | 92.0% | 94.5% |
| Playstyle Distance (mix) | 1.5% | 14.0% | **46.5%** | **75.5%** | 87.5% | 94.5% | 96.5% |
| Playstyle Inter. Similarity (mix) | 2.5% | 15.0% | 42.0% | 68.5% | 84.0% | 94.5% | 94.5% |
| Playstyle Inter BC Similarity (mix) | 2.5% | 14.0% | 42.5% | 72.0% | **89.0%** | **96.5%** | 95.5% |
| Playstyle Jaccard Index (mix) | **3.0%** | 9.5% | 21.5% | 49.0% | 78.0% | 88.0% | 95.0% |
| Playstyle Similarity (mix) | **3.0%** | 21.5% | 40.0% | 65.0% | 87.5% | 94.0% | **97.0%** |
| Playstyle BC Similarity (mix) | **3.0%** | **24.0%** | 45.5% | 70.5% | 88.0% | **96.5%** | **97.0%** |
| **Full Game Moves** | M=1 | M=5 | M=10 | M=25 | M=50 | M=75 | M=100 |
| Playstyle Distance ($4^8$) | 2.5% | 5.5% | 11.5% | 20.0% | 31.0% | 39.5% | 50.0% |
| Playstyle Distance ($16^8$) | 2.0% | 10.0% | 44.5% | 74.0% | 86.0% | 93.5% | 96.5% |
| Playstyle Distance ($256^{361}$) | 2.0% | 8.0% | 39.5% | **75.5%** | 88.5% | 93.5% | 96.5% |
| Playstyle Distance (mix) | 2.0% | 17.0% | 33.0% | 56.5% | 76.0% | 83.0% | 90.0% |
| Playstyle Inter. Similarity (mix) | 3.0% | **19.0%** | 38.0% | 66.0% | 86.0% | 95.0% | 95.0% |
| Playstyle Inter BC Similarity (mix) | 2.5% | 16.5% | 37.5% | 66.0% | 90.0% | 96.0% | 97.0% |
| Playstyle Jaccard Index (mix) | 3.0% | 5.0% | 8.5% | 22.5% | 51.0% | 66.0% | 80.5% |
| Playstyle Similarity (mix) | **4.5%** | 15.5% | 29.5% | 56.5% | 81.5% | 90.5% | 94.0% |
| Playstyle BC Similarity (mix) | 4.0% | 22.0% | **45.5%** | 73.0% | **92.0%** | **97.5%** | **97.5%** |

discrete playstyles incorporating the Jaccard index (Playstyle Similarity and its variants) can further cover this weakness.

Additionally, the Bhattacharyya coefficient is tend to be more effective than using perceptual kernel with scaled W2 distance when there are sufficient samples in Go, especially in the full game moves case. It is possibly due to its distribution property that rapidly decreases similarity with few overlapping outcomes, and our Go result is an example. For slightly different playstyles like TORCS with continuous actions, using the scaled W2 metric can provide better accuracy. Our findings suggest that discrete playstyle measures can achieve significant accuracy with sufficient samples directly from the measure definition even in a complex multi-agent board game, without any predefined style labels.

## 7 Diversity Measurement in DRL

With these discrete playstyle measures, we can design an algorithm to quantify the diversity among DRL models, which is challenging to measure and quantify formally in environments with high-dimensional observations. Algorithm 1 provides a simple method to quantify diversity in decision-making by measuring the similarity between a new trajectory and observed trajectories. If a new trajectory is not similar enough to any observed trajectories, we count it as a different one.

We conducted experiments to evaluate this algorithm using the Atari DRL agent dataset. Each DRL algorithm (DQN, C51, Rainbow, IQN, (Mnih et al., 2015; Bellemare et al., 2017; Hessel et al., 2018; Dabney et al., 2018)) includes 5 models, each contributing 5 trajectories, resulting in 25 trajectories per algorithm. Using Algorithm 1, we assessed the diversity of trajectories produced by each algorithm. Results in Table 7, averaged across three discrete encoder models with a similarity threshold of $t = 0.2$, show the capacity of models trained under the same DRL algorithm to generate diverse game episodes within 25 attempts. The

---

**Algorithm 1** Measuring Policy Diversity

---

**Input:** Policy $\pi$, Environment $\mathcal{E}$, Similarity measure $M$
**Input:** Similarity threshold $t$, Number of trajectories $N$
 1: Initialize $S$ (store trajectories) and diverse trajectory count $d = 0$
 2: **for** $i = 1$ **to** $N$ **do**
 3:     Generate a trajectory $\tau_i \sim \pi, \mathcal{E}$
 4:     Set *is_diverse* = **true**
 5:     **for** each $\tau_j$ in $S$ **do**
 6:         Compute similarity $M(\tau_i, \tau_j)$
 7:         **if** $M(\tau_i, \tau_j) \geq t$ **then**
 8:             *is_diverse* = **false**
 9:             **break**
10:         **end if**
11:     **end for**
12:     **if** *is_diverse* **then**
13:         $d = d + 1$
14:     **end if**
15:     Store $\tau_i$ in $S$
16: **end for**
**Output:** Return $d$ (diverse trajectory count) and $N$ (total trajectories)

---

Table 7: Averaged diverse trajectory count of different DRL algorithms across 7 Atari games in 25 episodes.

| DRL Algorithm | Asterix | Breakout | MsPacman | Pong | Qbert | Seaquest | SpaceInvaders |
|---|---|---|---|---|---|---|---|
| DQN | 6.00 | 6.00 | 5.33 | 4.00 | 6.00 | 11.00 | 25.00 |
| C51 | 6.00 | 7.00 | 7.00 | 6.00 | 7.00 | 21.00 | 25.00 |
| Rainbow | 8.67 | 5.33 | 8.00 | 5.00 | 5.00 | 24.00 | 25.00 |
| IQN | **25.00** | **14.00** | **10.00** | **9.00** | **10.00** | **25.00** | 25.00 |

IQN algorithm displays higher diversity across games, consistent with its risk sampling feature. For more details and use cases, please refer to Appendix A.5.

## 8   Conclusion and Future Works

In this research, we introduced three techniques to enhance discrete playstyle measures: adopting a multiscale state space, using perceptual similarity rooted in human cognition, and applying the *Jaccard index* to observed data. These advancements have been incorporated into playstyle measurement for the first time and collectively give rise to our playstyle measure, *Playstyle Similarity*. This measure stands out in terms of accuracy and explainability, requiring minimal predefined rules and data. Notably, the integration of a multiscale state space expands the measure's applicability, particularly for games that have a trade-off between small and large state spaces for game details. Furthermore, our literature review and theoretical proof about human perception bridge the gap between distance similarity and human cognition in playstyle. In addition to the common accuracy evaluations in our experiments, we also conducted a series of statistical tests using McNemar's test (McNemar, 1947) to report some results in the main paper with p-values in Section A.8. These tests help to determine whether two results have statistical significance rather than showing differences due to sampling uncertainty.

The *Playstyle Similarity* measure offers new potential for end-to-end game analysis and AI training with specific playstyles, such as diversity analysis or human-like behaviors (Fujii et al., 2013). As an example, we propose an algorithm to quantify the diversity among DRL models, and the playstyle classification tasks on human playstyles in RGSK and Go also support future applications for human-like agents. These insights emphasize that AI development can extend beyond simple measures like scores or win rates in games with

high-dimensional observations, encompassing more behavioral patterns. Additionally, the quantification of policy diversity becomes more tangible.

In conclusion, despite our validation efforts across several platforms, many games remain unexplored. Furthermore, some playstyles can be shared across different games, existing in similar scenarios for game recommendation systems (Fear, 2023). Beyond gaming, playstyle measures can provide more behavioral information for other decision-making topics, such as AI safety (Amodei et al., 2016) and the interactions among language model agents (OpenAI, 2024; Park et al., 2023).

### Broader Impact Statement

We advise caution when using these measures for analyzing policy diversity due to potential sensitivities to discretization techniques and inherent explainability challenges in neural networks. Additionally, employing unsupervised methods could raise privacy concerns, as they facilitate targeted advertising or fraudulent activities without the need for costly playstyle labeling.

### Acknowledgments

We would like to express our sincere gratitude to Chen (2023) for generously sharing the Go playstyle dataset based on the MiniZero Framework (Wu et al., 2024), which was instrumental in training our discrete encoder and conducting the playstyle tests. Additionally, we appreciate the valuable feedback and insights provided by the reviewers, which greatly helped in refining and improving this work. Finally, we would like to thank Ubitus K.K. for motivating the research topics related to video game playstyles in our early studies.

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

# A    Appendix

## A.1    A Proof of the Perceptual Kernel

In the main paper, we claim that $P(d) = \frac{1}{e^d}$ is the if and only if kernel function (as discussed in Section 3.2). We provide a proof of this claim using differential equations.

We make some assumptions about the perceptual kernel. First, $P(d)$ is a function that maps the distance $d$ between two given action distributions to a probability value describing their similarity. Since distance is a continuous random variable, we use a probability density function $f(d)$ to describe the mapping function $P(d)$.

We might intuitively think of $P(d)$ as equal to $f(d)$, even though the probability density value is not the same as the probability value. Thus, we use a cumulative distribution function $F(d)$ to describe $P(d)$. We redefine a real-valued random variable $D$ as the distance variable, where $d \in D$, and a random variable $X$, where $x \in X$, and $D \to X : X(d) = -d$. The probability of similarity, denoted as $P(X \leq x)$, is derived from the distance value $d$. Thus, $F_X(x) = P(X \leq x)$, and from the assumptions of similarity, $\lim_{x \to -\infty} F_X(x) = 0$, and $F_X(0) = 1$. Also, $F_X(x)$ can be described with a probability density function $f_X(x)$ as follows:

$$F_X(x) = \int_{-\infty}^{x} f_X(t)dt \tag{6}$$

The corresponding equation to $F_D(d) = P(d)$ becomes:

$$F_D(d) = \int_{d}^{\infty} f_D(t)dt \tag{7}$$

Additionally, we adopt another assumption from the field of psychophysics known as the *Weber–Fechner law* Fechner (1966). Fechner's law states that the relationship between stimulus $S$ and perception $p$ is logarithmic and can be described as a differential equation:

$$dp = k\frac{dS}{S} \tag{8}$$

Here, $k$ is a constant depending on the sense and type of stimulus.

By integrating the equation, we obtain:

$$p = k \ln S + C \tag{9}$$

Where $C$ is a constant of integration, and it is defined in Fechner's law assuming that the perceived stimulus becomes zero at some threshold stimulus $S_0$, where $p = 0$, and $S = S_0$. Thus, $C$ can be calculated as follows:

$$C = -k \ln S_0 \tag{10}$$

Combining Equation 9 and Equation 10, Fechner's law Fechner (1966) is:

$$p = k \ln \frac{S}{S_0} \tag{11}$$

Now, we discuss the roles of $p$ and $S$ in our similarity scenario to construct a probability density function $f_D(d)$. We assume that similarity weakens as the distance increases, and there is a finite maximal similarity when the distance is 0. Additionally, the density value and distance are always non-negative.

We first assume that $p$ is the density value of similarity and consider Equation 11. There are two cases:

1. If $S$ represents distance and $k$ is positive, this is incorrect since $p$ approaches $-\infty$ as $S \to 0$.

2. If we change the growth direction of distance so that $k$ is negative, this is still incorrect since $p$ still approaches $\infty$ as $S \to 0$.

Considering invert Equation 11 as follows:

$$S = S_0 \exp\left(\frac{p}{k}\right) \tag{12}$$

Now, we assume that $S$ is the density value of similarity and consider Equation 12. There are two cases:

1. If $p$ represents distance and $k$ is positive, this is incorrect since $S$ approaches $\infty$ as $p \to \infty$, although there is a finite maximal value $S_0$ when $p = 0$.

2. If we change the growth direction of distance so that $k$ is negative, this meets our desire since $S$ approaches 0 as $p \to \infty$, there is a finite maximal value $S_0$ when $p = 0$, and the density value is always non-negative.

Finally, we can simplify the equations by assuming, for the sake of simplicity, that $S_0$ equals 1. This assumption is based on the intuition that the trend of decreasing similarity and increasing distance is similar around distance 0 in various scenarios:

$$f_D(d) = \exp\left(\frac{d}{k}\right) \tag{13}$$

Returning to Equation 7, $F_D(d)$ can be described as follows:

$$\begin{aligned}
F_D(d) &= \int_d^\infty f_D(t)dt \\
&= \int_d^\infty \exp\left(\frac{t}{k}\right)dt \\
&= \left(\lim_{t\to\infty} k\exp\left(\frac{t}{k}\right) + C'\right) - \left(k\exp\left(\frac{d}{k}\right) + C'\right) \\
&= -k\exp\left(\frac{d}{k}\right)
\end{aligned} \tag{14}$$

Considering that the sum of density values must be 1 over $(-\infty, \infty)$, we rewrite Equation 6 as follows:

$$\begin{aligned}
\lim_{x\to\infty} F_X(x) &= \int_{-\infty}^x f_X(t)dt \\
&= \int_{-\infty}^\infty f_X(t)dt \\
&= 1
\end{aligned} \tag{15}$$

The corresponding equation to $F_D(d) = P(d)$ becomes:

$$
\begin{aligned}
\lim_{d \to -\infty} F_D(d) &= \int_d^\infty f_D(t)dt \\
&= \int_{-\infty}^\infty f_D(t)dt \\
&= \int_{-\infty}^0 f_D(t)dt + \int_0^\infty f_D(t)dt \\
&= 0 + \int_0^\infty f_D(t)dt \\
&= 1 \\
\implies F_D(0) &= 1
\end{aligned}
\tag{16}
$$

Combining Equation 14 and Equation 16:

$$
\begin{aligned}
F_D(0) &= -k \exp\left(\frac{0}{k}\right) \\
&= -k \\
&= 1 \\
\implies k &= -1 \\
\implies F_D(d) &= \exp\left(\frac{d}{-1}\right) \\
\implies F_D(d) &= e^{\frac{1}{d}} \\
\implies P(d) &= e^{\frac{1}{d}}
\end{aligned}
\tag{17}
$$

Therefore, we have verified the claim that $P(d) = \frac{1}{e^d}$. If there is a case where $S_0 \neq 1$, it is straightforward to derive the equations from Equation 12 to 17.

Besides, the expected value of distance is 1 can be obtained by the equations as follows:

$$
\begin{aligned}
\mathbb{E}[D] &= \int_{-\infty}^\infty x f_D(x)dx \\
&= 0 + \int_0^\infty x f_D(x)dx \\
&= \int_0^\infty x \frac{1}{e^x} dx \\
&= \left(\lim_{t \to \infty} \frac{-t-1}{e^t} + C'\right) - \left(\frac{-0-1}{e^0} + C'\right) \\
&= 0 - (-1) \\
&= 1
\end{aligned}
\tag{18}
$$

This concept of expected value is used to scaling the distance value in different scenarios, as described in Section 3.2 with the notation $\overline{D}_\Phi^M$.

## A.2 Perceptual Similarity Under Different State Spaces

There are various methods for generating discrete representations, and the effectiveness of perceptual similarity may vary under these representations, especially when combined with our proposed multiscale state space. In this section, we explore the impact of different state space choices on perceptual similarity.

### A.2.1 Bhattacharyya distance implementation

In this paper, we also provide some variants of *Playstyle Similarity*, which use Bhattacharyya distance or coefficient as an alternative to the 2-Wasserstein metric to assess the difference in playstyle from a different perspective. Bhattacharyya distance is related to the overlapping region between two distributions, and it is defined through Bhattacharyya coefficient $BC$. The value range of $BC$ is $[0, 1]$, and the corresponding distance $D_B$ is $D_B = -ln(BC)$. For discrete probability distribution, it is simple to compute the Bhattacharyya coefficient: $BC(P, Q) = \sum_{x \in \mathcal{X}} \sqrt{P(x)Q(x)}$. However, it is more challenging to calculate for continuous probability distributions, as in the case of actions in racing games like TORCS, since it involves the integration of probability density functions: $BC(P, Q) = \int_{x \in \mathcal{X}} \sqrt{p(x)q(x)}$. Thus, we adopt the formulation of multivariate normal distributions of Bhattacharyya distance ($D_B$) (Bhattacharyya, 1946) as follows, where $p_i = \mathcal{N}(\mu_i, \Sigma_i)$:

$$D_B(p_1, p_2) = \frac{1}{8}(\mu_1 - \mu_2)^T \Sigma^{-1}(\mu_1 - \mu_2) + \frac{1}{2}\ln(\frac{\det\Sigma}{\sqrt{\det\Sigma_1 \det\Sigma_2}})D_B,$$
$$\text{where} \quad \Sigma = \frac{\Sigma_1 + \Sigma_2}{2} \tag{19}$$

Additionally, we clip the maximum Bhattacharyya distance to 10 to prevent an extremely large value from affecting the average scaling ($\frac{1}{e^{10}} = 0.00004539992 \approx 0\%$). The small value $\epsilon$ for dealing with singular matrices in matrix determinant calculation is set to 1e-8.

Recalling our earlier discussion, we mentioned that the Wasserstein distance can be likened to the 'effort' required to transition between different playstyle action distributions (as described in Section 2.2). The Bhattacharyya distance, in contrast, isn't about this 'effort'. Instead, it gauges the likelihood that two playstyles will result in the same action. This is due to its relation to the overlapping regions between two distributions. Thus, while the *Playstyle Similarity* is built on the idea of the effort needed to change playstyles, the *Playstyle BC Similarity* (or its variant *Playstyle BD Similarity*) is built on the frequency of identical actions. This distinction might relate to different roles within a game. For instance, a player in the game might be more concerned with the effort required to shift playstyles, while an observer might focus more on the actions they witness. Think of it this way: players exert effort, like moving their fingers to press buttons or manipulate a joystick or even a mental effort to change their belief of playing. The observer, on the other hand, sees only the outcome of these actions, without much insight into the effort involved.

### A.2.2 Multiscale State Space with HSD

Figure 9 presents the results of experiments conducted with a multiscale state space $\{1, 2^{20}, 256^{\text{res}}\}$ generated from HSD models, as described in Section 4.2. The results indicate an improvement in accuracy for TORCS, while there is no clear improvement in RGSK. Notably, in RGSK, the accuracy of the perceptual kernel with sample sizes $2^5$ to $2^8$ decreases, suggesting that detailed information for distinguishing these styles has a negative effect. To further investigate, we conducted two ablation studies to understand the effectiveness of the proposed measures for playstyle similarity. The first study focuses on using only the base hierarchy of HSD with a very large state space $\{256^{\text{res}}\}$, while the second study explores the use of a single-state state space $\{1\}$ to assess the measures. Figure 9 illustrates that the measurement is unstable when there are few intersecting samples in a very large state space. However, the negative effect of detailed information is mitigated when considering intersection over union. Figure 9 also shows that even with single state space, the action statistics of the dataset can offer some information to differentiate playstyle, especially in RGSK, where Lin et al. (2021) made their human players follow some playstyles closely related to the keyboard actions, such as using the nitro system or braking in the racing game.

### A.2.3 Discrete Representation from Downsampling

There are several existing methods for generating discrete representations. One conventional method for image data is downsampling to a lower resolution. While the downsampling parameters often require tuning for effective processing, it is a straightforward method that does not require training a neural network model. Previous work by Lin et al. (2021) attempted to use low-resolution downsampling as a discretization method, but they encountered challenges due to the lack of intersection states in their settings.

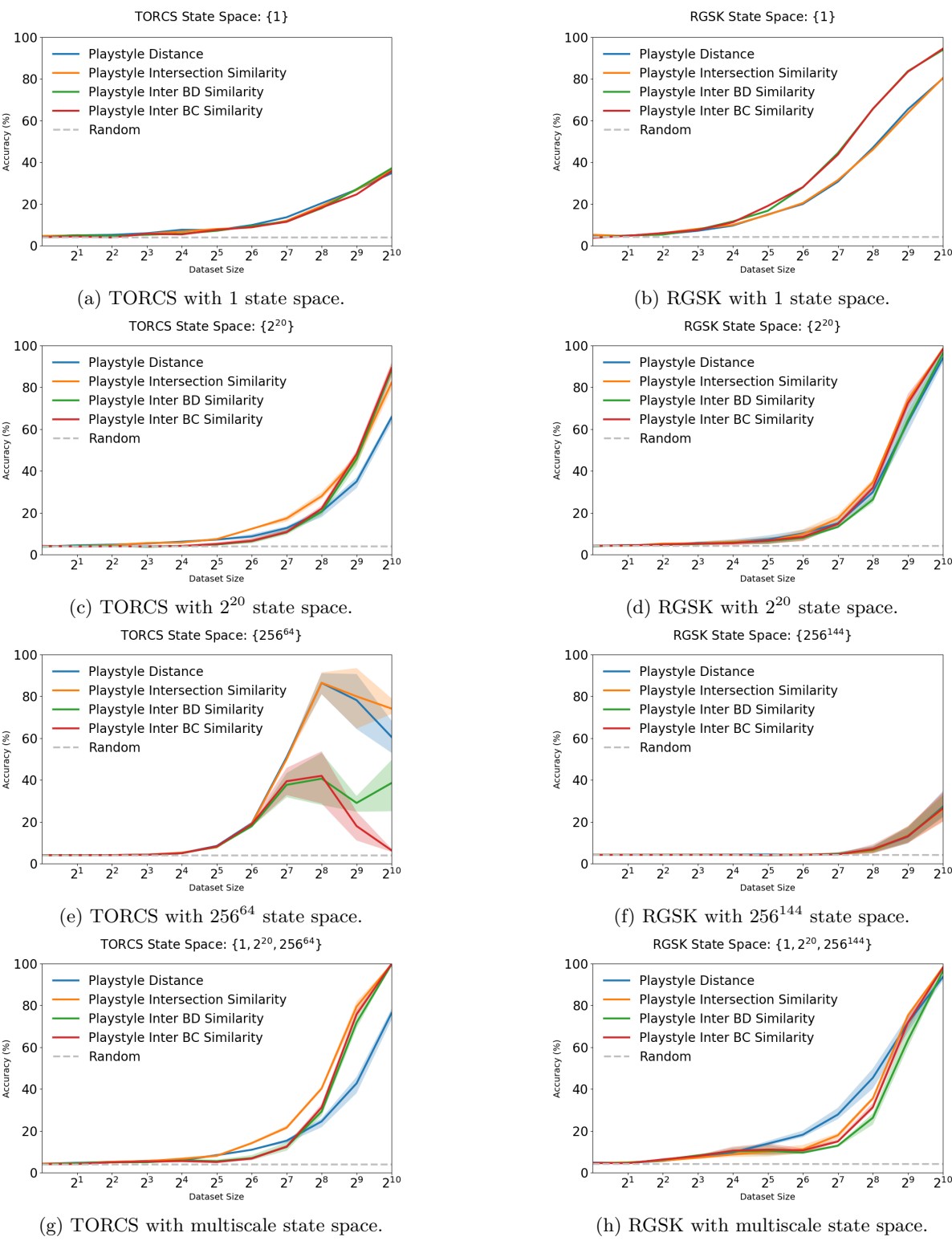

Figure 9: Comparison of Efficacy: Using different state spaces in discrete playstyle measurement for TORCS and RGSK, including single state space $\{1\}$, $\{2^{20}\}$ state space, the base hierarchy of HSD with state space $\{256^{res}\}$, and multiscale state space $\{1, 2^{20}, 256^{res}\}$. The shaded area indicates the range between the minimum and maximum accuracy among three encoder models.

In our experiments, we explore the use of downsampling to create discrete representations with different state spaces. In TORCS, we map original game screen observations to three levels of state space:

1. 1: Basic mapping with state space 1, which maps all observations identically.

2. $16^{8\times8}$: Downsampling from $4 \times [64,64,3]$ 256-intensity observations to $1 \times [8,8,1]$ 16-intensity observations.

3. $16^{8\times8\times4}$: Downsampling from $4 \times [64,64,3]$ 256-intensity observations to $4 \times [8,8,1]$ 16-intensity observations.

For RGSK, we similarly map original game screen observations to three levels of state space:

1. 1: Basic mapping with state space 1, which maps all observations identically.

2. $16^{9\times16}$: Downsampling from $4 \times [72,128,3]$ 256-intensity observations to $1 \times [9,16,1]$ 16-intensity observations.

3. $16^{9\times16\times4}$: Downsampling from $4 \times [72,128,3]$ 256-intensity observations to $4 \times [9,16,1]$ 16-intensity observations.

The results in Figure 10 show that downsampling can be a viable discretization method in some cases, but overall, the measurement is either unstable or shows no significant difference compared in these measures. These results highlight the importance of having discrete representations with high quality, providing proper granularity for playstyle features.

### A.3 More Results of Video Game Evaluation

In this supplementary section, we delve deeper into the results to provide a comprehensive analysis of measures evaluations under different state spaces.

#### A.3.1 Statistics of Each Discrete Encoder with Multiscale State Space

In this subsection, we provide the mean and standard deviation of the accuracy from Table 2 in Table 8. There are 3 available discrete encoders trained using the HSD method in our experiments. Their mean and standard deviation values are calculated based on 100 rounds of random subsampling, with each dataset in a subsampling consisting of 1024 observation-action pairs. We employ sampling without replacement (there are no duplicated observation-action pairs in the two comparing datasets) except for Atari games, where some games have fewer than $2 \times 1024$ pairs for sampling double the size of the dataset.

#### A.3.2 Individual Atari game results with multiscale state space

Figure 11 shows the relationship between playstyle classification accuracy and sampled dataset size for the seven Atari games. *Playstyle Similarity* ($PS_{\Phi}^{\cup}$) and its variant *Playstyle BC Similarity* ($PS_{\Phi}^{\cup BC}$) have nearly the same performance, and *Playstyle Jaccard Index* ($J_{\Phi}$) can have a decent result. This evidence justifies that some playstyles, especially in a deterministic environment, can be differentiated solely with observations, which explains why the work by Eysenbach et al. (2019) considers states only for diversity.

#### A.3.3 Atari game results with a smaller state space

Figure 12 shows the relationship between playstyle classification accuracy and sampled dataset size for the seven Atari games and the combined version (Atari Console). These results show that measures with intersection over union still perform well in Atari games even with a smaller state space. Although it seems that *Playstyle Jaccard Index* is a decent and easy measure, we know that it theoretically does not work as long as all states are visited in the sampled dataset, as described in Section 3.3. This potential problem is discussed in more detail in Section A.3.4, where even with a state space of $2^{20}$, the *Playstyle Jaccard Index* may not perform well.

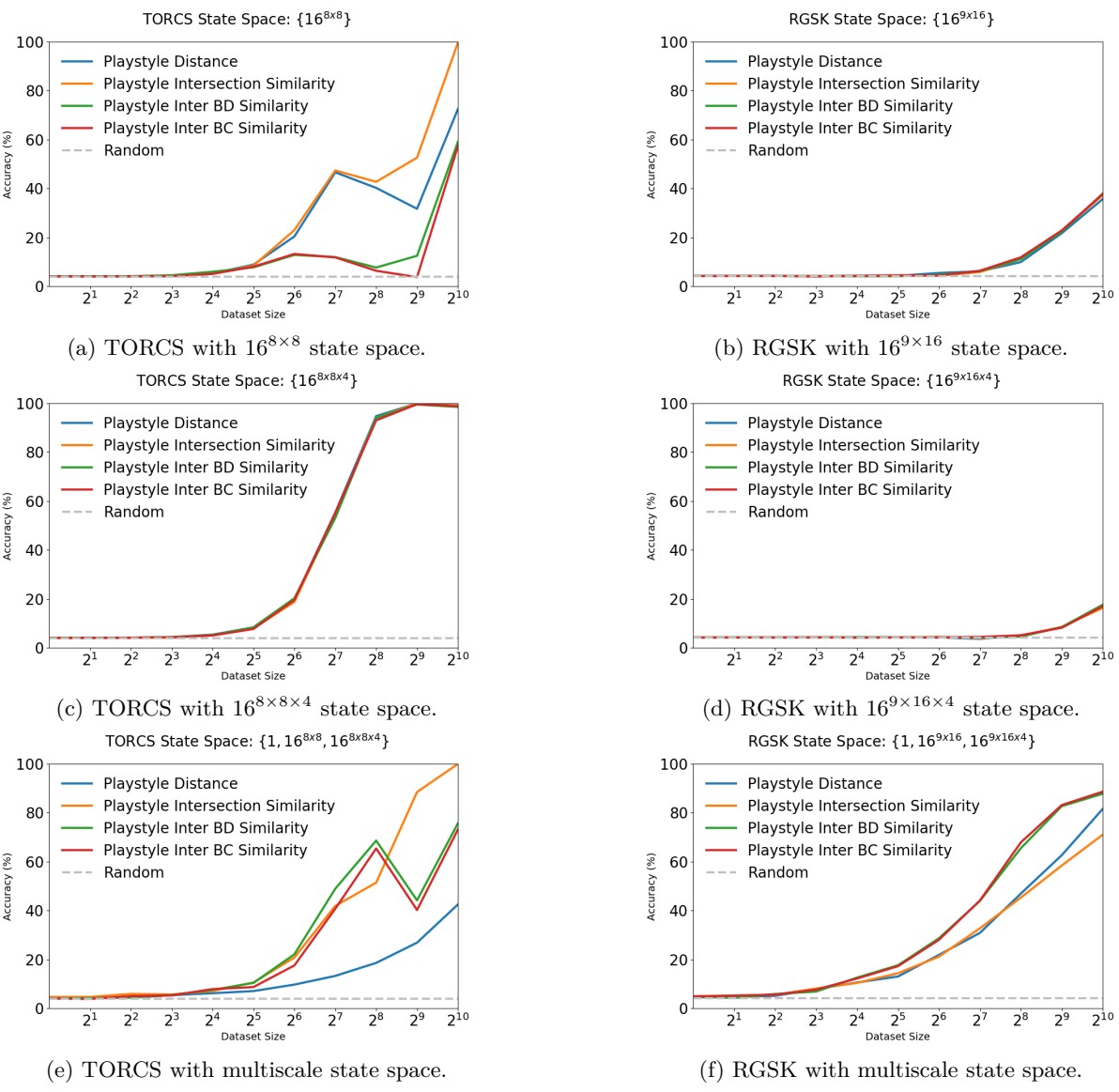

Figure 10: Evaluation of discrete representations using downsampling, considering intersection states in TORCS with state spaces $\{16^{8\times8}\}$, $\{16^{8\times8\times4}\}$, and $\{1, 16^{8\times8}, 16^{8\times8\times4}\}$, and in RGSK with state spaces $\{16^{9\times16}\}$, $\{16^{9\times16\times4}\}$, and $\{1, 16^{9\times16}, 16^{9\times16\times4}\}$.

Table 8: Playstyle accuracy (%) ± standard deviation (%) when employing various discrete state spaces in the multiscale version of *Playstyle Distance*, with a sample threshold count $t$ for intersecting states.

| | $2^{20}$ $t$=2 | $2^{20}$ $t$=1 | $256^{\mathrm{res}}$ $t$=2 | $256^{\mathrm{res}}$ $t$=1 | **mix** $t$=2 | **mix** $t$=1 |
|---|---|---|---|---|---|---|
| TORCS | $73.3 \pm 8.2$ | $66.5 \pm 7.9$ | $4.3 \pm 3.1$ | $60.9 \pm 9.4$ | $\mathbf{77.3} \pm 7.4$ | $\mathbf{77.5} \pm 7.9$ |
| Encoder1 | $70.3 \pm 9.1$ | $63.8 \pm 8.5$ | $5.8 \pm 4.3$ | $55.0 \pm 9.9$ | $\mathbf{74.2} \pm 8.1$ | $\mathbf{75.9} \pm 8.3$ |
| Encoder2 | $71.6 \pm 9.1$ | $70.2 \pm 7.6$ | $3.6 \pm 3.5$ | $60.1 \pm 9.8$ | $\mathbf{77.0} \pm 7.1$ | $\mathbf{75.5} \pm 7.9$ |
| Encoder3 | $77.9 \pm 6.6$ | $65.7 \pm 7.7$ | $3.4 \pm 1.4$ | $67.6 \pm 8.6$ | $\mathbf{80.8} \pm 7.1$ | $\mathbf{81.1} \pm 7.6$ |
| RGSK | $79.2 \pm 7.9$ | $\mathbf{93.7} \pm 4.7$ | $5.7 \pm 2.5$ | $25.6 \pm 7.2$ | $78.8 \pm 7.5$ | $\mathbf{93.5} \pm 4.3$ |
| Encoder1 | $80.4 \pm 7.5$ | $\mathbf{93.0} \pm 5.0$ | $5.6 \pm 2.4$ | $23.4 \pm 7.6$ | $81.2 \pm 7.4$ | $\mathbf{93.4} \pm 4.6$ |
| Encoder2 | $80.2 \pm 8.1$ | $\mathbf{97.4} \pm 2.9$ | $5.0 \pm 2.1$ | $20.5 \pm 6.9$ | $78.2 \pm 7.8$ | $\mathbf{96.9} \pm 3.1$ |
| Encoder3 | $76.8 \pm 8.1$ | $\mathbf{90.7} \pm 6.1$ | $6.3 \pm 2.9$ | $32.9 \pm 7.3$ | $77.2 \pm 7.4$ | $\mathbf{90.3} \pm 5.3$ |
| Asterix | $\mathbf{99.9} \pm 0.5$ | $\mathbf{100} \pm 0$ | $49.6 \pm 7.7$ | $32.7 \pm 8.0$ | $\mathbf{100} \pm 0$ | $\mathbf{100} \pm 0$ |
| Encoder1 | $\mathbf{99.9} \pm 0.7$ | $\mathbf{100} \pm 0$ | $48.7 \pm 7.3$ | $48.0 \pm 6.9$ | $\mathbf{100} \pm 0$ | $\mathbf{100} \pm 0$ |
| Encoder2 | $\mathbf{99.9} \pm 0.7$ | $\mathbf{100} \pm 0$ | $50.0 \pm 7.2$ | $24.5 \pm 8.3$ | $\mathbf{100} \pm 0$ | $\mathbf{100} \pm 0$ |
| Encoder3 | $\mathbf{100} \pm 0$ | $\mathbf{100} \pm 0$ | $50.2 \pm 8.5$ | $26.7 \pm 8.8$ | $\mathbf{100} \pm 0$ | $\mathbf{100} \pm 0$ |
| Breakout | $\mathbf{99.4} \pm 1.6$ | $\mathbf{99.9} \pm 0.6$ | $65.9 \pm 8.5$ | $29.9 \pm 9.4$ | $\mathbf{99.8} \pm 1.1$ | $\mathbf{99.9} \pm 0.2$ |
| Encoder1 | $\mathbf{98.9} \pm 2.4$ | $\mathbf{99.8} \pm 1.1$ | $67.5 \pm 8.8$ | $32.7 \pm 9.6$ | $\mathbf{99.8} \pm 1.2$ | $\mathbf{99.9} \pm 0.5$ |
| Encoder2 | $\mathbf{99.4} \pm 1.8$ | $\mathbf{99.9} \pm 0.9$ | $70.9 \pm 7.8$ | $30.8 \pm 9.5$ | $\mathbf{99.8} \pm 1.1$ | $\mathbf{100} \pm 0$ |
| Encoder3 | $\mathbf{99.9} \pm 0.5$ | $\mathbf{100} \pm 0$ | $59.2 \pm 8.9$ | $26.4 \pm 9.0$ | $\mathbf{99.8} \pm 1.0$ | $\mathbf{100} \pm 0$ |
| MsPac. | $\mathbf{99.9} \pm 0.5$ | $\mathbf{100} \pm 0$ | $92.8 \pm 4.0$ | $\mathbf{100} \pm 0$ | $\mathbf{100} \pm 0$ | $\mathbf{100} \pm 0$ |
| Encoder1 | $\mathbf{99.6} \pm 1.4$ | $\mathbf{100} \pm 0$ | $93.6 \pm 4.1$ | $\mathbf{100} \pm 0$ | $\mathbf{100} \pm 0$ | $\mathbf{100} \pm 0$ |
| Encoder2 | $\mathbf{100} \pm 0$ | $\mathbf{100} \pm 0$ | $92.8 \pm 4.3$ | $\mathbf{100} \pm 0$ | $\mathbf{100} \pm 0$ | $\mathbf{100} \pm 0$ |
| Encoder3 | $\mathbf{100} \pm 0$ | $\mathbf{100} \pm 0$ | $92.1 \pm 3.7$ | $\mathbf{100} \pm 0$ | $\mathbf{100} \pm 0$ | $\mathbf{100} \pm 0$ |
| Pong | $92.1 \pm 2.7$ | $92.3 \pm 2.6$ | $50.7 \pm 9.5$ | $52.2 \pm 9.9$ | $\mathbf{93.1} \pm 3.2$ | $92.4 \pm 2.6$ |
| Encoder1 | $92.0 \pm 2.7$ | $92.6 \pm 2.5$ | $52.2 \pm 9.4$ | $52.6 \pm 9.0$ | $\mathbf{93.7} \pm 3.4$ | $92.5 \pm 2.8$ |
| Encoder2 | $92.4 \pm 2.9$ | $92.2 \pm 2.7$ | $48.3 \pm 9.1$ | $51.4 \pm 10.4$ | $\mathbf{93.1} \pm 3.1$ | $92.4 \pm 2.6$ |
| Encoder3 | $92.0 \pm 2.6$ | $92.2 \pm 2.5$ | $51.5 \pm 10.0$ | $52.6 \pm 10.2$ | $\mathbf{92.5} \pm 3.1$ | $92.5 \pm 2.5$ |
| Qbert | $\mathbf{100} \pm 0$ | $\mathbf{100} \pm 0$ | $90.1 \pm 5.3$ | $91.6 \pm 4.6$ | $99.9 \pm 0.5$ | $\mathbf{100} \pm 0$ |
| Encoder1 | $\mathbf{100} \pm 0$ | $\mathbf{100} \pm 0$ | $95.4 \pm 3.8$ | $99.9 \pm 0.7$ | $99.9 \pm 0.5$ | $\mathbf{100} \pm 0$ |
| Encoder2 | $\mathbf{100} \pm 0$ | $\mathbf{100} \pm 0$ | $82.1 \pm 6.7$ | $82.8 \pm 8.0$ | $99.9 \pm 0.5$ | $\mathbf{100} \pm 0$ |
| Encoder3 | $\mathbf{100} \pm 0$ | $\mathbf{100} \pm 0$ | $92.8 \pm 5.5$ | $92.2 \pm 5.1$ | $99.9 \pm 0.5$ | $\mathbf{100} \pm 0$ |
| Seaquest | $\mathbf{99.7} \pm 1.2$ | $\mathbf{99.9} \pm 0.6$ | $17.1 \pm 5.2$ | $16.7 \pm 4.9$ | $\mathbf{99.9} \pm 0.3$ | $\mathbf{99.9} \pm 0.2$ |
| Encoder1 | $\mathbf{99.9} \pm 0.9$ | $\mathbf{99.9} \pm 0.5$ | $17.1 \pm 5.3$ | $15.8 \pm 4.5$ | $\mathbf{100} \pm 0$ | $\mathbf{100} \pm 0$ |
| Encoder2 | $\mathbf{99.9} \pm 0.7$ | $\mathbf{100} \pm 0$ | $17.2 \pm 5.3$ | $17.3 \pm 5.3$ | $\mathbf{99.9} \pm 0.9$ | $\mathbf{99.9} \pm 0.5$ |
| Encoder3 | $\mathbf{99.7} \pm 1.2$ | $\mathbf{99.6} \pm 1.4$ | $17.2 \pm 5.0$ | $17.0 \pm 5.0$ | $\mathbf{100} \pm 0$ | $\mathbf{100} \pm 0$ |
| SpaceIn. | $98.7 \pm 2.3$ | $\mathbf{99.7} \pm 1.2$ | $50.4 \pm 5.7$ | $49.6 \pm 8.4$ | $\mathbf{99.9} \pm 0.5$ | $\mathbf{99.9} \pm 0.6$ |
| Encoder1 | $98.8 \pm 2.5$ | $\mathbf{99.9} \pm 0.9$ | $51.2 \pm 8.4$ | $50.2 \pm 7.8$ | $\mathbf{99.8} \pm 1.0$ | $\mathbf{99.9} \pm 0.7$ |
| Encoder2 | $99.6 \pm 1.4$ | $\mathbf{99.9} \pm 0.9$ | $50.3 \pm 0.1$ | $50.2 \pm 8.7$ | $\mathbf{100} \pm 0$ | $\mathbf{99.9} \pm 0.5$ |
| Encoder3 | $97.9 \pm 3.0$ | $\mathbf{99.3} \pm 1.9$ | $49.9 \pm 8.8$ | $48.6 \pm 8.8$ | $\mathbf{99.9} \pm 0.5$ | $\mathbf{99.9} \pm 0.7$ |

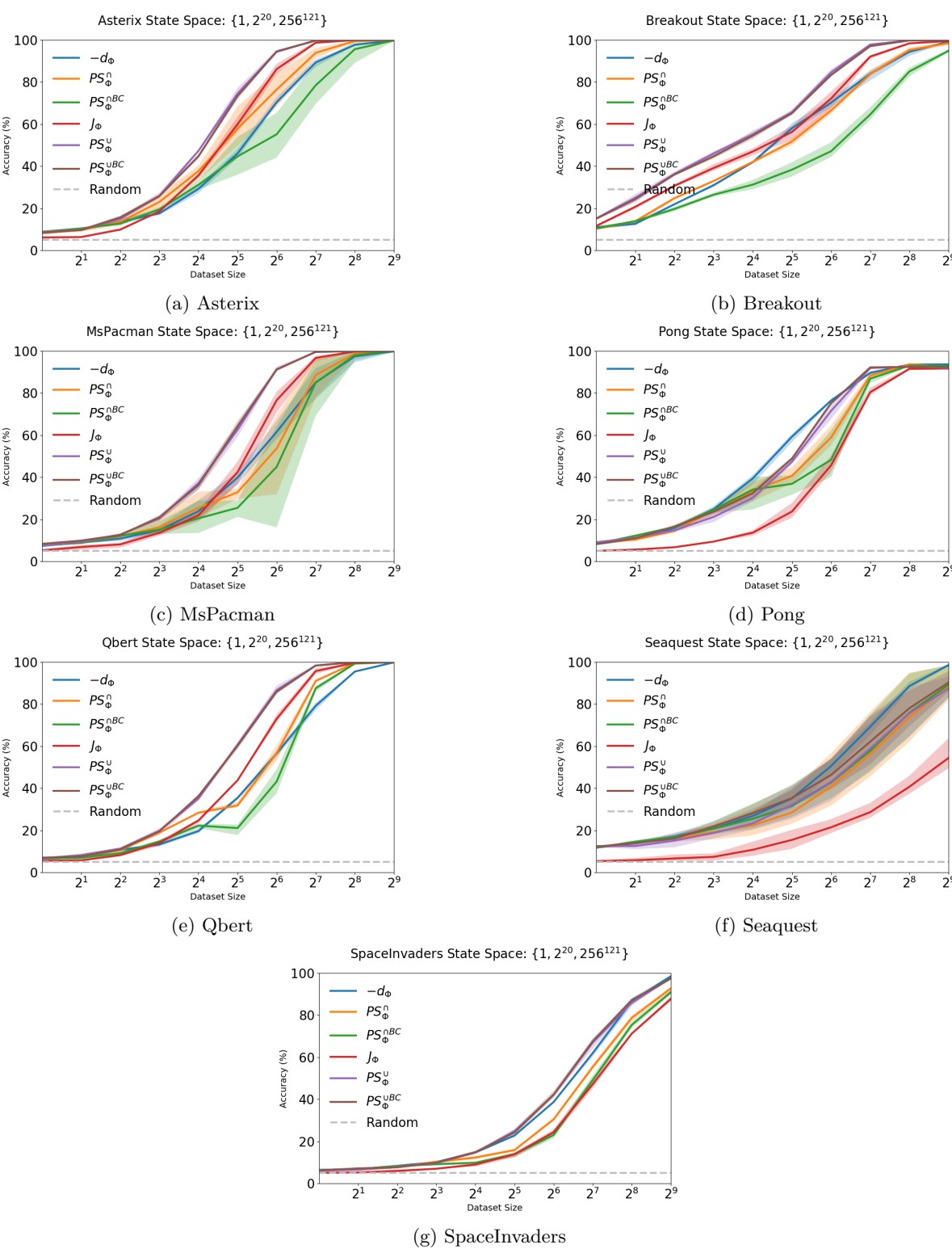

Figure 11: Playstyle Measure Evaluation in Atari games. The plots showcase the efficacy of different measures in the context of the "Full Data Evaluation" subsection. The shaded area indicates the range between min and max accuracy among three encoder models.

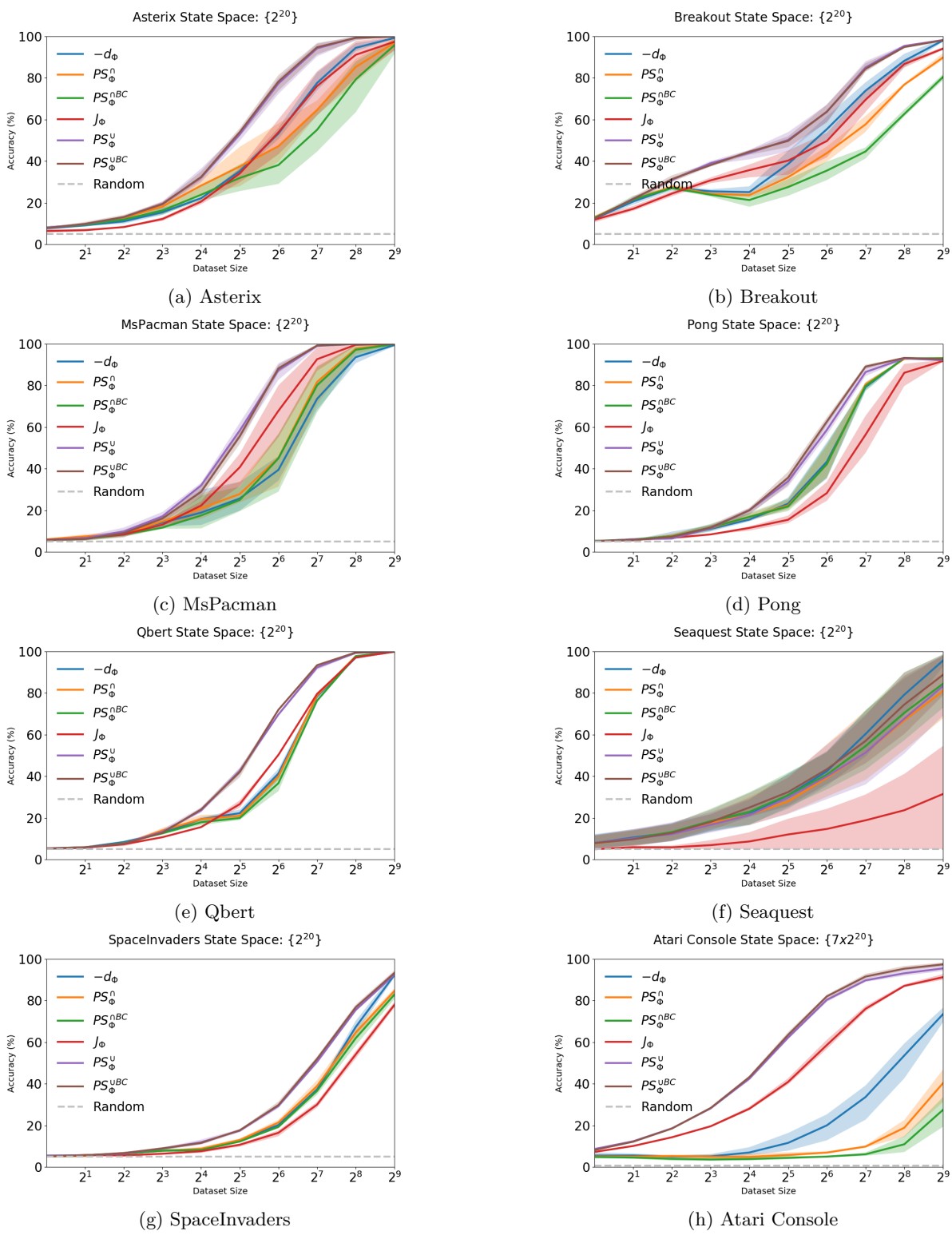

Figure 12: Playstyle Measure Evaluation in Atari games. The plots showcase the efficacy of different measures with a $2^{20}$ state space from HSD models. The shaded area indicates the range between min and max accuracy among three encoder models.

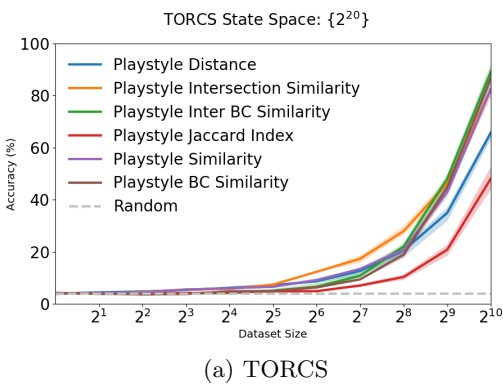

(a) TORCS

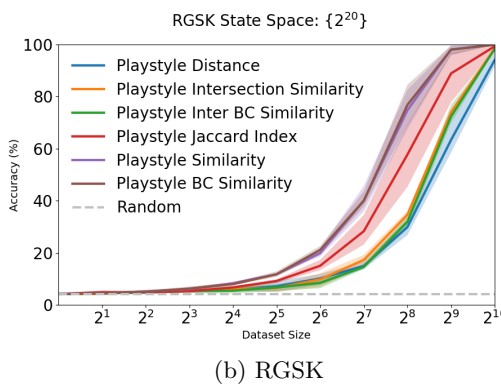

(b) RGSK

Figure 13: Playstyle Measure Evaluation in two racing games. The plots showcase the efficacy of different measures with a $2^{20}$ state space from HSD models. The shaded area indicates the range between min and max accuracy among three encoder models.

### A.3.4 TORCS and RGSK with a smaller state space

In this section, we conducted experiments using a reduced state space of $2^{20}$ for the two racing games, TORCS and RGSK, without employing the multiscale technique.

Figure 13 illustrates that the *Playstyle Jaccard Index* performs the poorest in TORCS and exhibits slightly inferior performance to *Playstyle Similarity* and *Playstyle BC Similarity* in RGSK. This observation provides valuable insights into the suitability of the *Playstyle Jaccard Index* for precise measurements, particularly in scenarios involving randomness (e.g., TORCS players employing different action noises) or where observations exhibit only slight variations (e.g., stable rule-based AI controllers in TORCS with slightly different target). Further investigation may be warranted to understand the reasons behind these performance differences.

### A.4 Continuous Playstyle Spectrum in TORCS

This experiment investigates the response of similarity measure values to variations in a continuous playstyle spectrum within the TORCS environment. It particularly focuses on whether these measures can accurately rank playstyles, ensuring precise predictions for the closest playstyle (Top-1 similarity) and maintaining a correctly ordered sequence of playstyles based on similarity measures.

Utilizing the TORCS dataset, which includes five levels of target speeds (60, 65, 70, 75, 80) and five levels of action noise, provides a broad spectrum for examining continuous playstyle changes. To illustrate, consider five playstyles labeled A through E, with A' being a variant closely aligned with A. We anticipate that the similarity between A' and A would be the highest, progressively decreasing towards E. This expectation sets the stage for our consistency test, wherein a similarity measure $M$ should validate the order $M(A', A) > M(A', B) > M(A', C) > M(A', D) > M(A', E)$ (Corner Case).

We assess the following measures:

1. Playstyle Distance with a $2^{20}$ state space (baseline)

2. Playstyle Distance using a mixed state space

3. Playstyle Intersection Similarity with a mixed state space

4. Playstyle Jaccard Index with a mixed state space

5. Playstyle Similarity with a mixed state space

With 100 rounds of random subsampling (each dataset consisting of 512 observation-action pairs) and using the first Hierarchical State Discretization (HSD) model, we examine the consistency of similarity values as

Table 9: Consistency count of two continuous playstyle spectrum cases.

|  | Corner Case | Center Case |
| --- | --- | --- |
| Playstyle Distance ($2^{20}$) | 3 | 2 |
| Playstyle Distance (mixed) | 4 | 2 |
| Playstyle Intersection Similarity (mixed) | 8 | 2 |
| Playstyle Jaccard Index (mixed) | 5 | 1 |
| Playstyle Similarity (mixed) | **9** | **3** |

Table 10: Corner Case uses Playstyle Distance ($2^{20}$): consistent count = 3

| Speed60N0 | Speed60 (C) | Speed65 | Speed70 | Speed75 | Speed80 |
| --- | --- | --- | --- | --- | --- |
| N0 (C) | -0.0044 (0.0012) | -0.0048 (0.0010) | -0.0055 (0.0010) | -0.0069 (0.0011) | -0.0100 (0.0014) |
| N1 (C) | -0.0054 (0.0011) | -0.0064 (0.0015) | -0.0069 (0.0016) | -0.0086 (0.0013) | -0.0108 (0.0016) |
| N2 | -0.0064 (0.0010) | -0.0058 (0.0010) | -0.0068 (0.0013) | -0.0086 (0.0010) | -0.0097 (0.0015) |
| N3 | -0.0073 (0.0011) | -0.0078 (0.0013) | -0.0071 (0.0010) | -0.0100 (0.0017) | -0.0118 (0.0016) |
| N4 | -0.0092 (0.0013) | -0.0083 (0.0011) | -0.0089 (0.0013) | -0.0110 (0.0015) | -0.0132 (0.0017) |

playstyle shifts. A decrease in similarity consistent with playstyle changes is marked with a (C), indicating measure reliability.

For instance, choosing Speed60N0 as a target playstyle, we observe how similarity measures adjust across a row or column in response to increasing speed or action noise levels. A practical demonstration reveals how the measure values consistently decrease across increasing speed levels and noise intensities, illustrating the measure's ability to capture playstyle divergence accurately.

Similar analyses extend to a Center Case scenario with Speed70N2 as the target playstyle, emphasizing the necessity for similarity measures to respect two key sequences for consistency:

1. $M(C', C) > M(C', B) > M(C', A)$

2. $M(C', C) > M(C', D) > M(C', E)$

These sequences confirm the measure's capacity to accurately reflect the gradual divergence of playstyles from a central reference point. The detailed findings, presented in Tables 9-19, underscore the nuanced performance of each evaluated measure. Additionally, we present the standard deviation of the measure values, denoted as (std), alongside the measure values themselves. The query playstyle is marked as orange, and those columns with consistency in measure values are marked in blue. For rows with consistency, marked as red, and for those values with consistency in both columns and rows, marked violet.

Table 11: Corner Case uses Playstyle Distance (mix): consistent count = 5

| Speed60N0 | Speed60 (C) | Speed65 | Speed70 | Speed75 (C) | Speed80 |
| --- | --- | --- | --- | --- | --- |
| N0 (C) | -0.0018 (0.0005) | -0.0022 (0.0005) | -0.0028 (0.0006) | -0.0037 (0.0006) | -0.0060 (0.0007) |
| N1 (C) | -0.0024 (0.0005) | -0.0028 (0.0007) | -0.0036 (0.0008) | -0.0043 (0.0007) | -0.0058 (0.0008) |
| N2 | -0.0028 (0.0005) | -0.0027 (0.0006) | -0.0031 (0.0006) | -0.0045 (0.0007) | -0.0056 (0.0008) |
| N3 (C) | -0.0030 (0.0005) | -0.0033 (0.0007) | -0.0035 (0.0006) | -0.0049 (0.0007) | -0.0060 (0.0009) |
| N4 | -0.0038 (0.0006) | -0.0037 (0.0006) | -0.0040 (0.0006) | -0.0055 (0.0008) | -0.0071 (0.0011) |

Table 12: Corner Case uses Playstyle Intersection Similarity (mix): consistent count = 8

| Speed60N0 | Speed60 (C) | Speed65 (C) | Speed70 (C) | Speed75 (C) | Speed80 (C) |
|---|---|---|---|---|---|
| N0 (C) | 0.8014 (0.0206) | 0.7502 (0.0187) | 0.7170 (0.0233) | 0.6538 (0.0231) | 0.5829 (0.0246) |
| N1 (C) | 0.6927 (0.0234) | 0.6865 (0.0240) | 0.6646 (0.0218) | 0.6254 (0.0265) | 0.5508 (0.0250) |
| N2 | 0.6260 (0.0266) | 0.6499 (0.0257) | 0.6354 (0.0299) | 0.5709 (0.0254) | 0.5450 (0.0284) |
| N3 | 0.5813 (0.0282) | 0.5857 (0.0250) | 0.5721 (0.0302) | 0.5507 (0.0276) | 0.4825 (0.0321) |
| N4 (C) | 0.5420 (0.0268) | 0.5390 (0.0298) | 0.5322 (0.0310) | 0.4708 (0.0288) | 0.4544 (0.0328) |

Table 13: Corner Case uses Playstyle Jaccard Index (mix): consistent count = 5

| Speed60N0 | Speed60 (C) | Speed65 | Speed70 (C) | Speed75 (C) | Speed80 (C) |
|---|---|---|---|---|---|
| N0 (C) | 0.0938 (0.0059) | 0.0863 (0.0042) | 0.0841 (0.0044) | 0.0840 (0.0048) | 0.0816 (0.0042) |
| N1 | 0.0845 (0.0046) | 0.0853 (0.0045) | 0.0833 (0.0040) | 0.0821 (0.0043) | 0.0815 (0.0040) |
| N2 | 0.0827 (0.0045) | 0.0816 (0.0037) | 0.0830 (0.0047) | 0.0812 (0.0043) | 0.0799 (0.0042) |
| N3 | 0.0821 (0.0040) | 0.0820 (0.0048) | 0.0790 (0.0042) | 0.0806 (0.0040) | 0.0773 (0.0049) |
| N4 | 0.0780 (0.0046) | 0.0762 (0.0045) | 0.0757 (0.0038) | 0.0750 (0.0042) | 0.0760 (0.0041) |

Table 14: Corner Case uses Playstyle Similarity (mix): consistent count = 9

| Speed60N0 | Speed60 (C) | Speed65 (C) | Speed70 (C) | Speed75 (C) | Speed80 (C) |
|---|---|---|---|---|---|
| N0 (C) | 0.0753 (0.0046) | 0.0650 (0.0035) | 0.0608 (0.0034) | 0.0555 (0.0026) | 0.0472 (0.0035) |
| N1 (C) | 0.0590 (0.0035) | 0.0579 (0.0033) | 0.0549 (0.0034) | 0.0513 (0.0035) | 0.0446 (0.0031) |
| N2 | 0.0519 (0.0036) | 0.0533 (0.0033) | 0.0525 (0.0033) | 0.0458 (0.0032) | 0.0440 (0.0034) |
| N3 (C) | 0.0482 (0.0033) | 0.0473 (0.0034) | 0.0448 (0.0031) | 0.0435 (0.0032) | 0.0382 (0.0030) |
| N4 (C) | 0.0421 (0.0034) | 0.0413 (0.0031) | 0.0402 (0.0035) | 0.0349 (0.0030) | 0.0337 (0.0032) |

Table 15: Center Case uses Playstyle Distance ($2^{20}$): consistent count = 2

| Speed70N2 | Speed60 | Speed65 | Speed70 | Speed75 | Speed80 |
|---|---|---|---|---|---|
| N0 | -0.0068 (0.0013) | -0.0068 (0.0012) | -0.0065 (0.0011) | -0.0068 (0.0012) | -0.0097 (0.0017) |
| N1 | -0.0073 (0.0011) | -0.0069 (0.0013) | -0.0073 (0.0014) | -0.0084 (0.0014) | -0.0095 (0.0014) |
| N2 (C) | -0.0079 (0.0010) | -0.0075 (0.0015) | -0.0069 (0.0013) | -0.0087 (0.0014) | -0.0095 (0.0016) |
| N3 (C) | -0.0085 (0.0012) | -0.0080 (0.0013) | -0.0075 (0.0011) | -0.0090 (0.0014) | -0.0103 (0.0013) |
| N4 | -0.0105 (0.0013) | -0.0093 (0.0014) | -0.0094 (0.0013) | -0.0112 (0.0015) | -0.0127 (0.0016) |

Table 16: Center Case uses Playstyle Distance (mix): consistent count = 2

| Speed70N2 | Speed60 | Speed65 | Speed70 | Speed75 | Speed80 |
|---|---|---|---|---|---|
| N0 | -0.0033 (0.0006) | -0.0029 (0.0006) | -0.0029 (0.0006) | -0.0033 (0.0005) | -0.0046 (0.0007) |
| N1 | -0.0035 (0.0006) | -0.0032 (0.0007) | -0.0034 (0.0007) | -0.0036 (0.0007) | -0.0046 (0.0006) |
| N2 (C) | -0.0038 (0.0005) | -0.0035 (0.0008) | -0.0031 (0.0007) | -0.0037 (0.0006) | -0.0047 (0.0009) |
| N3 (C) | -0.0039 (0.0006) | -0.0036 (0.0006) | -0.0034 (0.0005) | -0.0040 (0.0007) | -0.0047 (0.0007) |
| N4 | -0.0046 (0.0006) | -0.0037 (0.0005) | -0.0040 (0.0006) | -0.0046 (0.0006) | -0.0060 (0.0009) |

Table 17: Center Case uses Playstyle Intersection Similarity (mix): consistent count = 2

| Speed70N2 | Speed60 | Speed65 | Speed70 | Speed75 | Speed80 |
|---|---|---|---|---|---|
| N0 (C) | 0.6774 (0.0241) | 0.6944 (0.0206) | 0.6995 (0.0233) | 0.6625 (0.0246) | 0.6286 (0.0265) |
| N1 | 0.6295 (0.0252) | 0.6609 (0.0259) | 0.6603 (0.0247) | 0.6456 (0.0253) | 0.6031 (0.0281) |
| N2 (C) | 0.5962 (0.0323) | 0.6376 (0.0308) | 0.7100 (0.0231) | 0.6086 (0.0291) | 0.6034 (0.0273) |
| N3 | 0.5891 (0.0272) | 0.5844 (0.0277) | 0.5999 (0.0261) | 0.5764 (0.0297) | 0.5486 (0.0256) |
| N4 | 0.5129 (0.0306) | 0.5664 (0.0274) | 0.5503 (0.0278) | 0.5135 (0.0297) | 0.4841 (0.0253) |

Table 18: Center Case uses Playstyle Jaccard Index (mix): consistent count = 1

| Speed70N2 | Speed60 | Speed65 | Speed70 | Speed75 | Speed80 |
|---|---|---|---|---|---|
| N0 | 0.0823 (0.0046) | 0.0829 (0.0047) | 0.0827 (0.0046) | 0.0844 (0.0043) | 0.0831 (0.0051) |
| N1 | 0.0836 (0.0050) | 0.0820 (0.0048) | 0.0846 (0.0043) | 0.0850 (0.0045) | 0.0855 (0.0050) |
| N2 (C) | 0.0821 (0.0050) | 0.0825 (0.0043) | 0.0930 (0.0049) | 0.0845 (0.0042) | 0.0842 (0.0045) |
| N3 | 0.0806 (0.0042) | 0.0827 (0.0044) | 0.0795 (0.0049) | 0.0839 (0.0043) | 0.0780 (0.0045) |
| N4 | 0.0802 (0.0046) | 0.0792 (0.0046) | 0.0799 (0.0048) | 0.0782 (0.0047) | 0.0804 (0.0050) |

Table 19: Center Case uses Playstyle Similarity (mix): consistent count = 3

| Speed70N2 | Speed60 | Speed65 | Speed70 | Speed75 | Speed80 |
|---|---|---|---|---|---|
| N0 (C) | 0.0554 (0.0040) | 0.0573 (0.0039) | 0.0579 (0.0043) | 0.0568 (0.0034) | 0.0522 (0.0037) |
| N1 (C) | 0.0527 (0.0036) | 0.0555 (0.0041) | 0.0560 (0.0035) | 0.0549 (0.0038) | 0.0518 (0.0037) |
| N2 (C) | 0.0486 (0.0038) | 0.0522 (0.004) | 0.0652 (0.0043) | 0.0512 (0.0035) | 0.0498 (0.0035) |
| N3 | 0.0486 (0.0033) | 0.0487 (0.0035) | 0.0479 (0.0037) | 0.0485 (0.0038) | 0.0438 (0.0030) |
| N4 | 0.0417 (0.0033) | 0.0452 (0.0033) | 0.0444 (0.0037) | 0.0407 (0.0035) | 0.0399 (0.0035) |

Table 20: Averaged diverse trajectory count of different DRL algorithms across 7 Atari games in 25 episodes.

| DRL Algorithm | Asterix | Breakout | MsPacman | Pong | Qbert | Seaquest | SpaceInvaders |
|---|---|---|---|---|---|---|---|
| DQN | 6.00 | 6.00 | 5.33 | 4.00 | 6.00 | 11.00 | 25.00 |
| C51 | 6.00 | 7.00 | 7.00 | 6.00 | 7.00 | 21.00 | 25.00 |
| Rainbow | 8.67 | 5.33 | 8.00 | 5.00 | 5.00 | 24.00 | 25.00 |
| IQN | **25.00** | **14.00** | **10.00** | **9.00** | **10.00** | **25.00** | 25.00 |

## A.5 More Experiments on Diversity Measurement in DRL

This section introduces additional experiments demonstrating the application of our method for quantifying decision-making diversity, detailed in Algorithm 1. We analyze the Atari DRL agent dataset from the main paper, which includes 5 models per DRL algorithm, each contributing 5 trajectories, totaling 25 trajectories per algorithm (Mnih et al., 2015; Bellemare et al., 2017; Hessel et al., 2018; Dabney et al., 2018). Using Algorithm 1, we assess the diversity of trajectories produced by each algorithm. Results in Table 20, averaged across three discrete encoder models with a similarity threshold of $t = 0.2$, show the capacity of models trained under the same DRL algorithm to generate diverse game episodes within 25 attempts. The IQN algorithm displays higher diversity across games, consistent with its risk sampling feature.

We also apply various levels of stochasticity using the Dopamine framework to illustrate another use case of this measure. Our focus is on the first IQN model from Dopamine, which is expected to adapt to a wide array of playstyles due to its risk functions and robust performance capabilities. To encourage diversity, we employ the Boltzmann distribution, commonly known as softmax, varying the temperature ($z$) to influence decision-making:

$$\pi(s) = \text{Softmax}\left(\frac{A(s)}{z}\right) \tag{20}$$

This approach, inspired by Fan & Xiao (2022), uses the advantage function ($A$) crucial in action selection in reinforcement learning, where a higher advantage indicates a preferable action.

We explore four levels of randomness ($z \in \{0.0001, 0.001, 0.01, 0.1\}$), anticipating increased diversity with greater randomness. We test similarity thresholds ($t$) of 0.5, 0.2, and 0.05 across seven Atari games with 100 trajectories each. Results are illustrated through shaded curves in Figures from Figure 14 to Figure 16, demonstrating the efficacy of our diversity measure.

Notably, games like *Seaquest* (Figure 17b) exhibit high diversity even at lower randomness levels, indicating intrinsic complexity in terms of playstyles. In contrast, *Qbert* (Figure 17a) becomes more monotonous since the goal is to achieve a higher score in the puzzle game. This observation suggests another application for our measure: identifying the complexity of game content. The time complexity of Algorithm 1 is $O(N^2)$, given the number of trajectories $N$. Future research could investigate more efficient methods, perhaps leveraging approximations or advanced data structures for quicker similarity checks.

The algorithm we introduce for diversity quantification shifts our understanding of gaming from the subjective to the quantitative. While various methodologies for measuring diversity exist in different domains, our approach is particularly apt to video game playing. In addition, recognizing and quantifying this diversity can inform the development of more adaptive DRL models, thereby addressing specific challenges in gaming and artificial intelligence. This new measure contributes to our progress toward models that are not only efficient but also demonstrate a variety of adaptable strategies, opening up vast avenues for future research.

## A.6 Game 2048 Experiment Details

We employed temporal difference learning to train our 2048 agents, utilizing training code available on GitHub[2]. We set the learning rate ($\alpha$) to 0.01 and maintained all other default settings. As this study does

---

[2]https://github.com/moporgic/TDL2048-Demo

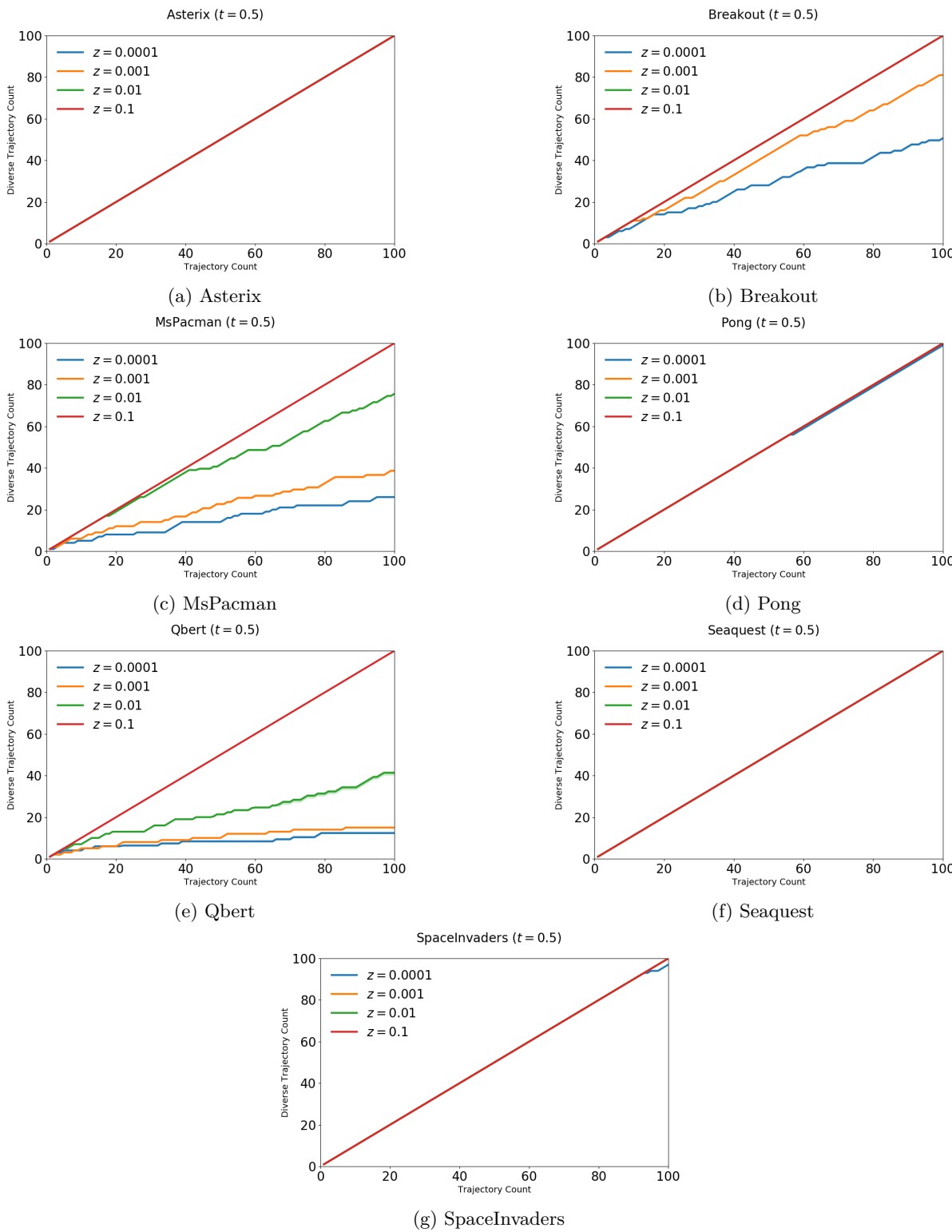

Figure 14: Diversity Measurement of IQN Models in Seven Atari Games ($t = 0.5$). The shaded area indicates the range between min and max accuracy among three encoder models.

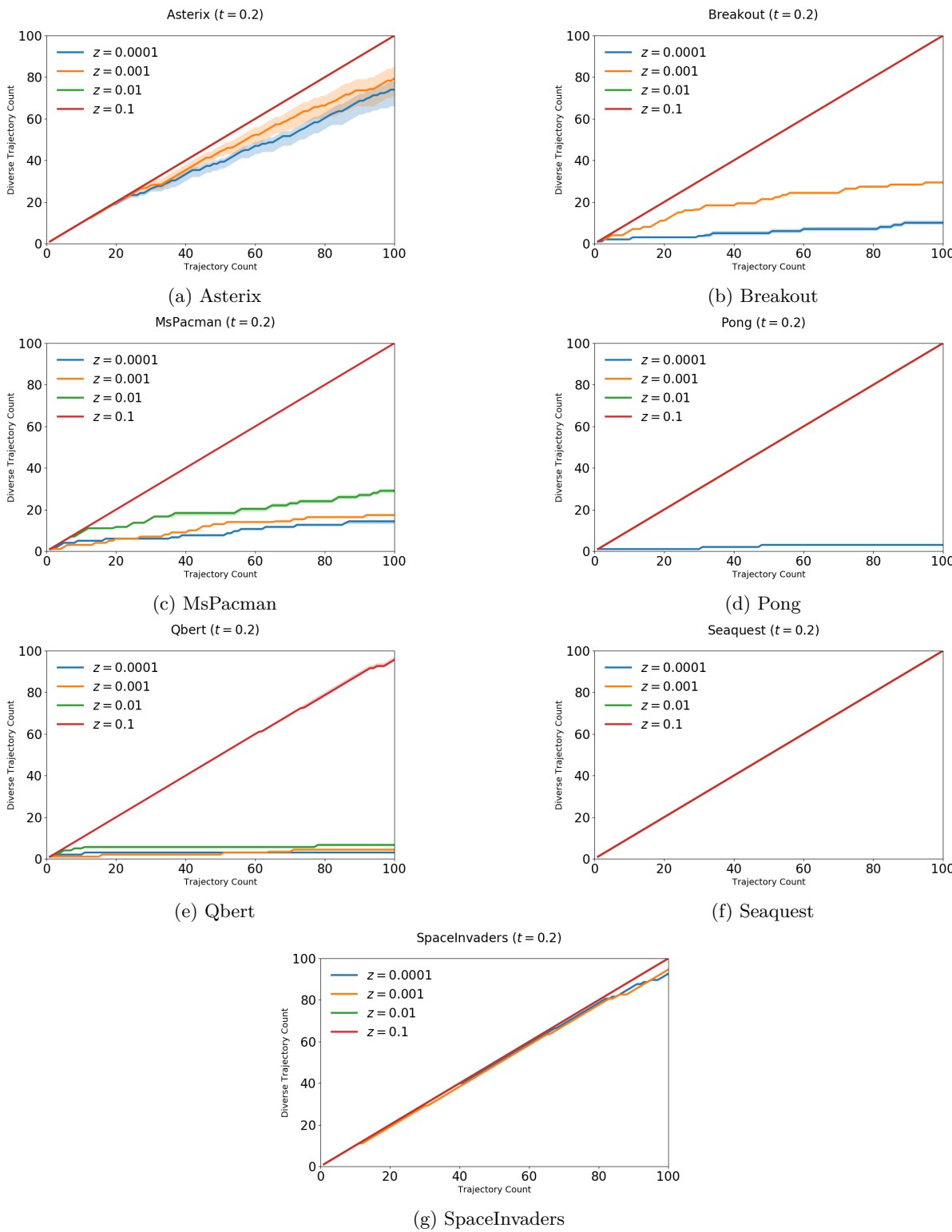

Figure 15: Diversity Measurement of IQN Models in Seven Atari Games ($t = 0.2$). The shaded area indicates the range between min and max accuracy among three encoder models.

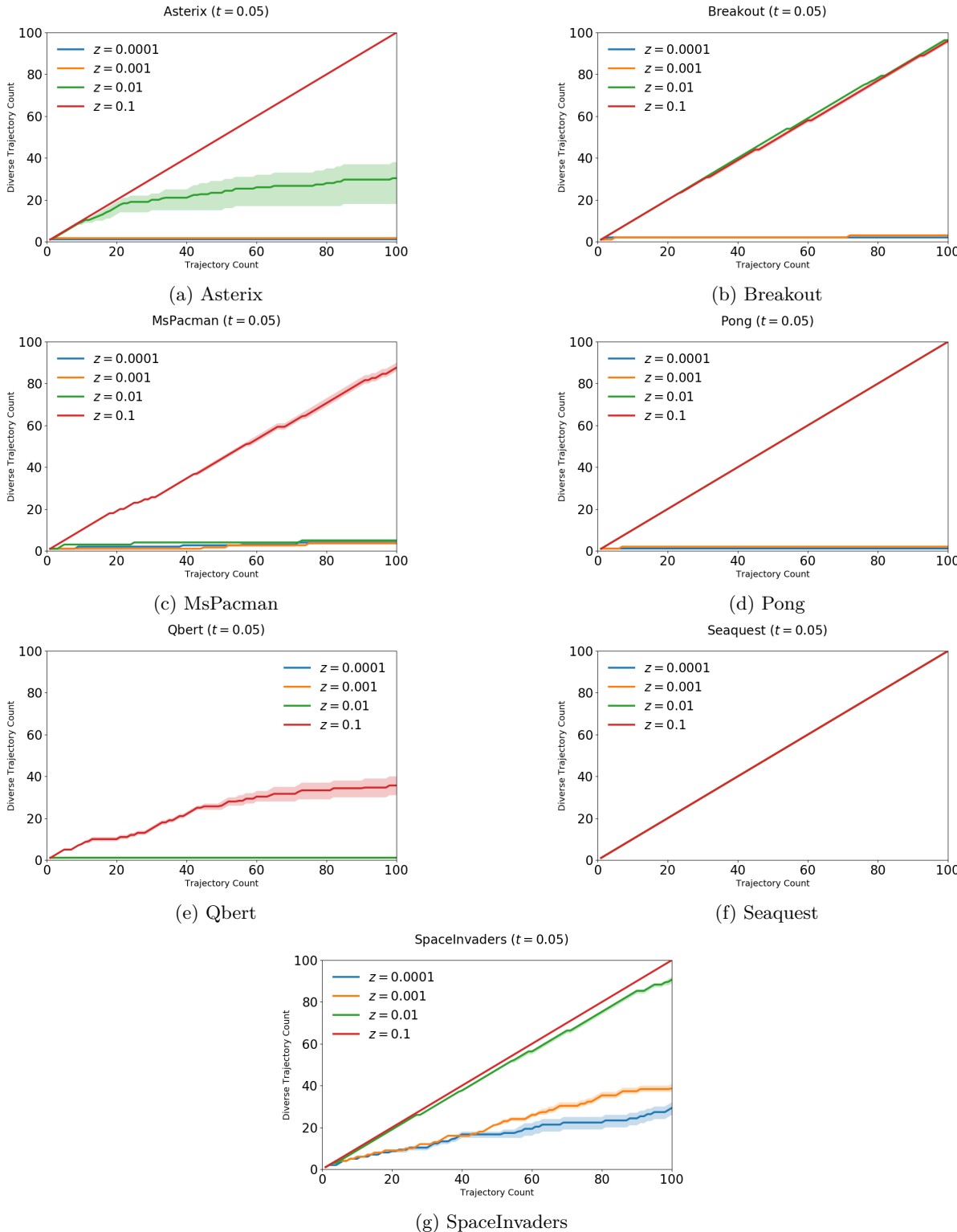

Figure 16: Diversity Measurement of IQN Models in Seven Atari Games ($t = 0.05$). The shaded area indicates the range between min and max accuracy among three encoder models.

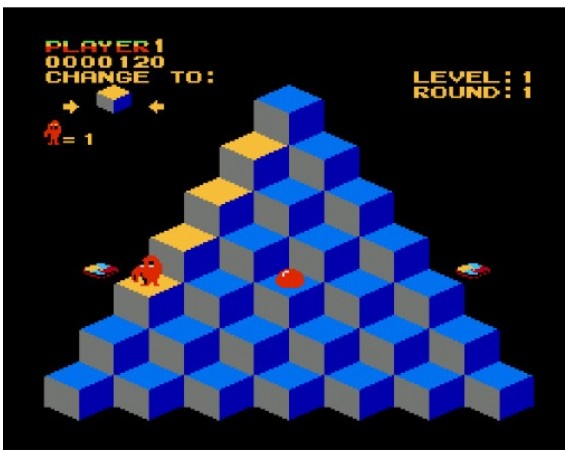

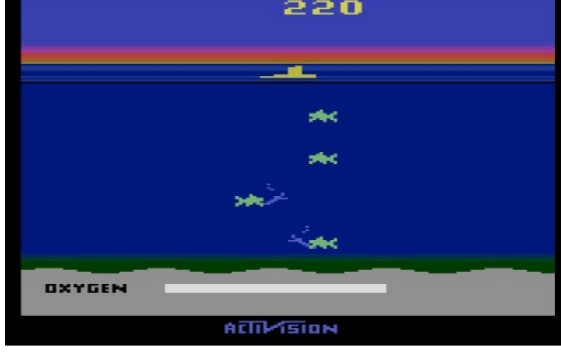

(a) Qbert: A 2D puzzle game where your goal is to change every cube in a pyramid to a target color. To do this, control the on-screen character, Q*bert, and make it jumps on top of the cube, avoiding obstacles and enemies.

(b) Seaquest: A 2D survival shooting game. The player sails a submarine to shoot at sharks and enemy submarines to rescue divers swimming in the water.

Figure 17: Game screens of Qbert and Seaquest.

not focus on optimizing discrete representations, we directly use the full board as the discrete states for playstyle analysis. The potential state space for the $4 \times 4$ version of 2048 is calculated to be $16^{16}$.

## A.7  Go Experiment Details

Our Go experiment follows the standard 19x19 rules with common observations, including the latest eight board views and player color, resulting in 18 channels (Silver et al., 2018; Wu et al., 2024). The action space includes 362 discrete actions (361 move actions and a pass).

Figure 18 shows the variant of HSD we used. The primary difference from the implementation in Playstyle Distance is that the reconstruction head for the autoencoder is replaced by a value head. Additionally, we use three levels of hierarchy with different numbers of embedding candidates: 256 candidate embeddings at the base level, 16 candidate embeddings at the next level, and 4 candidate embeddings at the final level. The smallest state space is $4^8 = 65536$, which ensures sufficient intersection states even with only a few game records. The datasets used for training the encoder and playstyle identification are private datasets provided by the MiniZero framework team (Wu et al., 2024). If you need these datasets to reproduce our results on Go, please contact them directly for authorization.

We train the encoder with a batch size of 1024 over 100 iterations, each iteration including 1000 network updates with the Adam optimizer. The learning rate starts at 0.00025 and linearly decays to 0 according to the iteration number. The coefficient $\beta$ in the vector quantization process is the commonly suggested 0.25 (van den Oord et al., 2017; Lin et al., 2021). The loss function for the policy head is cross-entropy, and the loss for the value head is mean square error, with the loss coefficients of these two heads both set to 1.

Tables 21-29 list the detailed accuracies for different query and candidate sizes using only the first 10 moves of games and using the full game moves as mentioned in Section 6.2. For the first 10 moves case, it is common and straightforward in board games that a preferred opening is a kind of playstyle and using a small state space can achieve 97.0% accuracy ($\{4^8\}$, Table 21). However, when we do not specifically focus on the playstyles in the opening, we may want to use all game moves to capture any possible playstyles, and

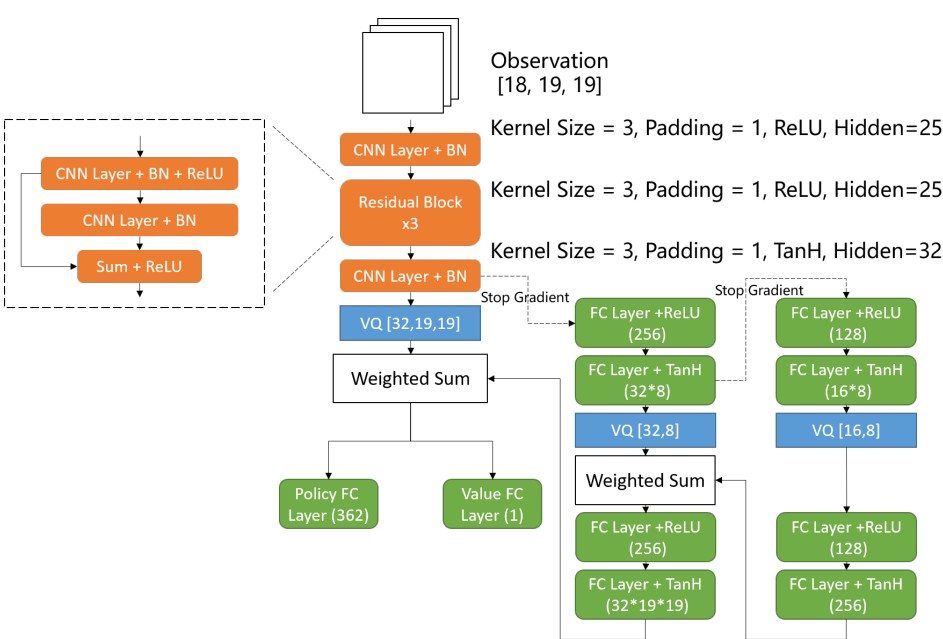

Figure 18: The neural network architecture of the HSD encoder for Go.

Table 21: Accuracy of 200 human Go player identification with the Playstyle Distance ($4^8$).

| First 10 Moves | Query 1 | Query 5 | Query 10 | Query 25 | Query 50 | Query 75 | Query 100 |
|---|---|---|---|---|---|---|---|
| Candidate 1 | 1.5% | 2.5% | 3.0% | 4.0% | 4.0% | 7.5% | 8.5% |
| Candidate 5 | 3.0% | 11.5% | 20.5% | 29.0% | 40.0% | 51.0% | 50.0% |
| Candidate 10 | 7.5% | 24.5% | 33.5% | 49.0% | 57.0% | 64.5% | 72.0% |
| Candidate 25 | 10.5% | 32.0% | 49.5% | 71.0% | 79.0% | 87.5% | 89.5% |
| Candidate 50 | 17.5% | 44.0% | 59.5% | 76.0% | 87.0% | 92.0% | 95.5% |
| Candidate 75 | 17.5% | 48.5% | 65.0% | 80.5% | 88.5% | 93.0% | 96.5% |
| Candidate 100 | 19.5% | 48.5% | 66.5% | 81.0% | 89.0% | 94.0% | 97.0% |
| **Full Game Moves** | Query 1 | Query 5 | Query 10 | Query 25 | Query 50 | Query 75 | Query 100 |
| Candidate 1 | 2.5% | 3.0% | 5.5% | 3.0% | 5.5% | 3.5% | 3.0% |
| Candidate 5 | 5.0% | 5.5% | 8.0% | 6.0% | 10.0% | 11.5% | 10.0% |
| Candidate 10 | 7.0% | 9.5% | 11.5% | 10.5% | 11.0% | 11.0% | 10.5% |
| Candidate 25 | 9.5% | 13.5% | 16.0% | 20.0% | 20.5% | 20.5% | 21.5% |
| Candidate 50 | 10.0% | 22.5% | 30.0% | 32.0% | 31.0% | 31.5% | 33.0% |
| Candidate 75 | 13.5% | 30.0% | 38.0% | 42.0% | 43.0% | 39.5% | 39.5% |
| Candidate 100 | 16.5% | 33.0% | 40.0% | 47.0% | 50.0% | 46.0% | 50.0% |

such a small state space negatively impacts the accuracy. Some boards cannot be compared in this kind of playstyle but the small state space cannot provide information to separate these cases. Instead, large state spaces like $\{16^8\}$ or $\{256^{361}\}$ can handle this kind of problem. If we do not know which state space can give the best result, our multiscale state space is a good choice that leverages all state spaces. Even if it may be influenced by a very bad state space like $\{4^8\}$ in the full games case, the damage is limited. Also, discrete playstyles incorporating the Jaccard index (Playstyle Similarity and its variants) can further cover this weakness. Additionally, the Bhattacharyya coefficient is more effective when a playstyle plays very diversely in a state, and our Go result is an example. For slightly different playstyles like TORCS with continuous actions, using the scaled W2 metric can provide better accuracy.

Table 22: Accuracy of 200 human Go player identification with the Playstyle Distance ($16^8$).

| First 10 Moves | Query 1 | Query 5 | Query 10 | Query 25 | Query 50 | Query 75 | Query 100 |
|---|---|---|---|---|---|---|---|
| Candidate 1 | 2.0% | 3.5% | 4.0% | 5.0% | 3.0% | 5.5% | 5.5% |
| Candidate 5 | 4.5% | 10.0% | 13.5% | 27.5% | 38.0% | 47.0% | 52.5% |
| Candidate 10 | 5.0% | 23.0% | 41.5% | 62.5% | 65.0% | 72.5% | 79.0% |
| Candidate 25 | 11.5% | 43.0% | 58.5% | 74.0% | 83.5% | 86.5% | 90.5% |
| Candidate 50 | 18.0% | 48.0% | 66.0% | 78.5% | 85.0% | 91.5% | 94.0% |
| Candidate 75 | 21.0% | 54.5% | 67.0% | 81.5% | 90.0% | 92.5% | 95.0% |
| Candidate 100 | 20.5% | 53.5% | 69.5% | 82.0% | 91.0% | 93.0% | 95.5% |
| **Full Game Moves** | Query 1 | Query 5 | Query 10 | Query 25 | Query 50 | Query 75 | Query 100 |
| Candidate 1 | 2.0% | 3.5% | 4.0% | 5.5% | 4.0% | 5.5% | 5.0% |
| Candidate 5 | 4.5% | 10.0% | 14.0% | 28.5% | 36.5% | 46.5% | 49.5% |
| Candidate 10 | 5.0% | 23.5% | 44.5% | 61.0% | 67.5% | 71.0% | 73.5% |
| Candidate 25 | 11.5% | 43.0% | 59.0% | 74.0% | 80.0% | 84.5% | 88.5% |
| Candidate 50 | 18.0% | 49.5% | 66.0% | 79.0% | 86.0% | 91.5% | 94.5% |
| Candidate 75 | 20.0% | 53.0% | 70.0% | 82.0% | 91.5% | 93.5% | 95.5% |
| Candidate 100 | 20.0% | 53.0% | 72.0% | 83.5% | 91.0% | 95.5% | 96.5% |

Table 23: Accuracy of 200 human Go player identification with the Playstyle Distance ($256^{361}$).

| First 10 Moves | Query 1 | Query 5 | Query 10 | Query 25 | Query 50 | Query 75 | Query 100 |
|---|---|---|---|---|---|---|---|
| Candidate 1 | 2.0% | 4.5% | 5.5% | 5.5% | 5.0% | 6.0% | 6.0% |
| Candidate 5 | 5.0% | 8.0% | 10.0% | 23.5% | 35.0% | 42.5% | 50.0% |
| Candidate 10 | 5.5% | 24.5% | 38.0% | 61.5% | 65.0% | 72.5% | 76.0% |
| Candidate 25 | 11.5% | 43.0% | 58.5% | 75.0% | 80.0% | 87.5% | 90.0% |
| Candidate 50 | 17.0% | 50.0% | 63.5% | 80.0% | 87.0% | 91.5% | 93.5% |
| Candidate 75 | 20.0% | 54.5% | 68.0% | 83.0% | 89.0% | 92.0% | 95.0% |
| Candidate 100 | 20.5% | 55.0% | 67.0% | 82.0% | 89.5% | 92.0% | 94.5% |
| **Full Game Moves** | Query 1 | Query 5 | Query 10 | Query 25 | Query 50 | Query 75 | Query 100 |
| Candidate 1 | 2.0% | 4.5% | 5.5% | 5.5% | 5.0% | 6.0% | 5.5% |
| Candidate 5 | 5.0% | 8.0% | 10.0% | 23.5% | 35.5% | 42.5% | 49.5% |
| Candidate 10 | 5.5% | 25.0% | 39.5% | 62.0% | 65.0% | 73.0% | 76.5% |
| Candidate 25 | 11.5% | 44.0% | 60.0% | 75.5% | 80.5% | 88.5% | 90.5% |
| Candidate 50 | 17.0% | 50.5% | 65.0% | 81.5% | 88.5% | 93.0% | 94.5% |
| Candidate 75 | 20.0% | 56.5% | 70.0% | 84.5% | 90.5% | 93.5% | 96.5% |
| Candidate 100 | 21.0% | 57.0% | 69.5% | 84.0% | 91.0% | 94.0% | 96.5% |

Table 24: Accuracy of 200 human Go player identification with the Playstyle Distance (mix).

| First 10 Moves | Query 1 | Query 5 | Query 10 | Query 25 | Query 50 | Query 75 | Query 100 |
|---|---|---|---|---|---|---|---|
| Candidate 1 | 1.5% | 3.0% | 2.5% | 5.5% | 6.5% | 7.5% | 9.5% |
| Candidate 5 | 4.5% | 14.0% | 27.5% | 42.0% | 50.0% | 59.0% | 65.5% |
| Candidate 10 | 7.5% | 30.5% | 46.5% | 59.5% | 69.0% | 77.5% | 79.5% |
| Candidate 25 | 16.5% | 45.0% | 62.0% | 75.5% | 86.5% | 91.5% | 93.0% |
| Candidate 50 | 20.0% | 49.5% | 70.0% | 80.5% | 87.5% | 92.5% | 96.5% |
| Candidate 75 | 21.0% | 54.5% | 72.5% | 84.5% | 90.5% | 94.5% | 97.0% |
| Candidate 100 | 25.0% | 57.5% | 72.5% | 85.0% | 91.5% | 93.5% | 96.5% |
| **Full Game Moves** | Query 1 | Query 5 | Query 10 | Query 25 | Query 50 | Query 75 | Query 100 |
| Candidate 1 | 2.0% | 3.5% | 7.5% | 7.5% | 11.0% | 10.5% | 10.0% |
| Candidate 5 | 6.5% | 17.0% | 20.5% | 30.5% | 34.0% | 41.0% | 41.0% |
| Candidate 10 | 11.0% | 24.5% | 33.0% | 39.0% | 43.0% | 49.0% | 53.0% |
| Candidate 25 | 13.5% | 36.5% | 46.0% | 56.5% | 62.0% | 63.0% | 69.5% |
| Candidate 50 | 17.0% | 45.0% | 56.0% | 70.5% | 76.0% | 77.5% | 81.0% |
| Candidate 75 | 20.0% | 49.0% | 65.0% | 75.5% | 79.5% | 83.0% | 86.5% |
| Candidate 100 | 22.0% | 53.0% | 65.5% | 75.0% | 81.0% | 85.0% | 90.0% |

Table 25: Accuracy of 200 human Go player identification with the Playstyle Intersection Similarity (mix).

| First 10 Moves | Query 1 | Query 5 | Query 10 | Query 25 | Query 50 | Query 75 | Query 100 |
|---|---|---|---|---|---|---|---|
| Candidate 1 | 2.5% | 3.0% | 3.5% | 5.5% | 5.5% | 8.0% | 1.15% |
| Candidate 5 | 5.0% | 15.0% | 28.5% | 39.0% | 46.5% | 54.0% | 57.5% |
| Candidate 10 | 8.5% | 29.0% | 42.0% | 54.0% | 60.0% | 71.0% | 76.5% |
| Candidate 25 | 14.5% | 41.5% | 56.0% | 68.5% | 75.0% | 85.5% | 91.5% |
| Candidate 50 | 18.0% | 48.0% | 63.0% | 75.5% | 84.0% | 89.0% | 95.0% |
| Candidate 75 | 21.5% | 51.5% | 65.0% | 79.0% | 88.0% | 94.5% | 95.0% |
| Candidate 100 | 23.5% | 56.5% | 66.0% | 80.5% | 89.0% | 93.0% | 94.5% |
| **Full Game Moves** | Query 1 | Query 5 | Query 10 | Query 25 | Query 50 | Query 75 | Query 100 |
| Candidate 1 | 3.0% | 3.0% | 7.0% | 7.5% | 13.0% | 16.0% | 20.0% |
| Candidate 5 | 7.0% | 19.0% | 27.0% | 33.0% | 44.0% | 52.0% | 55.5% |
| Candidate 10 | 12.5% | 27.0% | 38.0% | 46.0% | 56.5% | 66.0% | 72.0% |
| Candidate 25 | 19.0% | 40.0% | 53.5% | 66.0% | 80.0% | 84.5% | 85.5% |
| Candidate 50 | 18.5% | 47.0% | 61.0% | 76.0% | 86.0% | 90.5% | 94.5% |
| Candidate 75 | 21.0% | 50.5% | 64.0% | 79.5% | 91.0% | 95.0% | 96.0% |
| Candidate 100 | 26.5% | 55.5% | 67.5% | 81.0% | 92.0% | 93.5% | 95.0% |

Table 26: Accuracy of 200 human Go player identification with the Playstyle Int BC Similarity (mix).

| First 10 Moves | Query 1 | Query 5 | Query 10 | Query 25 | Query 50 | Query 75 | Query 100 |
|---|---|---|---|---|---|---|---|
| Candidate 1 | 2.5% | 2.5% | 3.0% | 6.5% | 6.5% | 9.5% | 12.0% |
| Candidate 5 | 5.5% | 14.0% | 26.0% | 40.5% | 48.5% | 58.5% | 61.5% |
| Candidate 10 | 9.0% | 30.0% | 42.5% | 53.5% | 66.5% | 76.0% | 78.0% |
| Candidate 25 | 16.0% | 46.0% | 60.5% | 72.0% | 81.5% | 89.5% | 93.5% |
| Candidate 50 | 19.5% | 55.0% | 67.5% | 81.0% | 89.0% | 93.0% | 95.5% |
| Candidate 75 | 23.0% | 57.5% | 72.5% | 85.0% | 91.5% | 96.5% | 96.0% |
| Candidate 100 | 28.5% | 61.0% | 76.5% | 86.0% | 92.5% | 94.5% | 95.5% |
| **Full Game Moves** | Query 1 | Query 5 | Query 10 | Query 25 | Query 50 | Query 75 | Query 100 |
| Candidate 1 | 2.5% | 4.0% | 7.5% | 8.5% | 14.0% | 15.0% | 18.5% |
| Candidate 5 | 7.5% | 16.5% | 26.0% | 34.5% | 41.5% | 49.0% | 51.0% |
| Candidate 10 | 12.5% | 27.0% | 37.5% | 45.5% | 58.0% | 64.0% | 70.5% |
| Candidate 25 | 17.0% | 39.5% | 55.0% | 66.0% | 79.5% | 83.5% | 87.5% |
| Candidate 50 | 19.5% | 48.5% | 65.0% | 78.0% | 90.0% | 92.5% | 95.0% |
| Candidate 75 | 24.5% | 60.5% | 72.5% | 83.0% | 92.5% | 96.0% | 97.5% |
| Candidate 100 | 30.0% | 63.0% | 76.5% | 85.0% | 94.5% | 95.5% | 97.0% |

Table 27: Accuracy of 200 human Go player identification with the Playstyle Jaccard Index (mix).

| First 10 Moves | Query 1 | Query 5 | Query 10 | Query 25 | Query 50 | Query 75 | Query 100 |
|---|---|---|---|---|---|---|---|
| Candidate 1 | 3.0% | 7.5% | 9.5% | 11.0% | 14.5% | 16.0% | 19.0% |
| Candidate 5 | 7.5% | 9.5% | 15.0% | 20.5% | 25.0% | 29.0% | 32.5% |
| Candidate 10 | 11.5% | 13.5% | 21.5% | 35.0% | 42.5% | 48.0% | 48.5% |
| Candidate 25 | 14.5% | 28.0% | 37.0% | 49.0% | 67.0% | 72.0% | 76.5% |
| Candidate 50 | 17.0% | 34.5% | 42.5% | 60.5% | 78.0% | 80.5% | 86.0% |
| Candidate 75 | 18.5% | 40.5% | 53.0% | 69.0% | 80.5% | 88.0% | 90.0% |
| Candidate 100 | 22.0% | 46.0% | 53.5% | 72.0% | 83.5% | 90.5% | 95.0% |
| **Full Game Moves** | Query 1 | Query 5 | Query 10 | Query 25 | Query 50 | Query 75 | Query 100 |
| Candidate 1 | 3.0% | 4.0% | 3.0% | 3.5% | 3.5% | 2.5% | 3.0% |
| Candidate 5 | 3.0% | 5.0% | 3.0% | 5.0% | 6.0% | 7.0% | 10.0% |
| Candidate 10 | 4.0% | 6.5% | 8.5% | 10.5% | 15.5% | 12.0% | 14.0% |
| Candidate 25 | 9.0% | 12.5% | 15.5% | 22.5% | 23.5% | 24.5% | 25.0% |
| Candidate 50 | 8.5% | 18.5% | 25.5% | 40.0% | 51.0% | 53.5% | 51.0% |
| Candidate 75 | 13.0% | 21.5% | 29.0% | 46.0% | 59.0% | 66.0% | 68.5% |
| Candidate 100 | 16.0% | 23.0% | 30.0% | 45.0% | 65.5% | 72.0% | 80.5% |

Table 28: Accuracy of 200 human Go player identification with the Playstyle Similarity (mix).

| First 10 Moves | Query 1 | Query 5 | Query 10 | Query 25 | Query 50 | Query 75 | Query 100 |
|---|---|---|---|---|---|---|---|
| Candidate 1 | 3.0% | 9.0% | 12.5% | 18.0% | 21.0% | 24.5% | 28.5% |
| Candidate 5 | 11.0% | 21.5% | 28.0% | 38.0% | 49.0% | 54.0% | 55.5% |
| Candidate 10 | 16.0% | 28.5% | 40.0% | 52.0% | 62.0% | 66.0% | 70.0% |
| Candidate 25 | 23.0% | 42.0% | 58.5% | 65.0% | 81.5% | 86.0% | 89.0% |
| Candidate 50 | 24.0% | 48.5% | 61.0% | 73.5% | 87.5% | 89.5% | 92.5% |
| Candidate 75 | 27.0% | 57.0% | 68.5% | 80.0% | 88.5% | 94.0% | 95.5% |
| Candidate 100 | 30.0% | 55.5% | 69.5% | 79.5% | 90.0% | 95.0% | 97.0% |
| **Full Game Moves** | Query 1 | Query 5 | Query 10 | Query 25 | Query 50 | Query 75 | Query 100 |
| Candidate 1 | 4.5% | 9.0% | 11.5% | 11.5% | 10.5% | 13.5% | 10.0% |
| Candidate 5 | 8.0% | 15.5% | 17.5% | 21.0% | 28.5% | 29.5% | 31.0% |
| Candidate 10 | 12.0% | 23.0% | 29.5% | 39.0% | 44.5% | 44.0% | 46.0% |
| Candidate 25 | 16.0% | 31.5% | 44.5% | 56.5% | 66.5% | 69.5% | 72.0% |
| Candidate 50 | 19.0% | 38.5% | 58.5% | 73.0% | 81.5% | 83.5% | 85.0% |
| Candidate 75 | 23.5% | 46.5% | 56.5% | 76.5% | 87.0% | 90.5% | 93.0% |
| Candidate 100 | 27.0% | 46.5% | 55.5% | 77.5% | 88.5% | 93.5% | 94.0% |

Table 29: Accuracy of 200 human Go player identification with the Playstyle BC Similarity (mix).

| First 10 Moves | Query 1 | Query 5 | Query 10 | Query 25 | Query 50 | Query 75 | Query 100 |
|---|---|---|---|---|---|---|---|
| Candidate 1 | 3.0% | 9.0% | 13.0% | 20.0% | 23.5% | 28.0% | 33.0% |
| Candidate 5 | 12.0% | 24.0% | 36.0% | 43.0% | 49.5% | 57.0% | 58.5% |
| Candidate 10 | 18.5% | 32.5% | 45.5% | 55.0% | 66.0% | 72.0% | 75.0% |
| Candidate 25 | 22.0% | 45.5% | 63.5% | 70.5% | 83.0% | 91.0% | 93.0% |
| Candidate 50 | 27.0% | 55.5% | 68.0% | 76.0% | 88.0% | 92.5% | 95.5% |
| Candidate 75 | 29.5% | 63.5% | 74.5% | 83.5% | 90.0% | 96.5% | 97.0% |
| Candidate 100 | 31.5% | 64.5% | 74.5% | 83.0% | 92.0% | 95.5% | 97.0% |
| **Full Game Moves** | Query 1 | Query 5 | Query 10 | Query 25 | Query 50 | Query 75 | Query 100 |
| Candidate 1 | 4.0% | 9.5% | 13.5% | 21.0% | 26.0% | 29.0% | 33.5% |
| Candidate 5 | 13.5% | 22.0% | 33.0% | 42.0% | 54.0% | 57.5% | 61.5% |
| Candidate 10 | 18.5% | 32.0% | 45.5% | 54.5% | 66.5% | 73.5% | 76.5% |
| Candidate 25 | 23.0% | 46.5% | 61.0% | 73.0% | 83.5% | 91.5% | 92.5% |
| Candidate 50 | 28.0% | 54.5% | 70.5% | 79.5% | 92.0% | 94.5% | 96.0% |
| Candidate 75 | 30.0% | 62.5% | 74.5% | 85.5% | 93.0% | 97.5% | 97.5% |
| Candidate 100 | 35.0% | 62.0% | 78.0% | 87.5% | 95.0% | 97.0% | 97.5% |

### A.8  Statistical Significance of Experiment Results

In this appendix, we present the statistical tests conducted to validate some of the claims made in our main paper. While accuracy is a common metric for evaluating the performance of classification models in machine learning, it is essential to establish that the observed differences are not due to sampling uncertainty. To this end, we employed McNemar's test (McNemar, 1947) for statistical significance. This non-parametric test is particularly suitable for our paired classification scenarios with binary outcomes—correct or wrong predictions—allowing us to compare whether the prediction correctness of two measures is statistically significant.

We applied McNemar's test to several comparisons in our study:

1. Playstyle Distance with multiscale state space can have better accuracy than using any state space alone on the TORCS playstyle dataset.

2. When using the same state space ($2^{20}$) on intersection samples, probabilistic similarity is not worse than distance similarity on RGSK and slightly better on TORCS.

3. Jaccard index can hurt accuracy in some extreme cases, such as in 2048 with the raw board as discrete states.

4. For Go playstyle with full game moves, Playstyle BC Similarity has better accuracy than Playstyle Similarity.

The following subsections provide detailed results of these tests, including the p-values, which indicate whether the observed improvements are statistically significant. Each subsection corresponds to a specific comparison, outlining the methodology and presenting the findings. For defining statistical significance, we refer to a p-value $< 0.05$ as the default threshold suggested by Fisher (1970), since there is no standard p-value threshold for classification tasks in machine learning.

#### A.8.1  Multiscale State Space on TORCS

In this subsection, we examine the performance of the multiscale state space in improving accuracy on the TORCS playstyle dataset. We conducted McNemar's test to determine if the multiscale approach offers statistically significant improvements over using individual state spaces alone. The corresponding section in the main paper is Section 5.1.

Since we use random subsampling in our experiments, we reran another 100 rounds of random subsampling, recording the contingency table of different state space settings with the same samples. In other words, the dataset sampled in each subsampling round was used in all state space settings rather than resampling for different settings. The accuracies are slightly different from those in Table 2, but the effectiveness of improvement with the multiscale state space remains consistent in Table 30. The mix version (multiscale) has better accuracy under the first two encoders and is slightly worse than $2^{20}$ $t$=2 with a p-value of 0.179. However, using the same $t$=2 on the multiscale state space still shows a clear accuracy improvement. Overall, using $t = 1$ or $t = 2$ is not much different in TORCS; thus, a simpler setting with $t = 1$ (no filtering) for filtering intersection states is what we recommend.

#### A.8.2  Using Perceptual Similarity as an Alternative to Distance Similarity

Here, we investigate the efficacy of using perceptual similarity as an alternative to distance-based similarity measures. We applied McNemar's test to compare the performance of these two approaches on the TORCS and RGSK datasets, particularly focusing on whether probabilistic similarity can offer equal or better performance. The corresponding section in the main paper is Section 5.2.

We reran 100 rounds of random subsampling with a dataset size of 1024 and used the same samples for comparing two variants of probabilistic similarity. The results in Table 31 show clear improvements in accuracy over Playstyle Distance with both Playstyle Intersection Similarity and Playstyle Inter BC Similarity.

Table 30: Statistical test on multiscale state space in TORCS.

| Discrete State Space | p-value | Accuracy Change |
|---|---|---|
| $1 \rightarrow$ **mix** $t=1$ | | |
| Encoder1 | 1.947e-142 | $37.3\% \rightarrow 73.8\%$ |
| Encoder2 | 1.021e-191 | $34.4\% \rightarrow 78.0\%$ |
| Encoder3 | 4.166e-176 | $36.6\% \rightarrow 77.9\%$ |
| $2^{20}$ $t=2 \rightarrow$ **mix** $t=1$ | | |
| Encoder1 | 0.002 | $70.7\% \rightarrow 73.8\%$ |
| Encoder2 | 4.876e-8 | $72.2\% \rightarrow 78.0\%$ |
| Encoder3 | 0.179 | $79.2\% \rightarrow 77.9\%$ |
| $2^{20}$ $t=1 \rightarrow$ **mix** $t=1$ | | |
| Encoder1 | 4.285e-34 | $62.8\% \rightarrow 73.8\%$ |
| Encoder2 | 3.227e-29 | $68.2\% \rightarrow 78.0\%$ |
| Encoder3 | 2.080e-36 | $67.4\% \rightarrow 77.9\%$ |
| $256^{\mathrm{res}}$ $t=2 \rightarrow$ **mix** $t=1$ | | |
| Encoder1 | <1e-323 | $8.3\% \rightarrow 73.8\%$ |
| Encoder2 | <1e-323 | $4.9\% \rightarrow 78.0\%$ |
| Encoder3 | <1e-323 | $4.4\% \rightarrow 77.9\%$ |
| $256^{\mathrm{res}}$ $t=1 \rightarrow$ **mix** $t=1$ | | |
| Encoder1 | 1.577e-46 | $54.9\% \rightarrow 73.8\%$ |
| Encoder2 | 5.728e-44 | $60.8\% \rightarrow 78.0\%$ |
| Encoder3 | 1.716e-16 | $67.2\% \rightarrow 77.9\%$ |
| **mix** $t=2 \rightarrow$ **mix** $t=1$ | | |
| Encoder1 | 0.966 | $73.8\% \rightarrow 73.8\%$ |
| Encoder2 | 0.318 | $77.0\% \rightarrow 78.0\%$ |
| Encoder3 | 1.360e-7 | $82.7\% \rightarrow 77.9\%$ |

Table 31: Statistical test on perceptual similarity in TORCS and RGSK.

| TORCS | p-value | Accuracy Change |
|---|---|---|
| Playstyle Distance → Playstyle Intersection Similarity | | |
| Encoder1 | 2.186e-53 | 62.4% → 79.1% |
| Encoder2 | 2.180e-46 | 69.4% → 84.3% |
| Encoder3 | 2.655e-70 | 67.0% → 85.1% |
| Playstyle Distance → Playstyle Inter BC Similarity | | |
| Encoder1 | 2.232e-137 | 62.4% → 92.8% |
| Encoder2 | 3.040e-28 | 69.4% → 82.3% |
| Encoder3 | 1.222e-91 | 67.0% → 90.7% |
| RGSK | p-value | Accuracy Change |
| Playstyle Distance → Playstyle Intersection Similarity | | |
| Encoder1 | 7.093e-23 | 92.5% → 98.0% |
| Encoder2 | 0.001 | 97.1% → 98.2% |
| Encoder3 | 3.974e-44 | 90.3% → 99.0% |
| Playstyle Distance → Playstyle Inter BC Similarity | | |
| Encoder1 | 3.589e-19 | 92.5% → 97.7% |
| Encoder2 | 9.411e-8 | 97.1% → 98.8% |
| Encoder3 | 5.327e-35 | 90.3% → 98.1% |

Table 32: Statistical test on the game 2048 model identification.

| | p-value | Accuracy Change |
|---|---|---|
| Playstyle Distance → Playstyle Intersection Similarity | 0.012 | 98.90% → 98.52% |
| Playstyle Distance → Playstyle Inter BC Similarity | 1.0 | 98.90% → 98.90% |
| Playstyle Distance → Playstyle Jaccard Index | < 1e-323 | 98.90% → 49.22% |
| Playstyle Distance → Playstyle Similarity | 1.072e-295 | 98.90% → 71.26% |
| Playstyle Distance → Playstyle BC Similarity | 1.072e-295 | 98.90% → 71.26% |

It is also clearer with tables than plots in Figure 4 that perceptual similarity can provide better accuracy on RGSK.

### A.8.3 The Potential Limitation of the Jaccard Index for Playstyle

In this part, we explore the potential limitations of using the Jaccard index in playstyle measurement. Specifically, we assess its impact on accuracy in the game 2048, using McNemar's test to highlight cases where the Jaccard index may negatively affect performance. The corresponding section in the main paper is Section 6.1.

The results in Table 32 show that playstyle measures including the Jaccard index have a clear accuracy reduction compared to the Playstyle Distance baseline, with the p-values demonstrating statistical significance.

### A.8.4 The Recommendation of Using Playstyle BC Similarity on Go

We conclude with an analysis of the Go playstyle dataset in Table 33, demonstrating the benefits of using Playstyle BC Similarity over Playstyle Similarity. McNemar's test was applied to confirm the statistical significance of the observed improvements in accuracy with the BC variant are not due to random sampling in the full game moves case. The corresponding section in the main paper is Section 6.2. With small samples, like using only one game record as a dataset, it is likely that these two measures have the same performance.

Table 33: Statistical test on the Go 200 player identification with M games as the query set and also M games as the candidate set. We compare the efficacy between Playstyle Similarity and Playstyle BC Similarity in the full game moves case.

| Playstyle Similarity (mix) $\rightarrow$ Playstyle BC Similarity (mix) | p-value | Accuracy Change |
|---|---|---|
| M=1 | 0.564 | $4.5\% \rightarrow 4.0\%$ |
| M=5 | 0.005 | $15.5\% \rightarrow 22.0\%$ |
| M=10 | 1.333e-5 | $29.5\% \rightarrow 45.5\%$ |
| M=25 | 2.553e-7 | $56.5\% \rightarrow 73.0\%$ |
| M=50 | 2.566e-4 | $81.5\% \rightarrow 92.0\%$ |
| M=75 | 0.002 | $90.5\% \rightarrow 97.5\%$ |
| M=100 | 0.052 | $94.0\% \rightarrow 97.5\%$ |

Conversely, when we have sufficient samples, like 100 games as a dataset, the p-value is slightly over 0.05, implying a 5.2% probability that Playstyle Similarity and Playstyle BC Similarity have nearly the same performance. In other sample sizes, we have enough confidence in claiming Playstyle BC Similarity has better performance than Playstyle Similarity due to the corresponding p-values being less than 0.05 and showing accuracy improvements.

