# OpenReview forum: "Perceptual Similarity for Measuring Decision-Making Style and Policy Diversity in Games"
_TMLR — Accepted by TMLR_

### Review · Reviewer_293a · 2024-06-10

**Summary Of Contributions:**

This submission covers a new metric for play style similarity based on observation and action sequences. The authors run a battery of experiments across three environments (TORCS, RGSK, and Atari), and demonstrate the ability to accurately identify players via matching reference observation and action sequences to withheld sequences. They then run additional tests in 2048 and Go, and propose a potential application for their metric.

**Audience:**

No

**Broader Impact Concerns:**

No broader impact concerns.

**Claims And Evidence:**

No

**Requested Changes:**

I do not feel that the current draft is in a state for acceptance. I would suggest the following changes:

1. Rework the introduction to focus on why studying similarity/difference in play styles is important and to remove the unsupported claims
2. Rework the related work to acknowledge existing similar metrics
3. Run a new evaluation with existing similar metrics as baselines that clearly identifies why the authors' proposed metric offers something distinct, either in terms of application areas or deeper understanding.

I believe these would be needed to represent sufficient contributions for acceptance, particularly 3.

**Strengths And Weaknesses:**

The primary strength of the paper is the large number of experiments demonstrating the results of the paper in terms of the authors' approach being able to accurately identify players within roughly 512 samples. This represents a large amount of work.

The paper has a number of weaknesses currently holding it back.

First, there is the lack of motivation for this work. The authors propose one possible application in terms of measuring policy diversity, and suggest others such as achieving human-like behaviour. However, there are no experiments run to determine if this new metric would have benefits to these application areas. The authors' claim that "understanding and analyzing playstyles remain a complex endeavor" but all of their examples are in terms of specific application areas, rather than why understanding and analyzing play styles itself is so complex. I would have preferred an argument about why we need to better understand similarity measures or why we need more accurate ones.

Second, there's a lack of engagement with prior work. Measuring player similarity or difference is a major part of the player modelling research field [1]. This area is focused on modelling human players, including finding similar and different players to a reference player. The types of metrics present in the paper, not over an embedded representation but directly on the game's internal state and action representation, are commonly found in this area [2,3].

Third, the evaluation setup is unclear. It's unclear what accuracy means here. I would guess that this is an accuracy calculated by finding the best match for some test sequence to some reference sequences, but this is not spelled out in the paper. Similarly, it's unclear if "Dataset Size" mentioned throughout the paper's graphs refers to the size of the reference sequences, the test sequences, or both. The "SL" and "CL" baselines are not described at all and appear to be random chance. The baselines in the paper that appear to be ablations of the authors' metric are also not clearly defined. I would suggest cutting the diversity algorithm to give the space to clarify these points.

Fourth, the results do not support the authors' claims. From my reading, the results seem to indicate that at low amounts of reference (or test?) data all metrics are the same and that at sufficiently large sizes they're the same. The benefit seems to be if one is in a situation where one only has access to datatsets of between 512 and 1024 samples. This does not to me seem like a "pivotal advance". They also do not "shed light on understanding similarity through the lens of human cognition", while human cognition is part of the inspiration for one part of the metric, there's no evaluation that supports this claim. I'm also not certain that this can count as zero-shot when the work relies on a learned or authored encoding for the domain. Overall, the results seem overstated.

1. Yannakakis, Georgios N., et al. "Player modeling." (2013).
2. Zhang, Xiang, Hans-Frederick Brown, and Anil Shankar. "Data-driven personas: Constructing archetypal users with clickstreams and user telemetry." Proceedings of the 2016 CHI conference on human factors in computing systems. 2016.
3. Rivera-Villicana, Jessica, et al. "Informing a BDI player model for an interactive narrative." Proceedings of the 2018 Annual Symposium on Computer-Human Interaction in Play. 2018.

---

> ### Author Response · Authors · 2024-06-10
>
> Thank you for your swift engagement.
>
> * **Motivation Concerns**: I noticed a possible contradiction between your comments on "Weaknesses" and "Requested Changes 1". The main issue regarding motivation seems to be the absence of direct experiments, such as those demonstrating human-like behavior, to validate the application of playstyle measures beyond our experiments on policy diversity for RL agents and the direct zero-shot classification task, rather than the significance of why playstyle is important.
>     * If "why playstyle is important" in practical terms is also a major concern, rather than requiring more application cases to support it, please revisit Section 1 (the latter part of "Yet, while DRL continues to show promise in diverse applications,...") in case anything was overlooked.
>         * To validate this aspect with our measures, the result in Table 15 (Section A.5) meets the general idea in DRL that IQN has better diversity capability than DQN.
>     * However, the RGSK (Section 4.1) and Go (Section 6.2) experiments measure human players, which can be considered a human-like measurement task where human-like is defined based on specific individuals. If a playstyle is not similar to any known human player, it can be considered as not known human-like, which shares the same idea as our Algorithm 1.
> * **Engagement with Prior Work**: You also raised the question, "why understanding and analyzing play styles itself is so complex," and noted the absence of prior works about playstyles.
>     * I would like to ask again if our discussion in Section 2.1 about approaches with "heuristic rules/in-game features/supervised learning/unsupervised clustering/contrastive learning/parametrized policies/discrete playstyle metirc" was not clear enough, or did it just miss the part about player modeling?
>         * For player modeling, we can find the discussion in the major baseline of this paper (Playstyle Distance): "Although it is known that player modeling has been a common domain in studying player behaviors (Yannakakis et al., 2013)..."
>             * It indicates that player modeling usually focuses on "capturing some playing behaviors" rather than the complex combinations of behaviors and their more general intention or preferences (playstyle). Thus, we did not specifically discuss player modeling but focused more on player policies. If you consider player modeling important and think it should be included in this paper, we can add some paragraphs to describe this topic and its correlation with playstyles.
> * **Technical Clarifications**: This paper **does not use game's internal states** but processes **game screen images** directly, which is why discrete playstyle measures are crucial and distinct from traditional player modeling.
>     * If there is any confusion about this, please review this paper and possibly the major baseline paper (Playstyle Distance) again, as this is the foundation of discrete playstyle measures. Without understanding this premise, there might be a misjudgment of this paper.
> * **Evaluation Setup**: Please see Section 4.2, where we describe the zero-shot classification before Section 5, and Figure 2d illustrates this task if our methodology in Sections 5 or 6 was unclear.
>     * Also, the dataset size refers to the number of observation-action pairs counts for one playstyle. For example, if we want to classify a query set against 200 reference sets, and the dataset size is 1024 for each playstyle, there are (1+200)x(1024) observation-action pairs including query and reference sets.
>     * Regarding "SL" and "CL", we discuss these in Sections 2.1 and 5.1. These are two learning paradigms that require some playstyle labels, and there are **no playstyle labels** for training. Thus, it would clearly be a random model, and it is **impossible to train these types of methods without labels**, otherwise, they would be classified apart from pure supervised or contrastive learning.
>
> I invite the reviewer to verify whether my understanding of your comments is accurate or if specific concerns or changes are still needed for improving this paper after a deeper understanding of it.

---

> > ### Comment · Reviewer_293a · 2024-06-20
> > **Re: Official Comment by Authors**
> >
> > I apologize for the delay in response as I've been travelling.
> >
> > * Motivation Concerns: I apologize for the lack of clarity. I recognize the argument from Section 1 that understanding playstyles is complex, but it's unclear to me why this means we need better/more accurate similarity measures. There are many ways to understand play styles, so I'm asking why this specific approach is being considered, what are the motivations for pursuing it. I'm asking for this as it will allow me to better understand the evaluations in context.
> > * Engagement with Prior Work: I agree that, that is an accurate summation of the 2013 survey paper, but there have been many relevant papers since then. Particularly relevant is so-called "data-driven player modeling" and particularly those with the research objectives of Player behavior modeling or Experience modeling. These often look much more like what the paper focuses on in terms of in-depth models of player behaviour. It's common to now see player modeling work where the log of a player's in-game actions are directly used such as in [2].
> > * Technical Clarifications: I understand that the approach uses game screen images.
> > * Evaluation Setup: I have read Sections 4.2, 5, and 6. I'm afraid they do not answer my questions. Yes, I understand that SL and CL had to be random chance, my concern was more that this was not stated in the paper.
> >
> > [1] Hooshyar, Danial, Moslem Yousefi, and Heuiseok Lim. "Data-driven approaches to game player modeling: a systematic literature review." ACM Computing Surveys (CSUR) 50.6 (2018): 1-19.
> > [2] Sarkar, Anurag, and Seth Cooper. "Using a disjoint skill model for game and task difficulty in human computation games." Extended Abstracts of the Annual Symposium on Computer-Human Interaction in Play Companion Extended Abstracts. 2019.

---

> > > ### Author Response · Authors · 2024-06-21
> > >
> > > > Motivation Concerns
> > >
> > > For end-to-end playstyle identification (from raw gameplay without internal or heuristic game states) with formal quantification of similarity, there are only two measures to my knowledge: [1] and [2]. Previous studies, especially in player modeling, model-tree, and data-driven methods, focus on feature-level analysis or visual-level observations, such as using t-SNE for latent distribution.
> > >
> > > You mentioned "Data-Driven Approaches to Game Player Modeling," which surveys from 2008 to 2016. The latest study in the clustering part is by Cowley et al. (2014), and there is no mention of VAE (proposed in 2013). It is clearly more about traditional methods. Another example you mentioned, "Using a disjoint skill model for game and task difficulty in human computation games," is about player skill level (related to difficulty). This is a specific and popular topic but far from general playstyles (players with the same skill level can exhibit different playstyles, as seen in many professional game players).
> > >
> > > Some recent player modeling studies use latent clustering (traced from "Player modeling," 2013), but most focus on the feature level or lack formal definitions or quantifications to be a metric or measure. If you know of any more formal methods for end-to-end playstyle metrics besides [1] and [2], please let us know for reference.
> > >
> > > - [1] Chiu-Chou Lin, Wei-Chen Chiu, I-Chen Wu. "An unsupervised video game playstyle metric via state discretization." UAI 2021.
> > > - [2] Reid McIlroy-Young, Yu Wang, Siddhartha Sen, Jon M. Kleinberg, Ashton Anderson. "Detecting Individual Decision-Making Style: Exploring Behavioral Stylometry in Chess." NeurIPS 2021.
> > > - Both discuss the lack of effective metrics for this type of method (from raw gameplay).
> > >
> > > Especially, [1] is more general than [2] for cases without playstyle groups in training data. One motivation for this paper is to improve this unsupervised metric for better accuracy. Besides improving playstyle identification, another benefit is measuring agent diversity and similarity between AI agents and humans. Currently, it is not easy to measure agent diversity, with many methods still using t-SNE to observe latent distribution and toy examples to count tabular state visit counts.
> > >
> > > Our new version (20 Jun 2024) has changed significantly from the previous one, and the major parts in Sections 1 and 2 address your concerns from player modeling and motivations. If there are still unclear parts or the motivation is not well described, please help us identify the weaknesses for polishing our revision.

---

> > > > ### Comment · Reviewer_293a · 2024-06-21
> > > > **Revision**
> > > >
> > > > Thanks to the authors for the work put into the revision, this has definitely addressed my first and second concerns. I appreciate the added clarity and engagement with prior work. The revision has also addressed my third concern in terms of clarity around the evaluation. However, as with reviewer ycj8, I am left with my fourth concern around the strength of the evaluation and how it supports or does not support the claims in the document.

---

> > > > > ### Author Response · Authors · 2024-06-21
> > > > >
> > > > > Thank you for your dedicated engagement and effort in reviewing this paper and providing detailed comments.
> > > > >
> > > > > Regarding the fourth remaining concern, are the following phrases considered overclaimed in the current version? Would rewording them to make weaker claims address these concerns? Alternatively, what kind of experiments or descriptions are missing for these concerns? We are trying to meet the standards of TMLR and check which parts of our paper lack evidence for our claims.
> > > > >
> > > > > - > "Beyond gaming, playstyle measures could be pivotal for other decision-making topics, ..."
> > > > > - > "These techniques not only improve precision in playstyle measurement but also shed light on understanding similarity through the lens of human cognition."
> > > > >
> > > > > We sincerely appreciate your assistance in helping us refine these aspects to ensure our claims are appropriately supported.

---

> > > > > > ### Comment · Reviewer_293a · 2024-06-24
> > > > > > **Re: Official Comment by Authors**
> > > > > >
> > > > > > Apologies for the delayed response again. For those two claims, I think the first is fairly speculative but the "could" means I have no issue with it. For the second, I have concerns around "improv[ing] precision in playstyle measurement". This is a fairly broad claim. My understanding of the results is that there's a sweet spot based on dataset size where the presented metric does better than ablations or variations. By sweet spot here I simply mean that if there's too little they're all equally bad and if there's enough they mostly seem to converge to the same point. The amount of improvement is also fairly minor, or at least seems to be from the presented results. It's unclear if we can qualify this as improvement. Thus I'm not sure the results broadly support a general claim around improving precision in playstyle measurement. I hope that's clear but let me know if not, and I'd also welcome any corrections of any misunderstanding on my part.
> > > > > >
> > > > > > As I perhaps too abstractly proposed in my original review, I'd suggest some additional evaluations to demonstrate that there's substantial or significant improvement using this metric compared to existing approaches, which would more clearly demonstrate improvement at least to me. Alternatively, the claim could be restated to gesture at some of my concerns above.

---

> > > > > > > ### Author Response · Authors · 2024-06-25
> > > > > > >
> > > > > > > Thank you for your response and patience.
> > > > > > >
> > > > > > > We will revise some parts of the claims of contribution in the introduction and conclusion to make the contributions clearer in the coming revision.
> > > > > > >
> > > > > > > Since some contributions, like addressing the trade-off between small or large state spaces in Playstyle Distance with the multiscale approach, are still based on foundational evaluation tasks like classification or its extended applications, do you think adding a comparison with the latent distance of observations (analogous to the FID in GAN evaluations for generative styles) can help judge the effectiveness of these playstyle measures?
> > > > > > >
> > > > > > > We would like to add this comparison in Section 5.3 Full Data Evaluation. Additionally, for Figure 5, do you think adding the original baseline of Playstyle Distance (using a $2^{20}$ state space, not the multiscale version we presented) would help further judgment, or are the evidences in Table 2 sufficient to justify the effectiveness of multiscale and keep clear comparisons with different measure formulations in Figure 5?
> > > > > > >
> > > > > > > We sincerely appreciate your assistance in refining these aspects to ensure our claims are appropriately supported.

---

> > > > > > > > ### Comment · Reviewer_293a · 2024-06-28
> > > > > > > > **Re: Official Comment by Authors**
> > > > > > > >
> > > > > > > > I apologize again for the delay, I think both that metric and that baseline would be helpful to understand the approach but there might be fundamental problems around where the approach is applicable that would remain.

---

> > > > > > > > > ### Author Response · Authors · 2024-06-29
> > > > > > > > >
> > > > > > > > > Thank you for your response.
> > > > > > > > >
> > > > > > > > > Do you mean the results in the new subsection 5.5 are still not sufficient to examine the effectiveness of playstyle measurement?
> > > > > > > > >
> > > > > > > > > To gain a clearer direction, could you please specify the areas where the problem lies?
> > > > > > > > > Is it more playstyle datasets/platforms required for evaluation, more methodology baseline comparisons needed, additional tasks besides the classification task necessary to quantify the efficacy of similarity measures, or is a different index on the confusion matrix, such as the F1 Score, intended to be checked?

---

> > > > > > > > > > ### Comment · Reviewer_293a · 2024-07-07
> > > > > > > > > > **Re: Official Comment by Authors**
> > > > > > > > > >
> > > > > > > > > > Thanks to the authors for their continual work to improve the paper. I finally had the opportunity to review the new revision and found the new results and text very convincing. The new revision has substantially improved the paper from my perspective and the authors can now consider all my concerns (outside of those raised by the other reviewers) to now be addressed.

---

### Review · Reviewer_rjfA · 2024-06-10

**Summary Of Contributions:**

The paper proposes an improvement of the Playstyle Distance for comparing policy style and diversity for deep reinforcement learning (DRL). More precisely, a Playstyle Similarity is introduced, with a three-fold contribution: (i) introducing multiscale analysis with varied state granularity, (ii) using a similarity measure that aligns to human perception instead of a distance metric, and (iii) considering all states visited by the compared policies, not just the common ones. An algorithm is proposed for measuring policy diversity. Experiments are performed on multiple tasks: TORCS, RGSK, Atari games (7 tasks), Go, and 2048, using pretrained policies.

**Audience:**

Yes

**Broader Impact Concerns:**

The paper contains a brief impact statement. I have no unaddressed concerns on this topic.

**Claims And Evidence:**

Yes

**Requested Changes:**

- It would be great to include formal definitions of distance, similarity, and kernel functions.
- Fig. 1 introducing the Playstyle Distance framework could benefit from step numbers on the arrows, showing the pipeline from input to output. The same figure could also be extended to compare the original Playstyle Distance and the modifications introduced in this paper. Currently, there is no figure representing the proposed Playstyle Similarity.
- Results' presentation: the efficacy comparison experiment is spread over three figures and two pages; it seems these figures could be fit in half a page. The reader would want to easily compare similar plots, especially when they are part of the same experiment. Using subplots, shared labels and potentially shared axes would improve readability. This could be applied to other experiments in the paper.
- Results' presentation: numerical results in Tab. 5-14 could arguably use rescaling, i.e., extracting a $10^{-3}$ factor in the table headers.
- Results' presentation: please highlight best results in bold in tables where it makes sense.

**Strengths And Weaknesses:**

Strengths:
- The paper covers a topic that is relevant to the reinforcement learning community.
- The experiments are quite extensive for the datasets and setups considered.
- The overall writing style is clear (some typos remain).
- The proposed Playstyle Similarity seems indeed to improve over Playstyle Distance, both by extending the concept and by displaying better performance.

Weaknesses:
- It is unclear to me what are the impact or implications of having alignment with human perception for the proposed exponential kernel. The kernel is applied to quantities over which humans do not have specific perception or intuition. Moreover, there are no experiments provided to support the human perception angle. These points weaken the interest of perceptual alignment for the present application.
- The method introduced in Algorithm 1 for measuring policy diversity functions similarly enough to zero shot classification and nearest neighbor rule, having limited interest to the reader. I would argue that it could be deferred to the Appendix, where its experimental results are introduced.
- The downside of complete or extensive experiments in the way they are currently presented is that it is difficult to interpret and draw conclusions from the multitude of plots and tables. Moreover, the results' interpretation seems a bit superficial. Otherwise stated, it is not clear how one would parametrize the proposed method for usage on a new dataset or task.
- Additionally, the results' presentation could further be improved. Please find a few suggestions in Requested Changes.
- The presentation of experimental setups as bullet points (p. 8-9) takes significant space. There is an opportunity to inline these and move some of the Appendix content to the main paper.

---

> ### Author Response · Authors · 2024-06-11
>
> Thank you for your response.
>
> There are two main reasons why a perceptual kernel is necessary:
> * First, using a probability value to describe similarity is more direct and understandable to humans than a distance value. For example, the values in Table 6 are negative distances (Playstyle Distance), which are difficult for humans to interpret as measures of similarity. In contrast, the probability values in Table 7 (Playstyle Intersection Similarity) range from 45% to 80% similarity, which are intuitively understandable. This is particularly useful for applications of our measure, such as our diversity measurement, which uses a probability threshold instead of a distance value that may require specific adjustments for different settings.
> * The second reason concerns the nature of similarity itself, as described at the beginning of Section 3.2. A larger distance value conveys less similarity information. For example, two individuals may exhibit similar playstyles in most states, but differ significantly in one state. It is reasonable to decrease the similarity as the distance increases in this state, but the decrease should be limited. For instance, a distance of 100 may correspond to <0.1% similarity when using probability to describe similarity. If the distance increased to 1000, humans might perceive it as merely different in that state, without affecting the similarity in other states. However, using raw distance values, the averaging operation would disproportionately influence the final similarity. In our experiments, using the suggested W2 distance on discrete action spaces minimally affects the results due to the small upper bound of this distance. If using other distances, like the Bhattacharyya distance, which can exceed thousands, it could pose significant problems.
>
> Thank you also for the suggestion regarding presentation. It seems challenging to clearly describe and provide sufficient evidence within the regular submission length of this paper. We plan to change the submission to a longer format to incorporate suggestions from all reviewers.
>
> Finally, I would like to check whether the definitions of distance, similarity, and kernel functions in Section 3 are unclear, or if other sections also need more formal explanations of these concepts.

---

### Review · Reviewer_ycj8 · 2024-06-11

**Summary Of Contributions:**

The submission concerns the question of quantifying the similarity of behaviors defined as trajectories of state-action sequences generated by some policy (human or synthetic). The technical work extends a prior trajectory-based metric that identifies similar trajectories using a cache of prior examples. The new technical extensions are:
1. Reweighting samples using an exponential weighting scheme to more closely match human similarity perception scaling properties.
2. Adjustments to the base metric to use a hierarchy of different state granularities and to use the Jaccard index (instead of a simple intersection).

The experiments provide evidence to support these contributions by comparing the baseline metric definition to extensions that incorporate the state hierarchy, perceptual reweighting, and variations thereof on the task of classifying a new trajectory to match prior playstyles (that is, to trajectories generated by the same policy). This shows success in racing games using either human or AI-generated traces and success in Atari with AI-generated traces. Additional experiments assess the robustness of the metric (and Jaccard index) when used on games with high stochasticity due to the game environment (2048) or due to opponent behavior (Go).

The submission contributes new knowledge of how to define similarity metrics on trajectories that more accurately identify the generating policy, specifically:
1. That the Jaccard index is a good proxy for many games but fails under high stochasticity.
2. That perceptual reweighting (inverse exponential scaling) can improve accuracy in many cases.
3. That a single sample threshold is typically sufficient for filtering trajectories, where prior evaluations suggested a higher threshold may be necessary for good performance.

**Audience:**

Yes

**Broader Impact Concerns:**

The broader impact statement already addresses potential ethical implications for privacy by being able to target advertising using behavior alone. As a technique for analyzing user behavior to identify (known) styles or users the technique has limited additional ethical concerns beyond these uses to limit user behavioral tracking with other methods.

In theory the technique can foster improved reinforcement learning algorithms using notions of similarity among policies. But this downstream application is generic and does not merit any additional mention.

**Claims And Evidence:**

No

**Requested Changes:**

# critical
- Add results on multiple trials for the experiments and statistical testing to support any claims made about differences among methods. This should include quantification of variability and effect sizes of ablations / different algorithms.
- Provide a narrative overview of the experiments and their results that clearly articulates how important different components of the proposed metric are and under which conditions.


# strengthen
- Provide a figure to illustrate the Go state encoding scheme (even if only in the appendix). Try to clarify the text to make it easier for readers to understand.
- Section 4.2
	- It would help to have a table that summarized the experiment configuration parameters. Particularly include at least the number of styles for the game and the state and action space size.
- Equation 3
	- Is $D(\pi_X, \pi_Y)$ here $B_D$?
- Section 5.1
	- This section would benefit from including results on Atari games as well. The evidence in Table 1 does not support the notion that a sample threshold is necessary (see Table 1 below) and the Atari games may clarify why that was the case for the TORCS and RGSK data.
- Table 1
	- t=2 is worse in all cases for the intersection calculation. So even the baselines did not need a sample count threshold. This weakens the claim that the multi-scale hierarchy has this benefit (made in Section 5.1).
	- On RGSK the $2^{20}$ state space is nearly the same as the mix model. Why is that true?
		- Is this related to the analysis of RGSK state space sizes in Appendix A.2.2?
- Table 3
	- On first 32 moves: Why does "Inter BC Similarity" do so much worse than "Intersection Similarity", but not with fewer first moves?
- Figure 3
	- What are the shaded areas?
		- Please include this in the captions to help readers.
		- Figure 5 in the appendix indicates 3 encoder models and a min-max range. If this is the case for other figures please add that information.
- Figure 4
	- The figure is very difficult to read due to being very small. It would help to make this figure larger.

**Strengths And Weaknesses:**

# strengths
- Evaluations include a wide variety of games (two racing games, Atari games, 2048, and Go) and sources of policies (human and bot-generated, including policies trained using reinforcement learning for varying amounts of time).
- Tests check both deterministic (or nearly so) and non-deterministic environments.
- Strong evaluation methodology: ablates the core algorithm to show the importance (or lack thereof) of algorithm additions.
- Good performance depends on few trajectories (~256-512 for racing games; ~64-128 for Atari).


# weaknesses
- In general I found myself losing the narrative of the paper and having to go back to get the big picture. The submission would benefit from some overview of the main outcomes and conclusions. There is some of this for the methodological changes, but the narrative threads get lost in the evaluations.
- The experiments did not include results from multiple seeds (in at least some cases) and lack any reporting of confidence intervals, or variability of results. No statistical tests of differences among conditions were reported. While many differences are large from visual inspection (some of the figures), many are not so large as to be obviously different. Tables do not report any variability. These points should be addressed to clarify the magnitude of improvements from the new techniques.
- The Go state description is difficult to follow. It would benefit from a figure and revised text discussion.
- Most discussion of the assessment of diversity using the metric is in the appendix. The submission would benefit from being longer to push this into the main text.
- Many changes to the baseline algorithm have only modest effects on classification accuracy. For example, the racing game results only need to add intersection. The paper would benefit from more clearly summarizing in which games (or policy sources) the changes helped and which they did not. At least some modifications were clearly helpful, but I found myself doing a lot of work to piece together the big picture of which were the minimal changes to get the biggest improvements (and when).
- Minor stylistic note: the paper sometimes references a variable like "N" ($Style_n$, $N$ first moves in Go, MxM games query) without explaining this will be varied in the experiments. That was confusing and took a while to sort out. It would help to make that clearer up front.

---

> ### Author Response · Authors · 2024-06-11
>
> Thank you for your response.
>
> We plan to switch this regular submission to a long submission to enhance the presentation and include detailed statistical results in the main paper.
>
> Regarding the questions in Requested Changes:
> * In Equation 3, $D$ represents a distance function, and we prefer using the W2 distance.
> * About the RGSK state space, a $2^{20}$ state space provides sufficient information for these playstyles. Including a larger state space contributes very little due to few intersection states. This matches the results in Figure 6 (Section A.2.2), which show that at least $2^8$ samples are required to aid playstyle classification.
> * Regarding Inter BC Similarity in Go experiments, the Bhattacharyya coefficient tends to be sensitive when two distributions have a large distance due to its geometric property of overlapping regions. For later games in Go, action distribution becomes more noisy if we do not consider the real situation on the board, leading to a high variance in BC value, whereas the value of W2 distance remains relatively stable.
> * All shaded areas in the figures indicate the min-max range of average results from different discrete state mappings (3 different discrete encoder models).
>
> Finally, I would like to understand whether the concerns about claims and evidence are related to the presentation and experiment details, or if there are other factors such as the design of the experiments being unconvincing or needing more applications and use cases for playstyle measures.

---

> ### Author Response · Authors · 2024-06-13
> **Revision of Go Experiment Methodology**
>
> We are currently implementing the HSD encoder for Go. Our preliminary results suggest that the new method in this paper can enhance Playstyle Distance. Would it be clearly beneficial to replace the current Go experiments, which use heuristic discretization (via move number on first N moves), with experiments using the learned discrete encoder instead?

---

> > ### Comment · Reviewer_ycj8 · 2024-06-20
> >
> > >We are currently implementing the HSD encoder for Go. Our preliminary results suggest that the new method in this paper can enhance Playstyle Distance. Would it be clearly beneficial to replace the current Go experiments, which use heuristic discretization (via move number on first N moves), with experiments using the learned discrete encoder instead?
> >
> > This would be a welcome addition that makes the experiments easier to think about and compare. It would also make the text simpler by removing the need to explain a complex manual discretization process.
> >
> > >All shaded areas in the figures indicate the min-max range of average results from different discrete state mappings (3 different discrete encoder models).
> >
> > Am I correct in understanding this means 3 trials?
> >
> > > I would like to understand whether the concerns about claims and evidence are related to the presentation and experiment details
> >
> > My primary interest is seeing clearer empirical evidence with more rigorous methodology in terms of repeated trials and statistical evidence of differences. That was followed by the presentation concerns. (That is, the two points made in "critical" in my evaluation). Thank you for checking!

---

> > > ### Author Response · Authors · 2024-06-21
> > >
> > > > Am I correct in understanding this means 3 trials?
> > >
> > > Not exactly. For each encoder and the corresponding dataset size (sampled from the corresponding playstyle playing data), we run 100 rounds of random subsampling (performing 100 classification tasks with different sampled datasets).
> > >
> > > For example, in the revised version of Table 2, for a classification task in TORCS with 25 reference playstyles, we compute 25+1 measure values for each classification prediction and perform 100x25 classification predictions for every type of reference playstyle, each type undergoing 100 predictions with subsampling.
> > >
> > > If a trial means an average accuracy using one encoder and performing 100x(Playstyle Count) predictions, then it is 3 trials.
> > >
> > > Besides, does our new Table 2 meet your expectations for more detailed statistics on multiscale state space?
> > >
> > > For the experiments in Section 6 (2048 and Go), we do not use random subsampling from a playstyle dataset. The comparison uses the playing trajectory from a single game or several games. We only use one discrete encoder in our new Section 6.2. If you think that more statistical results for Go using different encoder models trained from different random seeds are necessary, please notify us, and it will require several days for experiments.

---

> > > > ### Comment · Reviewer_ycj8 · 2024-07-01
> > > >
> > > > Thank you for clarifying the experiment settings. It may be better to split the results by encoder and use the variation among the 100 rounds of sampling as the variability metric. This comment would apply to many of the experiments and may clutter the results, but it also seems like the more "fair" comparison: for a given encoder, how variable are the outcomes and which metric that uses that encoder produces superior results. Averaging over encoders is hard to interpret as it combines samples across distinct methods (data representations) and then compares how well different algorithms use those.
> > > >
> > > > Overall comment:
> > > > - I'm still getting lost in the details in the experiments section. What would help me a lot (even if only in these comments): Could you succicently (1-2 sentences) state what claim each experiment is intended to assess and what the resulting evidence is for that claim?
> > > > - Explaining the above and then tying it together into the overall picture would go a long way to me understanding the strength of evidence for the method and the areas in which it improves over prior efforts. Right now the experiments are so detailed that it is hard to "see the forest for the trees", so to speak.
> > > >
> > > > # second revision comments
> > > > - Table 2: It looks like $2^{20} t=1$ and both $mix$ models overlap. The text (and caption) claims "It is evident that using a multiscale state space (mix) offers the best accuracy", but the values seem to be the same or within the deviation measure (for Pong and TORCS where the values are not nearly identical). Am I missing something as to why it's clearly better? Or why it removes the sample threshold (since t=1 is no sample threshold either)? I understand the original text claimed the need for a sample threshold, but the results here clearly contradict that.
> > > > - The new explanation (v1) in 5.2 is helpful, thanks!
> > > > - minor: "Results presented in Figure 4 suggest that probabilistic similarity can be a good alternative to distance-based similarity, offering improved explainability in terms of measure values." I'm not sure how Figure 4 provides evidence of interpretability. Maybe I'm misreading the sentence.
> > > > - Table 3: I have no idea how to read this table. The caption is quite confusing. I vaguely understand from the text that this is intended to represent decreasing similarity values for greater levels of noise or difference in speed setting. But the metric and table are not really discernable to me. What is the scale of these values? What do the points on that scale indicate? Maybe this could be boiled down to a simpler metric like whether a given sample is closest to the target class.
> > > > - Table 4: Maybe this is the boiled down version I'm looking for. It seems like everything mostly fails at the Center Case?
> > > > - "The Go dataset used in this study was sourced from Fox Weiqi and provided by the team of the MiniZero framework" What license was this obtained under? It is not clear what the EULA the data is covered by and what consent was used to obtain it.
> > > > - Table 6: It looks like at $M=10$ or more all methods do roughly the same with two exceptions: the smallest $4^8$ model and the Jaccard Index. I'm not sure this is strong evidence for the claim that "the Bhattacharyya coefficient is more effective when a playstyle plays very diversely in a state due to its distribution property that rapidly decreases similarity with few overlapping outcomes, and our Go result is an example"
> > > >
> > > > Overall I am not sure I find the claims about needing a mixed state space strong from the new evidence. It seems that a modestly granular discrete encoding with no overlap filter suffices for all these tasks. Jaccard index is clearly a bad choice under high stochatisticity or adversarial play. I'm not sure what to make of the claims about similarity metrics as the results seem to asmpytotically converge in most figures. But perhaps my impression is false: would you be able to list the explicit results rejecting this interpretation? That could help move the conversation forward.
> > > >
> > > > To be clear: I think the paper has interesting evidence and investigates important questions. But I cannot follow the narrative and find the evidence mixed for some of the particular claims being made.

---

> > > > > ### Author Response · Authors · 2024-07-01
> > > > > **# Intention of Experiments**
> > > > >
> > > > > * **Section 5: Whether new measures can help playstyle measurement in video games?**
> > > > > 	* **5.1: Intention:** Multiscale state space can help improve accuracy and make it easier to choose a proper state space (including possibly threshold count $t$).
> > > > > 		* **Evidence:** On TORCS, mix is better than using any state space alone.
> > > > > 		* **Evidence:** On RGSK, $2^{20} t=1$ is good enough, but using mix does not require considering which setting to use (there are 5 possible settings).
> > > > > 		* **Evidence:** On Atari, mix and $2^{20}$ are nearly the same, and it is still easier to choose state space with mix (just using all spaces).
> > > > > 	* **5.2: Intention:** Probabilistic similarity is a good alternative to distance similarity, and the measure values of probability share the same form in different games (via likelihood). For distance similarity, we must first know the property's and distribution of distance to explain the meaning of the measure value.
> > > > > 	    * **Evidence:** The result with probabilistic similarity is not worse than distance similarity and slightly better on TORCS (containing slightly different playstyles).
> > > > >     * **5.3: Intention:** Perceptual kernel + Jaccard index have better accuracy than distance similarity on several video games.
> > > > >         * **Evidence:** Playstyle Similarity performs the same or better than Playstyle Distance on all platforms and dataset sizes.
> > > > >         * **Evidence:** Jaccard index is effective in video games with no high randomness.
> > > > >     * **5.4: Intention:** New measures have a better ability for playstyle sorting, not only to identify max similarity.
> > > > >         * **Evidence:** Table 3 shows the trend of measure values (e.g., 0.0753 implies the query has a 7.53% probability of being Speed60N0).
> > > > >         * **Evidence:** Table 4 shows that Corner Case is simpler for sorting, and measures with perceptual kernel have improvements. Center Case is hard to get consistency, especially on both increasing and decreasing speed styles.
> > > > >     * **5.5: Intention:** Discrete playstyle measures have an advantage over other potential unsupervised measures, common in generative styles.
> > > > >         * **Evidence:** FID or Cosine Similarity on latent features is very dependent on games, and Playstyle Similarity is generally better.
> > > > >         * **Evidence:** The values of Discrete playstyle measures are the similarity of player policy defined on action space. The meaning of latent-based similarity is not clear and depends on implementation.
> > > > > * **Section 6: If the game has high randomness, can discrete playstyle measures still work?**
> > > > >     * **6.1: Intention:** Demonstrate the claim that Jaccard index may not be suitable sometimes.
> > > > >         * **Evidence:** Jaccard index hurts accuracy in 2048 with raw board as discrete states.
> > > > >     * **6.2: Intention:** Can Discrete playstyle measures work in a multi-player board game?
> > > > >         * **Evidence:** Discrete playstyle measures work on Go and achieve over 90% accuracy in 200 player identification with sufficient samples.
> > > > >         * **Evidence:** The choice of selecting a proper state space matters in different cases. Combining our methodology alleviates this problem.
> > > > > * **Section 7: Demonstrate an application on DRL using our playstyle measure.**
> > > > >     * **Evidence:** The diversity meets the intuition to the understanding of these DRL algorithms.

---

> > > > > ### Author Response · Authors · 2024-07-01
> > > > >
> > > > > Thanks for your response.
> > > > >
> > > > > We have logged the corresponding mean and standard deviation of each encoder in Table 2. If you consider this detailed statistic can help judge the effect of using different algorithm settings, we can add them to the appendix (those results are not much different from Table 2).
> > > > >
> > > > > Since our new methods are still based on a learning-based discrete encoder, analyzing only one encoder would be suspicious of cherry-picking; thus, we report the average result in most experiments, analogous to using the ensemble result from 3 encoders. We do not plan to report the single encoder version in current experiments. If you consider the results with the first encoder in video games necessary for publication and have no specific concerns about reporting only one encoder result, please let us know to add them to the appendix (or main paper to replace current results).
> > > > >
> > > > > # Minor change checks before revision and some explanation to questions
> > > > >
> > > > > * For Table 2, it is indeed not evident to have general accuracy improvement (only TORCS shows evident results), we will revise the advantage to handy on state space selection and potentially get accuracy improvement.
> > > > > * Probabilistic similarity is more interpretable since it is a direct factor to likelihood. Distance similarity needs more info like mean, std, and distance meaning to interpret the degree of similarity.
> > > > > * Values in Table 3 are the actual measure values. It is an example to demonstrate the trend of value changes corresponding to style changes. Table 4 provides a clearer comparison in this task and everything mostly fails at the Center Case, especially on the decreasing part of the speed style.
> > > > > * Go datasets from Fox Weiqi were not collected by our team, and we do not have the identification mapping to human players. It can be considered a private dataset rather than a publicly available dataset.
> > > > > * We discuss BC being better than W2 (Playstyle Similarity). Using $16^8$ or $256^{361}$ may be good enough, but this BC is a better claim based on the mix version of the state space.
> > > > >
> > > > > # Summary
> > > > >
> > > > > Overall, we agree that if we can obtain the best single state space, there is no need for mixed state space. However, mixed state space helps reduce the efforts of space selection and potentially get advantages like in TORCS.
> > > > >
> > > > > For the Jaccard index, if we know there are external factors that may disturb state visiting distribution like game randomness or other players, we can discard this part. But it is clear in video games with less randomness that incorporating all observations can improve accuracy. This idea and worry are described in Section 3.3 and 5.3.
> > > > >
> > > > > Thanks for your advice again.
> > > > >
> > > > > If you consider the training of new discrete encoders necessary for getting better accuracy with multiscale state space than any single state space in more games, please let us know.

---

> > > > > > ### Comment · Reviewer_ycj8 · 2024-07-06
> > > > > >
> > > > > > Thank you for the detailed replies!
> > > > > >
> > > > > > The clarifications about intentions and evidence are very helpful. Consider putting those into the text to help guide readers.
> > > > > >
> > > > > > These also help clarify the scope of the claims to be more modest (which is fine!): please make sure the text is consistent with the framing that these are sometimes helpful under some conditions, but practitioners would need to evaluate on a case-by-case bases.
> > > > > >
> > > > > > >Go datasets from Fox Weiqi were not collected by our team, and we do not have the identification mapping to human players. It can be considered a private dataset rather than a publicly available dataset.
> > > > > >
> > > > > > Even if it is a private dataset, if that was collected in violation of the EULA or without ethics board review then it is a problem. What approval was gathered for it?
> > > > > >
> > > > > > >If you consider the training of new discrete encoders necessary for getting better accuracy with multiscale state space than any single state space in more games, please let us know.
> > > > > >
> > > > > > I do not. Thank you for checking!
> > > > > >
> > > > > > >Since our new methods are still based on a learning-based discrete encoder, analyzing only one encoder would be suspicious of cherry-picking; thus, we report the average result in most experiments, analogous to using the ensemble result from 3 encoders.
> > > > > >
> > > > > > Ah, to clarify: the goal in reporting for an encoder is to make a fair comparison to the matching method. I understand the cherry-picking concern and it is good to report all 3 encoders separately somewhere.
> > > > > >
> > > > > > >For Table 2, it is indeed not evident to have general accuracy improvement (only TORCS shows evident results), we will revise the advantage to handy on state space selection and potentially get accuracy improvement.
> > > > > >
> > > > > > I'm not quite sure I follow the change being made, but look forward to the results!
> > > > > >
> > > > > > > this BC is a better claim based on the mix version of the state space.
> > > > > >
> > > > > > Given that BC is only better under certain conditions the claim should reflect that specificity. Stating "BC is better than W2" is misleading as that is not generally true across conditions.
> > > > > >
> > > > > > >Probabilistic similarity is more interpretable since it is a direct factor to likelihood. Distance similarity needs more info like mean, std, and distance meaning to interpret the degree of similarity.
> > > > > >
> > > > > > This is a good clarification! Please make sure it is highlighted in the text.
> > > > > >
> > > > > > **Please note** that the experiments still lack statistical tests for differences, part of the original comments I made (quote below). Comparing matching encoders for differences when varying the perceptual kernel is one such test:
> > > > > > > statistical testing to support any claims made about differences among methods
> > > > > >
> > > > > > These tests should help clarify the points above about which differences over methods are important. Be sure to be clear about which data is being pooled (across 3 encoders?) and which conditions are being compared (difference when using BC vs W2?).

---

> > > > > > > ### Author Response · Authors · 2024-07-07
> > > > > > >
> > > > > > > Thank you for your reply.
> > > > > > >
> > > > > > > We have updated the new revision with LaTeX color magenta. The revised parts correspond to the following questions:
> > > > > > >
> > > > > > > The Go datasets from Fox Weiqi follow the privacy guidelines in the default approval by users to play games on that platform.
> > > > > > > (https://edu-foxwq-com.translate.goog/complex/privacyguidelinesv2.htm?_x_tr_sl=auto&_x_tr_tl=en)
> > > > > > >
> > > > > > > For detailed statistics for each encoder with a multiscale state space, we added a new Section A.3.1 and also a hyperlink in the caption of Table 2.
> > > > > > >
> > > > > > > We also incorporated some explanations from the reply (Intention of Experiments) into the new version to clarify the intention of our methods.
> > > > > > >
> > > > > > > For the case where BC has some advantage, we revised the description in Section 6.2, as it is a specific case in Go. There is no clear difference in video games with discrete actions.
> > > > > > >
> > > > > > > If there are still some requested changes we missed, please remind us again. Thank you for your help.

---

> > > > > > > > ### Comment · Reviewer_ycj8 · 2024-07-09
> > > > > > > > **please address statistical tests comment**
> > > > > > > >
> > > > > > > > Thank you for the further improvements to the text.
> > > > > > > >
> > > > > > > > Regarding requested changes, the first of my two "critical" comments has not been addressed. This is a point I highlighted in the comment above as well:
> > > > > > > >
> > > > > > > > > *Please note* that the experiments still lack statistical tests for differences, part of the original comments I made (quote below). Comparing matching encoders for differences when varying the perceptual kernel is one such test:
> > > > > > > > > > statistical testing to support any claims made about differences among methods
> > > > > > > > >
> > > > > > > > > These tests should help clarify the points above about which differences over methods are important. Be sure to be clear about which data is being pooled (across 3 encoders?) and which conditions are being compared (difference when using BC vs W2?).

---

> > > > > > > > > ### Author Response · Authors · 2024-07-09
> > > > > > > > >
> > > > > > > > > Thank you for your reply. I apologize for my misunderstanding about the statistical test requirements, and I am still somewhat confused about this part.
> > > > > > > > >
> > > > > > > > > Could you please clarify if you are suggesting that we report p-values for each improvement compared to the baseline (perhaps Playstyle Distance) or some other specific index on the confusion matrix? Are you referring to reporting the mean and standard deviation of accuracy for each discrete encoder across all experiments, rather than just the multiscale state space in Section 5.1? Or are you specifically interested in the mean and standard deviation for each encoder on the perceptual kernel (Section 5.2)?
> > > > > > > > >
> > > > > > > > > More specifically, what statistical indices would you like us to check for which experiments? Depending on the target index, the presentation format may need to be very different from what is currently reported in the main paper (where we report accuracy, the most common index in classification tasks).
> > > > > > > > >
> > > > > > > > > Thank you for your guidance and continued support.

---

> > > > > > > > > > ### Comment · Reviewer_ycj8 · 2024-07-10
> > > > > > > > > >
> > > > > > > > > > >Could you please clarify if you are suggesting that we report p-values for each improvement compared to the baseline (perhaps Playstyle Distance) or some other specific index on the confusion matrix?
> > > > > > > > > >
> > > > > > > > > > > what statistical indices would you like us to check for which experiments?
> > > > > > > > > >
> > > > > > > > > > Good clarifying questions. I don't have any strong opinions on this. Whatever indices you believe best reflect the claims you wish to make in the paper. My comment was that the paper makes many claims about one method or approach being better than another. In the cases such a claim of any difference is made, they should have an accompanying test of differences.
> > > > > > > > > >
> > > > > > > > > > Some random examples from the reply you made above, which are not comprehensive of what I am requesting, but providing something to illustrate the idea:
> > > > > > > > > >
> > > > > > > > > > >Evidence: On TORCS, mix is better than using any state space alone.
> > > > > > > > > >
> > > > > > > > > > This should have a test showing mix outperforms using a single state space for each single state space reported. Accuracy seems to be the index used in the paper.
> > > > > > > > > >
> > > > > > > > > > Or another similar example:
> > > > > > > > > > > 5.3: Intention: Perceptual kernel + Jaccard index have better accuracy than distance similarity on several video games.
> > > > > > > > > >
> > > > > > > > > >
> > > > > > > > > > By contrast, I am not trying to convert literally everything into a quantitative claim. I understand the value of qualitative claims like:
> > > > > > > > > > >make it easier to choose a proper state space
> > > > > > > > > >
> > > > > > > > > > Hope that clarifies!

---

> > > > > > > > > > > ### Author Response · Authors · 2024-07-17
> > > > > > > > > > >
> > > > > > > > > > > We have uploaded a new revision (magenta color). Please refer to Section 8: Conclusion and Future Works and Appendix A.8.
> > > > > > > > > > > We use McNemar's test to report the p-values of the methodology changes to check the statistical significance, ensuring that the improvements are not due to sampling uncertainty.

---

### Author Response · Authors · 2024-06-20
**Revision Submission: Long Submission and Highlighted Changes**

We have revised our paper and updated the submission length to a Long Submission. Based on the reviewers' requested changes, we have made several modifications throughout the manuscript. These changes are highlighted in cyan for your convenience.

We kindly request the reviewers to review the highlighted sections to ensure all concerns have been addressed satisfactorily. Additionally, if you have not yet reviewed our previous responses to your comments, we would appreciate it if you could prioritize those as well.

Thank you for your time and effort in reviewing our paper.

---

### Author Response · Authors · 2024-06-26
**Second Revision**

Thanks all reviewers for valuable comments.

We have further revised the second version of our paper to provide additional evidence for our methods and to remove some potential overclaims.

Special thanks to Reviewer293a for highlighting that there may be a sweet spot regarding the data used and the discrete playstyle measures, indicating that these methods may be limited or not entirely convincing.

We have added a new subsection to Section 5.5 (Comparison of Potential Unsupervised Similarity Measures), where we compare some potential unsupervised similarity measures based on observation latent features common in generative models, including Euclidean Distance and Cosine Similarity.

The modified parts in this revision are highlighted in latex purple color (nearly red). The major revisions are in the latter part of Section 1, the initial paragraph of Section 5, the new Section 5.5, and Section 8. We have also adjusted some image layouts to improve presentation.

If there are still unclear or unsupported parts, please notify us so we can further polish this paper. Specifically, we seek feedback on whether the new experiments in revision 1 (cyan color) and revision 2 (purple color) provide clearer evidence for our methods or if there are additional aspects that need to be studied. The research on topics of end-to-end unsupervised playstyle, to our knowledge, is still not well-explored, and we need verification to ensure there are no incorrect claims or suggestions in these early explorations.

---

### Decision · Action_Editor_2iui · 2024-08-05

**Recommendation:** Accept as is

**Comment:**

While some of the reviewers aren't certain that the paper makes a distinctly novel or significant contribution, it meets TMLR's criteria that _the paper's claims are well supported and the findings are of interest to TMLR's audience_, and so is worthy of publication in TMLR.

**Audience:**

The paper extends prior work on playstyle distance with supported evidence of improvement.  There is an audience, even if small, for this line of work, and the improvements in the revision helps make the contributions and larger narrative more clear.

**Claims And Evidence:**

With the revised manuscript, and additional, including added statistical validation, the reviewers all agree that the experimental evidence supports the core claims of the paper.

---

> ### Author Response · Authors · 2024-08-15
>
> Thank you to all the reviewers and the action editor for your acknowledgment and feedback.
>
> We are currently preparing the code release and presentation video for the camera-ready version, which will take a few days and will be completed before September. This process includes an internal presentation to our lab members to gather questions and feedback for the presentation video.
>
> If you have any questions or points that you would like to see highlighted in our presentation slides for future readers, please feel free to provide them here.